# What Can Large Language Models Do for Sustainable Food?

**Anna T. Thomas** [1]   **Adam Yee** [2]   **Andrew Mayne** [1]   **Maya B. Mathur** [1]   **Dan Jurafsky** [1]   **Kristina Gligorić** [1]

## Abstract

Food systems are responsible for a third of human-caused greenhouse gas emissions. We investigate what Large Language Models (LLMs) can contribute to reducing the environmental impacts of food production. We define a typology of design and prediction tasks based on the sustainable food literature and collaboration with domain experts, and evaluate six LLMs on four tasks in our typology. For example, for a sustainable protein design task, food science experts estimated that collaboration with an LLM can reduce time spent by 45% on average, compared to 22% for collaboration with another expert human food scientist. However, for a sustainable menu design task, LLMs produce suboptimal solutions when instructed to consider both human satisfaction and climate impacts. We propose a general framework for integrating LLMs with combinatorial optimization to improve reasoning capabilities. Our approach decreases emissions of food choices by 79% in a hypothetical restaurant while maintaining participants' satisfaction with their set of choices. Our results demonstrate LLMs' potential, supported by optimization techniques, to accelerate sustainable food development and adoption. Our code is available at https://github.com/thomasat/llms-sustainable-food.

## 1. Introduction

Global food production practices are a leading contributor to climate change, deforestation, water pollution, biodiversity loss, antibiotic resistance, and zoonotic disease transmission (Allan et al., 2023; Hopwood et al., 2020; Scherer et al., 2019). Given the disproportionate harms of animal agriculture in particular, which accounts for 57% of food systems emissions, a number of companies have formed to develop more sustainable protein sources (Xu et al., 2021; Mylan et al., 2023). Additionally, programs such as the Menus of Change have developed to promote sustainable choices in foodservice operations (Menus of Change, 2024). These directions require knowledge of both food science and human preferences.

Though LLMs have shown promise in scientific discovery (Boiko et al., 2023; Guo et al., 2023) and modeling human behavior (Park et al., 2022; Horton, 2023), applications in sustainable food remain underexplored. Beyond its importance in sustainability and public health, this domain represents a compelling testbed for broader machine learning (ML) challenges, including modeling diverse human preferences, generating creative solutions, and aligning outputs with real-world constraints and societal values (Chakraborty et al., 2024; Yang et al., 2024). These challenges are not only central to sustainable food but also to other domains, making sustainable food tasks a valuable benchmark for advancing general ML methods. To initiate an exploration of what LLMs can contribute to sustainable food, we assembled a team consisting of culinary and food science professionals and researchers in ML, natural language processing, and statistics. We make three main contributions:

- Based on collaboration with expert culinary and food science professionals, we define a typology of tasks grounded in the needs of food scientists and chefs. Our typology covers design and prediction tasks across the resolutions of ingredients, recipes, and food systems.

- We evaluate six LLMs on four concrete instantiations of tasks in our typology, incorporating two datasets not previously studied by the machine learning community. Notably, LLMs meet or exceed expert human performance on our sustainable protein experimental design task, and also show promising performance in coarse-grained preference prediction. However, they exhibit an omnivore bias, produce suboptimal solutions when balancing multiple constraints (e.g. maintain satisfaction while reducing emissions), and fail to perform well in fine-grained preference prediction.

- We propose an approach that integrates LLMs with combinatorial optimization techniques to address LLMs' relative weakness in mathematical reasoning, and demonstrate that our approach achieves a 79% av-

[1]Stanford University [2]Umai Works. Correspondence to: Anna T. Thomas <thomasat@stanford.edu>, Kristina Gligorić <gligoric@stanford.edu>.

*Proceedings of the 42nd International Conference on Machine Learning*, Vancouver, Canada. PMLR 267, 2025. Copyright 2025 by the author(s).

erage emissions reduction in food choices while maintaining patron satisfaction with their set of choices.

## 2. Related Work

**Large Language Model Evaluations.** Rigorous evaluation of LLMs has clarified their strengths and weaknesses. Guha et al. (2024) identified six types of legal reasoning tasks and evaluated 20 models on them. In chemistry, Guo et al. (2023) identified three types of capabilities, established a benchmark of eight tasks, and evaluated five LLMs. However, no work, to our knowledge, has formalized key tasks and evaluated LLMs in the sustainable food domain.

**Large Language Models for Scientific Discovery.** In chemistry, Boiko et al. (2023) demonstrated that GPT-4 combined with internet search and code execution can perform experimental design and execution. In genomics, Roohani et al. (2024) showed that an LLM with access to several tools can design genetic perturbation experiments. In computational nanobody design, Swanson et al. (2024) propose a framework in which LLM agents receive periodic human feedback, and showed promising performance in the design of novel antibodies. In artificial intelligence research, Si et al. (2024) show that LLMs can generate novel research ideas. However, we are not aware of prior work on LLMs for discovery of novel sustainable foods.

**Large Language Models for Simulating Human Preferences.** While we are not aware of prior work studying how LLMs can model human preferences and perceptions related to food, a growing literature has shown evidence that LLMs can accurately simulate human behavior in tasks such as search behavior, participation in online communities, hiring scenarios, and classic economic, psycholinguistic, and social psychology experiments (Zhang et al., 2024; Park et al., 2022; Horton, 2023; Aher et al., 2023).

**Large Language Models and Optimization.** Our work also pertains to the use of LLMs for mathematical optimization, and combining LLMs with optimization techniques. Yang et al. (2024) evaluated LLMs for several optimization problems. Ahmaditeshnizi et al. (2024) introduce OptiMUS, an LLM-based agent for solving optimization problems. Motivated by sustainable food applications, we propose a general framework for integrating LLMs with combinatorial optimization techniques that leverages LLMs' knowledge on topics such as human preferences. We further discuss these works and additional related work in Appendix B.

## 3. Task Definitions

### 3.1. Typology

We identify a non-exhaustive typology of sustainable food tasks amenable to ML approaches, shown in Table 1. Based on the sustainable food literature and collaboration with experts, we identify two types of tasks, (1) design and (2) prediction, across three resolutions, (1) ingredients, (2) recipes, and (3) systems. *Design tasks* focus on creating or optimizing food products or systems to meet specific target properties, such as sensory dimensions (taste, texture, etc.), nutrition, cost, and climate impacts. *Prediction tasks* focus on inferring the mapping from a given representation of a food product or system to a property of interest. *Ingredients* are the fundamental components used to create food products, including raw materials (e.g. grains, legumes, oils) and processed components (e.g. protein isolates, flavorings). *Recipes* are structured formulations or instructions for combining ingredients to create food products. *Systems* refer to the broader processes that govern the production, distribution, and consumption of food. This includes supply chains, foodservice operations, and consumer behavior.

We additionally distinguish between novel and traditional foods. Novel sustainable foods, often referred to as "sustainable proteins" or "alternative proteins", are defined as foods developed using modern food science and technology to emulate or improve upon animal products with respect to dimensions such as taste. The three main types of sustainable protein technologies are plant-based, cultivated, and fermentation (Good Food Institute, 2025). Here we focus on plant-based products. Example of novel plant-based products include plant-based meat analogs (e.g. Beyond and Impossible burgers) and plant-based dairy alternatives (e.g. oat milk). While fermentation and cultivated meat are promising approaches for shifting consumption, and we believe our frameworks can be extended to these domains as well, they are outside the scope of the present work. We refer the reader to Waschulin & Specht (2018) and Todhunter et al. (2024) to learn about these two technologies.

### 3.2. Selected Tasks

Starting from this general typology, we then selected four tasks to address in the context of both sustainable protein and traditional foods. These tasks include two design tasks (experimental design and menu design) and two prediction tasks (sensory profile prediction and recipe preference prediction). The four tasks were selected based on interviews with food scientists and chefs.

**Experimental Design.** Taste is a key determinant of food choice (Glanz et al., 1998). Food companies depend on "sensory panels", a group of individuals who provide feedback on food, to routinely quantify sensory properties of their products (Kerth, 2013). Once the sensory panel evaluation has been run, food companies will then synthesize the feedback and revise their product accordingly via a series of experiments (Beckley et al., 2008; Tomsov, 2023). We consider the common task of, given a product formulation

| Task \ Resolution | Ingredients | Recipes | Systems |
|---|---|---|---|
| **Design** | 1. Design breeding strategies for chickpeas to improve yield
2. Design a plant-based protein blend to mimic egg white texture | 1. Design a lentil soup recipe meeting nutritional and cost constraints
2. Design experiments to improve a plant-based product formulation in response to feedback | 1. Design a menu meeting a food-service operation's constraints
2. Design an optimized supply chain for a plant-based meat company to reduce climate impacts |
| **Prediction** | 1. Predict complementary ingredient pairings for traditional plant proteins
2. Predict functional properties (e.g., solubility) of a proposed plant-based analog for casein | 1. Predict which of two recipes a target population will prefer
2. Predict sensory profile of a plant-based product formulation | 1. Predict change in patron satisfaction when making plant-based the default
2. Predict climate and economic impacts of a regional transition to plant-based meat |

*Table 1.* **A typology of tasks at the intersection of ML and sustainable food development.** Tasks are categorized into design and prediction across the resolutions of ingredients, recipes, and food systems. For each task, we list examples corresponding to both traditional (1) and novel (2) foods. Tasks we study are highlighted.

and sensory panel feedback, designing a set of experiments to address the identified issues and improve the product.

**Menu Design.** Foodservice operations, such as restaurants and dining halls, aiming to reduce their climate impacts must also balance other factors, such as patron satisfaction, nutrition, and cost. We thus consider the task of revising a menu to promote more sustainable choices while maintaining patron satisfaction and other relevant factors (Banerjee et al., 2023; Parkin & Attwood, 2022).

**Sensory Profile Prediction.** Sensory panel evaluations are very expensive and time consuming to run, and smaller food companies usually cannot afford to run them (Varela & Ares, 2012). Due to these challenges, we consider the task of estimating a sustainable protein product's sensory profile. While we do not expect to eliminate the need for sensory panels altogether, accurate sensory modeling could reduce the need for experiments and help to efficiently prioritize them. We note that prior work has shown that ML can accurately model the human sense of smell (Lee et al., 2023), which is closely related to taste (Spence, 2015).

**Recipe Preference Prediction.** Similarly, in the domain of traditional foods, chefs must anticipate patron preferences and design their recipes and menus accordingly. We thus consider the task of predicting the mapping from a traditional recipe to average consumer satisfaction within a specified population. We study how accuracy on this task compares for plant-based versus animal-based recipes. While food preferences of course vary greatly from individual to individual, in both of our prediction tasks the goal is to simulate a representative sample of humans from a specified population (Aher et al., 2023).

## 4. Datasets

**NECTAR Sustainable Protein Dataset.** The NECTAR Initiative's (`nectar.org`) sensory panel data[1] is freely available to academic researchers. The dataset, which will continue to expand in size, consists of 47 products across five categories. One product in each category is a meat-based reference product, and one product is a hybrid beef and mushroom burger, yielding 41 plant-based products. For each product, at least 100 sensory evaluations from American omnivores were performed along 21 quantitative and qualitative dimensions. More details are in Appendix C.1.

**Food.com Recipe Dataset.** The `Food.com` dataset[2] contains 522,517 recipes, including ingredients and preparation instructions. We use the associated 1,401,982 reviews, containing ratings and text, to capture online users' preferences. Given the large number of recipes, the dataset allows for finding pairs of comparable recipes, i.e., similar in dish type and ingredients, but significantly differing in the users's ratings. An example recipe is shown in Appendix C.2.

## 5. Experimental Evaluation

We evaluate Claude 3.5 Sonnet, Gemini 1.5 Pro, GPT-3.5 Turbo, GPT-4o, Llama 3.1 70b Instruct, and o1-preview. In the experimental design task, due to limited availability of expert food scientists, we used o1-preview only, based on feedback on relative LLM performance from our food scientist team member. We use a combination of automated evaluations and human subjects evaluations, including both food science experts and participants recruited on Prolific. IRB approval was obtained. All evaluations were in a zero-shot setting. Additional details for each task, including prompts and additional results, are in Appendices D and E.

---

[1]Access can be requested here.
[2]Publicly available here.

## 5.1. Experimental Design (Sustainable Protein)

**Methods.** In this task we study whether LLMs can generate an experimental design for systematically improving a sustainable protein product on the basis of qualitative and quantitative sensory panel data. We evaluate the performance of o1-preview via the feedback of 20 expert food scientists, with an average of five years of experience in plant-based food science and three years of experience in plant-based meat specifically. In Phase 1, we ask both o1-preview and expert food scientists to generate experimental designs. In Phase 2, we ask food scientists to evaluate both o1-preview and a fellow food scientist (blinded, and in randomized order) along the dimensions of accuracy, specificity, complementarity (to the evaluating food scientist's own thought process), and estimated time saved by collaborating with the anonymous "scientist." We ensured that no food scientists evaluated their own ideas. The food scientists were compensated via $50 Amazon gift cards, and an additional $50 bonus was provided to the highest scoring food scientist from Phase 1 to incentivize effort. Before Phase 2, following Si et al. (2024) we perform style standardization (also with o1-preview) on both the LLM and human responses to avoid confounding by style. In order to approximately match the length of the human responses, o1-preview was instructed to limit its response to 250 words.

**Results.** Our results are shown in Figure 1. Across the 30 products, the mean performance of LLMs was higher on all four dimensions, though only statistically significant (in a $t$-test) for specificity ($p$=0.003) and percent time saved ($p$=0.0002). Mean estimated percent time saved was 22% for the human food scientists and 45% for o1-preview.

## 5.2. Menu Design (Traditional Foods)

**Methods.** Here we study what LLMs can contribute to greater adoption of existing sustainable foods. Extensive past work has studied how to shift consumption toward sustainable options in foodservice operations (Lohmann et al., 2024; Attwood et al., 2020; Banerjee et al., 2023; Weijers et al., 2024). However, these studies either obtained relatively small improvements in climate impact or did not measure patron satisfaction with their set of choices. Additionally, none accounted for the trade-off known as the "small body problem", in which simply substituting beef by chicken or fish, a common approach for reducing climate impacts, can be much worse for animal welfare due to the relative sizes of these animals and their typical conditions (Mathur, 2022). Public opinion polling suggests high levels of concern for animal welfare (Gallup, 2015). Given that LLMs' training data includes many recipes and menus, we explore what LLMs can contribute to designing appealing menus under the constraints of ingredient availability, greenhouse gas (GHG) emissions, and animal welfare. Our

final approach, described in Section 6, can also easily be extended to other constraints such as nutrition, cost, allergies, other dimensions of climate impact, etc.

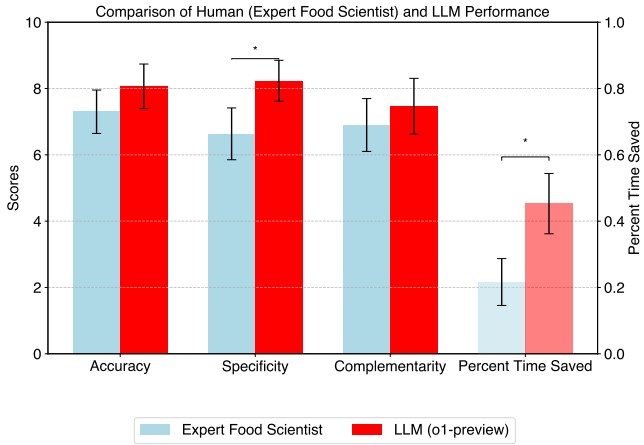

*Figure 1.* o1-preview meets or exceeds the performance of expert food scientists on a sustainable protein design task. o1-preview and food scientists were asked to generate experimental designs for improving product formulations in response to sensory panel feedback, a common task for food scientists. Error bars are 95% confidence intervals. Example outputs are in Appendix E.1.

As a reference menu, we use a 36-item menu from Banerjee et al. (2023), created using Deliveroo's and Just Eat's top 100 items ordered in the United Kingdom in 2019. Each item consists of a title and an appealing description, e.g. "Chicken Curry Ramen. Japanese fried chicken and noodles in a delicious curry broth." Each LLM was instructed to reduce emissions of food choices by 75% while maintaining satisfaction, cost, nutrition, preparation time, and animal welfare, and using only the same ingredients as the original menu other than standard vegetarian protein sources (most of which were already in the original menu, other than eggs and chickpeas). Adherence to the ingredient availability constraint, as well as meeting the constraint of generating 36 properly formatted recipes, was checked, and the LLM was given up to five chances to correct itself. GPT-3.5 Turbo and Llama 3.1 70b Instruct did not meet the formatting or ingredient constraints.

**Results.** We found that this approach yielded entirely vegetarian or vegan menus (shown in Appendix F), known to generally decrease satisfaction in a typical omnivore population (Hartmann & Siegrist, 2017). This decrease in satisfaction was further supported in our later experiments (Figure 2), motivating our approach described in Section 6.

| | Claude 3.5 Sonnet | Gemini 1.5 Pro | GPT-3.5 Turbo | GPT-4o | Llama | o1-preview | Baseline |
|---|---|---|---|---|---|---|---|
| **Sensory Profile Prediction: LLM Prediction vs. Sensory Panel (Sustainable Protein)** | | | | | | | |
| All Dimensions | **0.61** | 0.53 | 0.52 | **0.59** | **0.57** | **0.64** | **0.67** |
| Overall Satisfaction | 0.60 | 0.41 | 0.38 | 0.39 | 0.51 | 0.64 | **0.73** |
| Meatiness | 0.61 | 0.57 | 0.51 | 0.62 | 0.52 | 0.66 | 0.48 |
| Greasiness | 0.58 | **0.64** | 0.63 | **0.74** | **0.73** | **0.64** | **0.69** |
| Juiciness | 0.62 | 0.60 | 0.63 | 0.63 | 0.51 | 0.64 | **0.71** |
| Sweetness | 0.36 | 0.61 | 0.61 | 0.61 | 0.50 | 0.64 | 0.75 |
| Saltiness | **0.70** | 0.67 | 0.57 | **0.73** | **0.70** | **0.75** | **0.73** |
| Purchase | 0.65 | **0.29** | 0.37 | 0.45 | 0.43 | 0.54 | 0.61 |
| **Recipe Preference Prediction (`Food.com`): LLM Prediction vs. Rating (Traditional Foods)** | | | | | | | |
| Rating | **0.63** | 0.48 | 0.57 | **0.64** | 0.51 | **0.60** | 0.50 |
| **Recipe Preference Prediction (Menu Design): LLM Prediction vs. Order Frequency (Traditional Foods)** | | | | | | | |
| Order Frequency | **0.73** | **0.62** | - | **0.67** | **0.60** | **0.56** | 0.50 |

*Table 2.* We evaluate LLMs' pairwise ranking accuracy for three forms of preference prediction: 1) the mapping from a sustainable protein product's ingredients and nutritional information to sensory properties 2) the mapping from a recipe to its online rating 3) the mapping from a recipe to frequency of orders in a hypothetical restaurant. Sample sizes are in Appendix Table 9. Statistically significant results, based on a chi-squared test, are in bold. For the sustainable protein task, the baseline is described in Section 5.3. For recipe preference prediction, the baseline corresponds to a random choice. GPT-3.5 Turbo did not produce a valid output in the menu design experiment.

## 5.3. Sensory Profile Prediction (Sustainable Protein)

**Methods.** Here we study whether LLMs can make accurate predictions about the mapping from a sustainable protein product's ingredients and nutritional information to its sensory properties. Of the 21 sensory dimensions in the NECTAR dataset, seven dimensions (overall satisfaction, meatiness, greasiness, sweetness, juiciness, saltiness, purchase intent) were selected for testing, to reduce the number of tests. Within each of the five product categories, and each sensory dimension, we ask LLMs to compare pairs (with randomized ordering, to account for positional biases) of plant-based products with a statistically significant ($p < 0.05$ in a $t$-test) difference on the sensory dimension, as evaluated by human omnivores. This yielded 495 pairs. For the baseline, we use the available nutritional information and prior knowledge from the food science literature. Specifically, for greasiness and juiciness, meatiness, sweetness, and saltiness, the ratio of total fat, protein, sugar, and sodium respectively per serving to serving size are used to predict the sensory dimension. For example, the product that is higher on protein is ranked higher on meatiness according to the baseline. For overall satisfaction and purchase intent, the average of normalized fat and sodium content is used (Drewnowski & Greenwood, 1983; Mattes, 1997).

**Results.** Accuracies are shown in Table 2. Accuracies significant according to a chi-squared test with Bonferroni correction for multiple testing are bolded (VanderWeele & Mathur, 2019). Performance is generally not meaningfully superior to the baseline. Accuracy for the best (across all dimensions) LLM, o1-preview, improves in the highest quartile of the ground truth difference between the recipes, e.g.

to 81% and 75% for overall satisfaction and purchase intent. However, it is still not meaningfully superior to the simple baseline, which achieved 86% and 70% accuracy on those dimensions in this subset of pairs (Appendix Table 18). An expert human baseline for the "Overall Satisfaction" dimension was collected from a collaborator with deep domain expertise (11 years of food science experience, including co-founding two plant-based meat companies), who is also a co-author on this paper. The results are shown in Appendix Table 14.

## 5.4. Recipe Preference Prediction (Traditional Foods)

**Methods.** We created a set of 500 recipe pairs with similar ingredients but significantly different average ratings ($p < 0.05$ in a $t$-test), ensuring that one recipe is consistently preferred by users. Examples pairs are shown in Appendix C.2. To identify recipe pairs with the highest similarity, the overlap of ingredients was calculated for each pair using the ratio of shared ingredients to the total number of unique ingredients. We addressed the possibility of overlap with LLM training data by curating an additional set of 58 recipe pairs from `epicurious.com`. These recipes were posted behind a paywall after the release dates of all the models we tested. Our primary metric was accuracy—whether the models correctly identified the preferred recipe—compared to a random baseline. We tested statistical significance using a chi-squared test with Bonferroni correction when evaluating on multiple conditions. The order in which the recipes were shown was randomized.

**Results.** Claude 3.5 Sonnet, GPT-4o, and o1-preview outperform the random baseline (Table 2). The

highest accuracy achieved is 64% (GPT-4o), with accuracy improving to 85% in the highest quartile of the ground truth review score gap, i.e., pairs of recipes for which people's preferences were most different (Appendix Table 19). Moreover, performance varies widely by comparison type (Appendix Table 20). Across all the models, accuracy is the highest when two non-vegetarian meals are compared, followed by vegetarian vs. non-vegetarian comparisons, with the worst accuracy observed in vegetarian vs. vegetarian meal comparisons. This aligns with existing literature showing that minority preferences and opinions are modeled less accurately (Santurkar et al., 2023). Within the vegetarian vs. non-vegetarian meal comparisons, accuracy is lower when vegetarian options are rated higher, reflecting LLMs' omnivore bias, as LLMs more frequently prefer non-vegetarian options compared to ground truth human preferences. This finding is consistent with the literature on bias amplification in the food domain (Luo et al., 2024) and points to the need to improve preference prediction performance in order to develop tools that can better support the development of sustainable food. Finally, similar pairwise accuracy across models on post-cut-off data suggests that performance cannot be fully attributed to memorization (Appendix Table 21).

# 6. Augmenting LLMs with Combinatorial Optimization

When prompted directly to revise a menu to reduce emissions by at least 75% while maintaining patron satisfaction and other factors, LLMs generate entirely plant-based menus (Section 5.2, Appendix F). While this approach achieves the emissions target, it significantly decreases satisfaction (Figure 2). This failure highlights a broader, previously documented challenge: LLMs struggle with optimization problems (Ahmaditeshnizi et al., 2024; Yang et al., 2024). However, frequently in real-world applications we need to consider some subjective, human-evaluated property (like satisfaction) while also satisfying other constraints (like emissions targets or budgets). For instance, in education, we may want to maximize student engagement while meeting curriculum requirements. In fitness, we may aim to maximize enjoyment while covering specified muscle groups within time constraints. In travel planning, we may seek to maximize trip enjoyment while meeting budget and scheduling constraints. Additionally, in some settings it may take considerable effort to quantify dimensions such as emissions or other climate impacts of a food item.

We propose to address these challenges by combining LLMs' background knowledge and ability to model human preferences, demonstrated in Horton (2023), Park et al. (2022) and in our own experiments, particularly for coarse-grained prediction (Tables 2,11,18,19), with mathematical optimization. While LLMs alone struggle with optimization, and traditional optimization methods do not have access to LLMs' background knowledge, combining them allows for leveraging the strengths of both approaches.

## 6.1. LLM-Guided Combinatorial Optimization

We begin with the general constrained optimization setting:

$$\max_{x} \quad f(x)$$
$$\text{s.t.} \quad g_i(x) \leq 0 \,\forall i \in [1, ..., m] \tag{1}$$

We make three assumptions:

1. The optimization problem involves selecting a subset of items $S$ from a ground set $U = \{u_1, ..., u_N\}$. Each $u_i$ represents an item, e.g. a recipe or exercise. The selection is represented by a binary vector $x \in \{0, 1\}^N$, where $x_i = 1$ if $u_i$ is selected, and 0 otherwise.

2. At least one of $f(x)$ or the constraints $g_i(x)$ can be expressed in terms of a *scoring function* $p(u_i)$, which computes a scalar property (e.g. satisfaction) for each item $u_i$. The scoring function can be estimated by an LLM. For example, if the objective is to maximize the total satisfaction of the end user with the selected items, $f(x)$ can be written as $f(x) = \sum_{i=1}^{N} p(u_i)x_i$.

3. Once $p(u_i)$ is estimated for each $u_i$ in $U$, (1) has a tractable form and can be solved exactly or approximately via standard techniques, e.g. submodular optimization or integer programming for small to medium problem sizes (Fujishige, 2005; Wolsey, 2020). We give a concrete example in Section 6.2.

We can then solve (1) via the following steps:

1. Generate the ground set $U = \{u_1, ..., u_N\}$, e.g. a diverse set of recipes or exercises.

2. Obtain the estimates $\hat{p}(u_i) \,\forall i \in [1, ..., N]$ via an LLM.

3. Solve the combinatorial optimization problem using standard techniques, e.g. submodular optimization or integer programming, depending on the forms of $f(x)$ and $g_i(x)$. This will yield a subset $S \subset U$.

Step 1 could be performed via an LLM or via other techniques. We discuss the relationship of this framework to prior work in Appendix B. A natural question is how performance depends on the accuracy of the LLM's estimates $\hat{p}(u_i)$, and other parameters. For our analysis we consider the cardinality constrained setting where $f(x)$ has a component that is linear in $p(u_i)$ for the selected subset, and another component $d(x)$ that does not depend on $p(u_i)$. Concretely, we aim to maximize $f(x) = \sum_{i=1}^{N} p(u_i)x_i + \lambda d(x)$ subject to the constraint $\sum_{i=1}^{N} x_i = K$, though this result can be extended to more general objectives and constraints.

**Proposition 6.1.** *Let $x^*$ be the optimal solution with respect to actual preferences $p(u_i)$, and $\hat{x}^*$ be the optimal solution with respect to estimated preferences $\hat{p}(u_i)$. Let $p(u_i), \hat{p}(u_i) \in [0,1]$, $d(x) \in [0,D]$. If $|\hat{p}(u_i) - p(u_i)| \leq \epsilon$ for all $u_i \in U$, then $|f(\hat{x}^*) - f(x^*)| \leq 2K\epsilon$.*

A simple proof is in Appendix A. Thus, the error can be bounded as a linear function of the number of items selected and the maximum item-level error of the LLM.

### 6.2. Constrained Menu Design Formulation

Now we show how to implement this framework for the setting of constrained menu design. First, we prompt the LLM to generate a set $C$ of plant-based candidate recipes, though this could be modified to allow for also generating animal-based recipes at this stage. As in Section 5.2, the LLM is also instructed to not worsen cost, nutrition, or preparation time. We then combine this set with the recipes in the original menu $O$, where $K = |O|$, to form the ground set $U = C \cup O$. We then estimate $p(u_i) \, \forall u_i \in U$ by prompting an LLM to rate each recipe title in the ground set by expected preferences of the population of interest (here, American omnivores). Then, integer quadratic programming (IQP) is applied to select a subset of recipes from $U$ for the menu.

More specifically, we maximize expected satisfaction with the selected recipes, plus a $\lambda d(x)$ term, where $\lambda$ controls the weight on diversity, to encourage a diverse set of recipes, subject to constraints on climate impacts and animal welfare. This approach easily extends to other constraints such as nutrition and cost. The predicted preferences $\hat{p}(u_i)$ are used to compute expected emissions and animal welfare impacts of choices by weighting more preferred options more highly, under the assumption that these options will be chosen more often. Our problem formulation is thus as follows:

$$
\begin{aligned}
\max_x \quad & \sum_{i=1}^{N} \hat{p}(u_i) x_i + \lambda d(x) \\
\text{s.t.} \quad & E_{\hat{p}}[l_j(x)] \leq C_j E_{\hat{p}}[l_j(x_O)] \; \forall j \in [1,2] \\
& \sum_{i=1}^{N} x_i = K \\
& x_i \in \{0,1\} \; \forall i \in [1,...,N]
\end{aligned}
\tag{2}
$$

$x_O$ corresponds to the subset of recipes in the original menu. $E_{\hat{p}}[l_j(x)]$ computes the expected emissions or animal welfare impacts of the chosen subset of recipes, with respect to the probability distribution associated with $\hat{p}(u_i)$.

We set $d(x)$ to be negative pairwise similarity between the recipes in the selected subset. This yields an IQP since all other terms are linear in $x$. $C_i$ controls the reduction in emissions or animal welfare, and is set to 0.25 for emissions

and 1.0 for animal welfare, corresponding to reducing emissions by at least 75% while maintaining or improving upon animal welfare. We set $K = 36$, to match the length of the original menu from Banerjee et al. (2023). We prompt the LLM to generate 20 additional recipes. Thus, $N = 56$. Finally, we set $\lambda = 100$. For computing pairwise recipe similarity we use the Ratcliff/Obershelp sequence matching algorithm as implemented in Python's difflib (Ratcliff et al., 1988). The recipes were ordered first by whether they were LLM generated, and second by predicted rating. We included a baseline in which the original menu was simply re-ordered so that vegetarian items were first, to account for the possibility that benefits of the LLM+IQP approach were simply due to placing the LLM generated (and therefore plant-based) recipes first.

We then evaluated our approach via two human subjects experiments (total $n = 800$) in which participants recruited via Prolific, constrained to be in the United States and fluent English speakers, were instructed to make a selection from a menu. In the first experiment, the menu was randomly assigned from one of the five LLMs (augmented with IQP) and three baselines (original menu, original menu with beef removed, and a vegetarian subset of the original menu).[3] In the second experiment, conducted the following day, the menu was randomly assigned from one of three additional baselines (o1-preview on its own, without IQP; o1-preview+IQP, but with the o1-preview generated descriptions replaced with a simple list of ingredients; and placing the vegetarian items first in the original menu). The participants were paid \$16 per hour. The evaluation metrics consisted of satisfaction metrics (satisfaction with their set of choices, likelihood of visiting, and likelihood of recommending the restaurant to a friend), emissions of food choices, and animal welfare impacts of food choices.

### 6.3. Results

The results for the best LLM, o1-preview, are shown in Figure 2, with results for other LLMs and baselines in Appendix E.2. When controlling for dietary preference, age, gender, and race, we estimate that the o1-preview+IQP approach reduces emissions by 79% while maintaining patron satisfaction with their set of choices ($p$=1.37e-18). We obtain similar results when removing vegan and vegetarian participants (Appendix E.2). We observe a small (5%) reduction in the likelihood of recommending the restaurant to a friend, and a small increase in animal usage (3.5%), but these were not statistically significant. The original, baseline, and generated menus are shown in Appendix F. We also evaluated the accuracy of the LLM's predicted preference ratings, in the same pairwise ranking fashion as the

---

[3]Removal of beef is a common strategy for reducing climate impacts (Grummon et al., 2023).

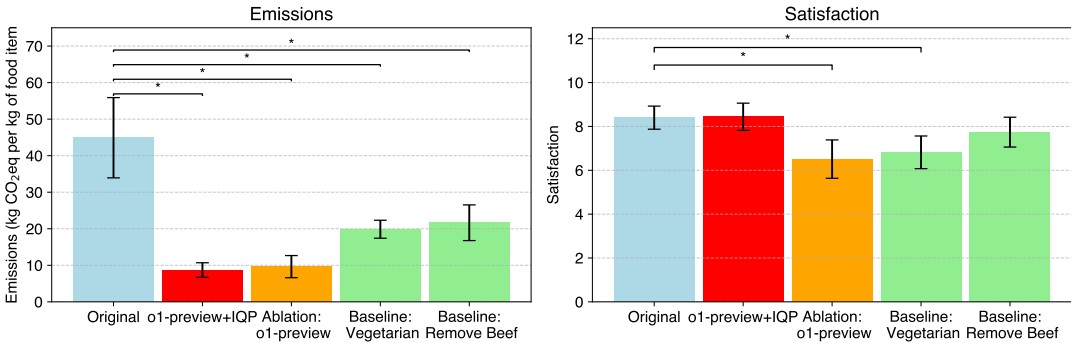

*Figure 2.* Our o1-preview+IQP approach reduces GHG emissions of food choices by 79% compared to the original menu from Banerjee et al. (2023) while maintaining participants' satisfaction with their set of choices. o1-preview+IQP additionally outperforms the baselines of removing meat from the original menu ("Vegetarian") and removing beef ("Remove Beef"), and maintains animal welfare and other satisfaction metrics, as shown in Appendix E.2. As an ablation we also prompt o1-preview directly to revise the menu. Error bars are 95% confidence intervals. Statistical significance was assessed with a $t$-test, with Bonferroni correction for multiple hypothesis testing.

sensory profile and recipe preference prediction tasks, with the best LLM, Claude 3.5 Sonnet, achieving 73% accuracy (Table 2). This increases to 85% in the highest quartile of the preference gap (Appendix Table 11). An expert human baseline was collected from a collaborator with deep domain expertise (32 years of culinary experience), who is also a co-author on this paper. The results are shown in Appendix Table 13.

## 7. Discussion

Our results identify both strengths and weaknesses of LLMs in the domain of sustainable food. One actionable finding is that, as determined by expert food scientists, o1-preview appears to be useful, even superior to fellow expert human food scientists, in generating ideas for revision of sustainable protein products in response to sensory panel feedback. Our evaluation suggests that the benefits may be driven by increased specificity in particular, relative to the human food scientists (Figure 1). LLMs display a weakness in the menu design task, where they are instructed to satisfy multiple constraints. They generate fully plant-based menus, which we find to significantly reduce satisfaction. We remedy this by integrating LLMs with combinatorial optimization techniques. Our approach achieves a 79% emissions reduction while maintaining patron satisfaction (Figure 2). Our work also contributes to the literature on reasoning in LLM-based systems (further discussed in Appendix B), providing an approach for solving a broad class of optimization problems that cannot be easily specified mathematically, e.g. those involving human preferences. Future work can apply this framework to other domains such as education and health.

We evaluated LLMs on three types of preference prediction. The first is sensory profile prediction, in which we compared LLM predictions to evaluations from an actual taste tasting.

The second is the `Food.com` recipe rating task, where we compared LLM predictions to online recipe ratings. The third is recipe rating in the context of the menu design experiment, where the LLM's recipe-level predictions are compared against the order frequency - the frequency with which people chose the dishes in our human subjects experiment - for the recipes that were selected for the final menu. For the holistic measures of overall satisfaction and purchase intent in the sensory profile prediction task, no LLMs outperform a random baseline, though some do on specific dimensions such as greasiness or saltiness. For online recipe rating, three of six LLMs outperform a random baseline. All evaluated LLMs outperform a random baseline in context of the menu design experiment, where no actual tasting occurs and the orders are purely based on the text of the recipe (Table 2). Additionally, we studied how performance varies with the magnitude of the ground truth preference gap. When restricting the evaluated recipe pairs to be in the top quartile of preference gaps, LLM performance improves, with the best LLMs achieving 81%, 86%, and 85% accuracy respectively in the three prediction tasks (Appendix Tables 11,18,19). Thus, our analysis suggests that LLMs can be useful for coarse-grained preference prediction tasks, but that further work is needed for fine-grained prediction, particularly for the mapping from recipes or product formulations to actual taste. Additionally, consistent with past research on LLMs' modeling of minority preferences (Santurkar et al., 2023), our results showed that LLMs model human preferences related to plant-based food less accurately. Thus, regular human feedback, e.g. taste testings, from diverse populations remains critical. Overall, these results suggest that LLMs can be useful in experimental design, generation of sustainable recipes (as evaluated in the menu design task), and coarse-grained preference prediction.

**Limitations.** We did not do significant prompt engineering,

and leave this as an area for future work. In the sustainable protein tasks, we did not have access to the full product formulation, only ingredients and nutritional information. However, we consider this a positive, that users do not necessarily have to provide sensitive intellectual property to receive useful outputs from LLMs. In our menu design task, we did not provide incentives to make a realistic choice, such as delivering the meals to participants. Additionally, we did not study dimensions of food choice such as food waste and portion size. A stronger baseline for this task would have been a human chef given the same menu and constraints, though we note that our "Vegetarian" baseline is based on the top vegetarian meals on online delivery apps. Future work could establish an expert human (food scientist or chef) baseline for the preference prediction tasks. In the experimental design task, implementing the suggestions and running a second sensory panel to determine whether the suggestions improve the product is an area for future work.

## Impact Statement

Our paper is directly motivated by broader societal impacts, specifically in sustainability and public health. Our task definitions and results could spur progress in both development of novel sustainable foods and greater adoption of existing sustainable foods. The potential benefits of a sustainable protein transition are significant. For example, a recent life cycle assessment found that, relative to a beef burger patty, the production of a plant-based burger patty made with soy protein is associated with 98%, 87%, and 99% lower GHG emissions, land use, and air pollution respectively (Saerens et al., 2021), with similar estimates for other types of sustainable protein (Good Food Institute, 2023). Additionally, the Intergovernmental Panel on Climate Change (IPCC) estimates that a societal transition from animal-based food (ABF) to plant-based food (PBF) by 2050 would reduce land-based GHG emissions by two-thirds relative to the business-as-usual case (Intergovernmental Panel on Climate Change, 2019).

In the menu design task, we consider ethical aspects such as the trade-offs among patron satisfaction, sustainability, and animal welfare, and show that our approach can improve or maintain all dimensions. Designing menus to shift the distribution of patron choices may raise questions of respect for individual autonomy. We thus use three different metrics of patron satisfaction, and show that all are maintained with the revised menu (Appendix Figures 17,18,19).

We also note that current sustainable proteins are sometimes considered "ultra-processed" and therefore unhealthy (The New York Times, 2019). We view this as a challenge for the next generation of sustainable proteins that AI could potentially help with, via jointly optimizing for satisfaction, nutrition, climate impacts, cost, and other relevant factors.

Use of LLMs for developing unhealthy or less sustainable foods is also a concern. We acknowledge that such work is likely ongoing, and we study here how LLMs can also be useful for applications that benefit society.

Lastly, the benefits of LLM-supported development of sustainable food should be interpreted jointly with the environmental impacts of LLMs themselves (such as energy and water use). While pre-training in particular requires significant resources (Van Wynsberghe, 2021), we note that work to reduce the climate impacts of LLMs is ongoing (Patterson et al., 2021; Li et al., 2024). Future research can continue to explore ways to leverage pre-trained models for applications that contribute to positive environmental outcomes.

## Acknowledgments

This work is supported by the Stanford Plant-Based Diet Initiative (PBDI). Anna Thomas is supported by National Institutes of Health Grant R01LM013866. Kristina Gligorić is supported by the Swiss National Science Foundation (Grant P500PT-211127). We thank Caroline Cotto and Max Elder for access to the NECTAR dataset, and Moses Charikar, Ali Teshnizi, Madeleine Udell, and Katie Yoon for helpful discussions. We thank Yifan Mai and the Center for Research on Foundation Models (CRFM) for their support.

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

## A. Proof of Proposition 6.1

The problem we would like to solve is:

$$\max_{x} \quad f(x) := \sum_{i=1}^{N} p(u_i)x_i + \lambda d(x)$$
$$\sum_{i=1}^{N} x_i = K \tag{3}$$
$$x_i \in \{0, 1\} \, \forall i \in [1, ..., N]$$

However, we do not have access to the true scores $p(u_i)$, e.g. actual satisfaction of a specified population with a recipe. Instead, we use an LLM to produce the estimates $\hat{p}(u_i)$, and solve:

$$\max_{x} \quad \hat{f}(x) := \sum_{i=1}^{N} \hat{p}(u_i)x_i + \lambda d(x)$$
$$\sum_{i=1}^{N} x_i = K \tag{4}$$
$$x_i \in \{0, 1\} \, \forall i \in [1, ..., N]$$

Let $x^*$ be the optimal solution of (3), and let $\hat{x}^*$ be the optimal solution of (4).

**Proposition A.1.** *Let $p(u_i), \hat{p}(u_i) \in [0, 1]$, and $d(x) \in [0, D]$. If $|\hat{p}(u_i) - p(u_i)| \leq \epsilon$ for all $u_i \in U$, then $|f(\hat{x}^*) - f(x^*)| \leq 2K\epsilon$.*

*Proof.* First, note that $\forall x, |f(x) - \hat{f}(x)| \leq K\epsilon$. Then, combining this with the optimality of $x^*$ and $\hat{x}^*$, $f(\hat{x}^*) \geq \hat{f}(\hat{x}^*) - K\epsilon \geq \hat{f}(x^*) - K\epsilon \geq f(x^*) - 2K\epsilon$. $\square$

## B. Extended Related Work

**Large Language Models and Reasoning.** Our work broadly pertains to the integration of LLMs with other tools for reasoning tasks. Ahmaditeshnizi et al. (2024) develop an agent called OptiMUS for formulating and solving (mixed integer) linear programming problems from their natural language descriptions. Their goal is to expand access to solvers to individuals who lack the expertise required to formulate and solve these problems. Jiang et al. (2025) propose a five-element formulation for parameterizing optimization problems and fine-tune LLMs for generating five-element formulations and solver code from natural language descriptions of optimization problems. Ye et al. (2024) propose SatLM, which combines LLMs with SAT solvers to solve problems in arithmetic reasoning, logical reasoning, and symbolic reasoning. Pan et al. (2023) propose Logic-LM, which integrates LLMs with symbolic solvers to improve logical reasoning. Relative to these works, our setting is fundamentally different, where the underlying optimization problem is not fully specified due to involving subjective human preferences or other components that are difficult to formalize mathematically. We are specifically motivated by the food domain, in which human satisfaction must be considered (alongside other factors such as nutrition, cost, emissions, etc.), but is difficult to write down mathematically. Similar problems arise in settings such as education (e.g. curriculum design) and health (e.g. fitness regimen design). Our framework is also applicable to settings where it may take considerable effort to formally specify the optimization problem, e.g. computing climate impacts as well as cost, nutrition, etc. for each food item, but an LLM may be able to generate sufficiently accurate estimates. We also note past work in the natural language processing literature on combining integer programming with machine learning, e.g. Denis & Baldridge (2007), which used integer linear programming for coference resolution.

**Large Language Models for Climate, Sustainability, and Food.** Bulian et al. (2024) propose an evaluation framework for LLMs in the domain of climate information. Morio & Manning (2023) create a benchmark dataset for assessing corporate climate policy engagement and evaluate several language models. Huang et al. (2024) introduce the FoodPuzzle dataset and create an LLM-based agent for flavor profile prediction. They define two tasks: molecular food prediction, in which

the goal is to predict food sources based on their molecular composition, and molecular profile completion, in which the goal is to identify the missing molecules needed to complete the molecular profile of a given food item. Within our paper, the most closely related task is sensory profile prediction. Our work differs in that we evaluate LLMs on their ability to predict dimensions otherwise evaluated by a human sensory panel, such as overall satisfaction, purchase intent, meatiness, etc. prevalent in the current practice of product development in food science. Sensory panels are essential but also very expensive and time consuming to both run and analyze. Additionally, our evaluation specifically focuses on sustainable protein rather than general food science. Finally, Huang et al. (2024) develop an agent based on in-context learning and retrieval augmented generation, whereas we evaluate LLMs on our sustainable protein tasks via zero-shot prompting, as well as integration with combinatorial optimization.

**Data-Driven Optimization** Our work also pertains to the literature on data-driven optimization. Previous work has proposed the "predict-then-optimize" framework in which a predictive model is used to define the parameters of an optimization problem (Elmachtoub & Grigas, 2022; Bertsimas & Kallus, 2020). We extend the predict-then-optimize framework in the setting of combinatorial optimization by incorporating a component where the elements of the ground set (e.g. recipes) are generated, and applying LLMs to both the generation and prediction steps.

**Artifical Intelligence for Climate, Sustainability, and Food.** Previous work has studied climate- and environment-related applications of artificial intelligence (Tuia et al., 2022; Chapman et al., 2024; Madadkhani et al., 2024; Kaack et al., 2022). Within the food domain, previous work has explored how AI can reduce energy use in plant factories to support sustainable food production (Decardi-Nelson & You, 2024). St. Pierre et al. (2024) demonstrated that a combination of mechanical testing and ML can describe food texture in a similar manner to human taste testers.

## C. Supplementary Information on Datasets

### C.1. NECTAR Dataset

The NECTAR dataset was collected over the period of June to August 2023 in Precision Research's Chicago research center. It is only available to academic researchers to reduce harm to the reputations of companies whose products performed poorly in the taste testing. Additional information can be found at `https://www.nectar.org/taste`. It consists of 5,516 sensory evaluations of 47 products, and will continue to grow in size over the coming years. Each sensory evaluation consists of 21 dimensions, shown in Table 3. The products were prepared in a test kitchen according to manufacturers' instructions. Plant-based product selection criteria were based on popularity, availability (i.e. distributed in-market at the time of the test), and similarity to analog animal offerings (i.e. veggie burgers made from whole plants were not included, whereas plant-based burgers aiming to mimic the eating experience of animal-based burgers were included). All tasting was blind and monadic (one product at a time), and the panel was untrained.

### C.2. `Food.com` Recipe Dataset

Figure C.2 shows an example recipe from the `Food.com` dataset.

We also show some example pairs from this dataset.

## D. Supplementary Methods

### D.1. Experimental Design

The 30 sustainable protein products used in this task consisted of six products from each of the five categories. Further information on the food scientists is in Table 6. The prompt for Phase 1 is shown in Figure 7, and our standardization prompt is shown in Figure 8. Precise definitions of the dimensions of evaluation in Phase 2 are in Section D.1.2. In Phase 1, 15 food scientists participated, each evaluating 2 products. In Phase 2, again 15 food scientists participated, each evaluating 2 products (with one human and one LLM response per product), with an overlap of 10 food scientists between Phase 1 and Phase 2.

| Sensory Dimension | Question Type | Burger | Hot Dog | Bacon | Chicken Nuggets | Chicken Tenders |
|---|---|---|---|---|---|---|
| Appearance | 7 pt Likert Scale | X | X | X | X | X |
| Color | 7 pt Semantic Differential | X | X | X | X | X |
| Flavor Liking | 7 pt Likert Scale | X | X | X | X | X |
| Spiciness | 7 pt Semantic Differential | X | X | X | X | X |
| Like | Open Ended | X | X | X | X | X |
| Dislike | Open Ended | X | X | X | X | X |
| Overall Liking | 7 pt Likert Scale | X | X | X | X | X |
| Meatiness | 7 pt Semantic Differential | X | X | X | X | X |
| Greasiness | 7 pt Likert Scale | X | X | X | X | X |
| Juiciness | 7 pt Semantic differential | X | X | X | X | X |
| Smokiness | 7 pt Semantic Differential | | X | X | | |
| Sweetness | 7 pt Semantic Differential | | X | X | | |
| Saltiness | 7 pt Semantic Differential | | X | X | X | X |
| Crispiness | 7 pt Semantic Differential | | | X | X | X |
| Aftertaste Strength | 5 pt Likert Scale | X | X | X | X | X |
| Aftertaste | 5 pt Semantic Differential | X | X | X | X | X |
| Purchase Intent | 5 pt Likert Scale | X | X | X | X | X |
| Texture: Liking | 7 pt Likert Scale | X | X | X | X | X |
| Chewiness | 7 pt Semantic Differential | X | X | X | X | X |
| Firmness | 7 pt Semantic Differential | X | X | X | X | X |
| Breading Flavor | 7 pt Semantic Differential | | | | X | X |

*Table 3.* Overview of sensory dimensions in NECTAR dataset, by category of sustainable protein product.

| Task | Data | Techniques Used | Evaluation Metrics |
|---|---|---|---|
| **Sustainable protein: experimental design** | NECTAR | Zero-shot prompting | Accuracy, Specificity, Complementarity, Time Saved (Expert Human Food Scientist Evaluation) |
| **Traditional foods: menu design** | Menu from Banerjee et al. (2023), emissions data from Poore & Nemecek (2018), animal welfare data from Faunalytics (2020) | Zero-shot prompting, integer quadratic programming | Satisfaction (Human Subjects Evaluation), Emissions, Animal Welfare (Automated) |
| **Sustainable protein: sensory profile prediction** | NECTAR | Zero-shot prompting | Accuracy (Automated) |
| **Traditional foods: recipe preference prediction** | Food.com | Zero-shot prompting | Accuracy (Automated) |

*Table 4.* Overview of tasks, techniques used, and evaluation metrics.

| | Original | Edited |
|---|---|---|
| o1-preview | 319.8 (37.7) | 327.6 (87.2) |
| Food Scientist | 311.3 (223.9) | 311.6 (195.3) |

*Table 5.* Original and edited lengths of o1-preview's and food scientists' experimental designs. Standard deviation is in parentheses.

### D.1.1. PROPERTIES OF EXPERT FOOD SCIENTISTS

20 food science experts were recruited via direct outreach on LinkedIn or email from our food scientist team member. Their properties are shown in Table 6.

---

**Example Recipe**

Name: Carina's Tofu-Vegetable Kebabs
Description: This dish is best prepared a day in advance to allow the ingredients to soak in the marinade overnight.
Ingredients: extra firm tofu, eggplant, zucchini, mushrooms, soy sauce, low sodium soy sauce, olive oil, maple syrup, honey, red wine vinegar, lemon juice, garlic cloves, mustard powder, black pepper.
Instructions:
1. Drain the tofu, carefully squeezing out excess water, and pat dry with paper towels.
2. Cut tofu into one-inch squares.
3. Set aside. Cut eggplant lengthwise in half, then cut each half into approximately three strips.
4. Cut strips crosswise into one-inch cubes. 5. Slice zucchini into half-inch thick slices.
6. Cut red pepper in half, removing stem and seeds, and cut each half into one-inch squares.
7. Wipe mushrooms clean with a moist paper towel and remove stems.
8. Thread tofu and vegetables on to barbecue skewers in alternating color combinations: For example, first a piece of eggplant, then a slice of tofu, then zucchini, then red pepper, baby corn and mushrooms.
9. Continue in this way until all skewers are full.
10. Make the marinade by putting all ingredients in a blender, and blend on high speed for about one minute until mixed.
11. Alternatively, put all ingredients in a glass jar, cover tightly with the lid and shake well until mixed.
12. Lay the kebabs in a long, shallow baking pan or on a non-metal tray, making sure they lie flat. Evenly pour the marinade over the kebabs, turning them once so that the tofu and vegetables are coated.
13. Refrigerate the kebabs for three to eight hours, occasionally spooning the marinade over them.
14. Broil or grill the kebabs at 450 F for 15-20 minutes, or on the grill, until the vegetables are browned.
15. Suggestions This meal can be served over cooked, brown rice. Amounts can easily be doubled to make four servings.

*Figure 3.* Example recipe from the `Food.com` dataset.

---

**Example Pair 1: Recipe 1**

Corned Beef Dinner - Crock Pot
Ingredients: corned beef brisket, onion, celery, potatoes, carrots, bay leaf, garlic clove, Worcestershire sauce, dry mustard, cabbage, caraway seed.
Instructions: Trim brisket of all visible fat cut to fit 4 qt or larger crockpot if necessary. Place onion celery potatoes and carrots in bottom of crockpot lay brisket on top. Whisk together bouillon bay leaf garlic Worcestershire sauce and dry mustard. Pour over brisket cover pot. Cook on low setting for 8 to 10 hours adding cabbage wedges and caraway seed for the last hour of cooking. To serve discard cooking liquid slice meat onto hot serving plates accompany with the cooked potatoes carrots celery cabbage wedges and your favourite mustard.

---

**Example Pair 1: Recipe 2**

Crock Pot Corned Beef and Cabbage
Ingredients: corned beef brisket, onions, cabbage, pepper, vinegar, sugar, water.
Instructions: Combine ingredients in crock pot with cabbage on top. Cut meat to fit if necessary. Cover and cook on low 10-12 hours; high 6-7 hours or auto 6-8 hours.

*Figure 4.* Example pair 1 from the `Food.com` dataset.

### D.1.2. EVALUATION DIMENSIONS

The dimensions of evaluation were defined to the food scientists as follows:

Example Pair 2: Recipe 1

Oven-Fried Chicken Chimichangas
Ingredients: salsa, ground cumin, dried oregano leaves, cheddar cheese, green onions, flour tortillas, margarine, cheddar cheese, green onion.
Instructions: Mix chicken picante sauce or salsa cumin oregano cheese and onions. Place about 1/4 cup of the chicken mixture in the center of each tortilla. Fold opposite sides over filling. Roll up from bottom and place seam-side down on a baking sheet. Brush with melted margarine. Bake at 400°F for 25 minutes or until golden. Garnish with additional cheese and green onion and serve salsa on the side.

Example Pair 2: Recipe 2

Chicken Rice Casserole
Ingredients: butter, flour, dried thyme, chicken broth, milk, salt, black pepper, cooked rice, parmesan cheese.
Instructions: Melt 3 tablespoons of butter in a saucepan over medium heat whisk in flour and thyme and cook for 1 minute. Gradually stir in broth and milk stirring until thick and smooth. Stir in chicken and add salt and pepper; set aside. Grease a 1 1/2 quart shallow baking dish. Spread rice in prepared baking dish sprinkle with the peas and then pour creamed chicken mixture over. Dot with remaining tablespoon of butter and sprinkle with bread crumbs and cheese which you mix together. Bake at 400° until hot and bubbly about 20 to 25 minutes.

*Figure 5.* Example pair 2 from the `Food.com` dataset.

Example Pair 2: Recipe 1

Bow Tie Pasta With Broccoli and Broccoli Sauce
Ingredients: broccoli, water, bow tie pasta, butter, salt.
Instructions: Cut the broccoli buds into small florets leaving as short a stem as possible; set aside; cut the remaining broccoli stems into 1/2" pieces; boil them covered in the 1 1/4 cups water lightly salted for 15 minutes or until very soft. Meanwhile boil the pasta in salted water until al dente; drain reserving 1/2 cup of the cooking water; return the pasta to the pot and stir until all moisture is evaporated. Place the reserved florets into a small pot and steam or boil 5 minutes retaining the bright green color and a bit of crunch. In a food processor puree the cooked broccoli and its cooking water until very smooth adding a bit of the reserved pasta water if needed to make a smooth sauce; add the butter and pulse until melted; add salt to taste. Pour warm sauce over the pasta and stir gently to combine; plate and sprinkle with drained florets.

Example Pair 2: Recipe 2

Lemon Garlic Pasta
Ingredients: garlic, extra virgin olive oil, lemon juice, chicken flavor instant bouillon, pepper, spaghetti, angel hair pasta, parmesan cheese, parsley.
Instructions: In large skillet cook garlic in olive oil until golden. Add lemon juice bouillon and pepper. Cook and stir until bouillon dissolves. In large bowl toss pasta garlic mixture cheese and parsley; serve immediately.

*Figure 6.* Example pair 3 from the `Food.com` dataset.

- Accuracy: Are these suggestions likely to be useful in improving the product, to the best of your knowledge? Are they consistent with the provided feedback?

- Specificity: How detailed and actionable are the suggestions?

- Complementarity: Did this scientist see anything you missed? Would they be complementary as a brainstorming partner?

---

**Prompt**

You are an expert plant-based meat food scientist. You have devised a *<category>* product with the following ingredient list: *<ingredient list>*.
Additionally, it has the following nutritional information: *<nutrition facts>*.
You ran a blind taste test of American omnivores and received the following quantitative feedback on your product: *<quantitative feedback>*.
Additionally, you received the following qualitative feedback about what people liked: *<positive feedback>*.
You also received the following qualitative feedback about what people disliked: *<negative feedback>*.
What changes would you consider making to your product? Could you design a set of experiments on the key areas that need improvement? You will be evaluated on metrics including accuracy and specificity.

---

*Figure 7.* Prompt for Phase 1 of the experimental design task, in which o1-preview and expert food scientists generated experimental designs for improving sustainable protein products on the basis of sensory panel feedback. For the food scientists, the first line was removed. o1-preview was additionally instructed to limit its responses to 250 words, to approximately match the mean length of the food scientists' responses. Even though the mean original human response length was 311 words (Table 5), 250 was chosen in the instruction to o1-preview because we noticed that o1-preview tended to exceed the word limit.

---

**Prompt**

You are a writing assistant specializing in editing writing produced by food scientists. I will give you some text to edit.
Instructions:
1. Convert all suggestions to a numbered list, with a title for each suggestion. Do not include any content that is not part of the numbered list.
2. Do not change the length.
3. Remove any references to the author's personal experience or to other writing.
4. Rewrite it as if it could have come from either a human or LLM.
5. Use complete sentences.
6. Do not add any prefix like 'Here is the edited text'. Just output the edited text.
7. Do not add or remove any of the meaning, unless necessary for following instruction #3.
8. Do not use asterisks.
Here is the text:
*<original text>*

---

*Figure 8.* Prompt for style standardization step of the experimental design task.

|  | Phase 1 ($n = 15$) | Phase 2 ($n = 15$) | All ($n = 20$) |
| --- | --- | --- | --- |
| Mean Years of Experience in Plant-Based Food Science | 4.33 (2.78) | 4.73 (5.30) | 5.05 (4.73) |
| Mean Years of Experience in Plant-Based Meat | 2.23 (2.01) | 3.27 (3.82) | 3.08 (3.38) |

*Table 6.* Properties of food scientists. Standard deviation is in parentheses. 10 food scientists participated in both Phase 1 and Phase 2, but we ensured that they did not evaluate their own experimental designs.

- Time saved: How much time, if any, do you think collaborating with this scientist would save you in designing a full (detailed and implementable) experimental design plan to improve this product? Please specify in hours.

**D.2. Menu Design**

The original menu we use is from Banerjee et al. (2023), who chose the items from Deliveroo's and Just Eat's top 100 items ordered in the United Kingdom in 2019. They also adjusted the items following pilot surveys. The menu includes 19 non-vegetarian and 17 vegetarian or vegan items. The original paper says 18 non-vegetarian and 18 vegetarian or vegan

options, but a dish called "Panchetta Carbonara" was included in the vegetarian section. For some items, the original menu included a "(v)" parenthetical indicating that a vegan option is available. We removed these parentheticals for simplicity in survey design and computing GHG emissions. Following Banerjee et al. (2023), GHG emissions were estimated based on the main ingredient of the dish. Data from Poore & Nemecek (2018) were used to compute emissions. For animal welfare, data from Faunalytics, a leading animal welfare organization, on number of animals killed per kilogram of various meats were used (Faunalytics, 2020). Two meats in the original menu, lamb and duck, were not contained in the Faunalytics data, and instead were imputed using the value for turkey, an animal of roughly similar size. Table 7 shows the sample sizes per arm in the Prolific experiments.

| Arm | Sample Size |
| --- | --- |
| Original | 50 |
| Vegetarian Subset | 50 |
| Remove Beef | 50 |
| Vegetarian First | 50 |
| o1-preview | 49 |
| o1-preview+IQP | 50 |
| o1-preview RD | 50 |
| GPT-4o+IQP | 50 |
| Llama+IQP | 52 |
| Claude+IQP | 50 |
| Gemini+IQP | 51 |

*Table 7.* Sample sizes per arm in Prolific randomized experiments for the menu design task. "o1-preview RD" is an ablation in which o1-preview's descriptions were replaced with a simple list of ingredients.

Other than the choice of dish, the participants were also asked the following questions:

- How satisfied are you with your set of choices? 1: not at all. 10: very satisfied.

- How likely would you be to visit this restaurant, assuming it is affordable and a convenient distance from you? 1: not likely at all. 10: very likely.

- How likely would you be to recommend this restaurant to your friends? 1: not likely at all. 10: very likely.

### D.3. Sensory Profile Prediction

Figure 12 shows the prompt for the sensory profile prediction task. Table 8 shows the number of pairs for each sensory dimension. Table 9 shows sample sizes across all prediction tasks.

| | $N$ |
| --- | --- |
| Overall Satisfaction | 85 |
| Meatiness | 61 |
| Greasiness | 103 |
| Juiciness | 73 |
| Sweetness | 28 |
| Saltiness | 63 |
| Purchase | 82 |
| All Dimensions | 495 |

*Table 8.* Sample sizes (number of pairs) for each sensory dimension in the sensory profile prediction task.

### D.4. Recipe Preference Prediction

Figure 13 shows the prompt for the recipe preference prediction task.

---

**Prompt**

You are a brilliant chef experienced at creating sustainable and delicious food.
Here is a menu: *<original menu>*
Please generate a revised menu, with the same number of recipes (*<n>*) and no new ingredients other than tofu, lentils, mushrooms, chickpeas, eggs, and cheese.
Design the menu to achieve at least a 75% $CO_2$ emissions reduction in people's choices while maintaining or improving patron satisfaction with their set of choices.
Patrons will be American omnivores. Emissions will be computed based on the main (first) ingredient.
Please output each recipe in same format as this example:
*Tofu curry ramen*
*Fried tofu, noodles, curry broth, pak choi, pickled onions.*
*Appealing description.*
The ingredients must be in order of usage, i.e the main ingredient must come first.
Very important: you must only use ingredients in the original menu or the list above. For every ingredient, there must be an exact match in the original menu or the list above.
Do not worsen cost, nutrition, animal welfare (number of animals used, computed based on the first ingredient), or preparation time.
Do not include any stars, asterisks, hashtags, underscores. Do not number the recipes. Do not include any text other than recipe information, e.g. do not say 'Here are the recipes'.

---

*Figure 9.* LLM prompt used in the menu design ablation, in which o1-preview on its own was asked to directly revise the original menu.

---

**Prompt**

You are a brilliant chef experienced at creating sustainable and delicious food. Here is a menu: *<original menu>*
Please generate (*<k>*) new, delicious, and diverse vegan or vegetarian dishes from this set of ingredients. You are also allowed to use tofu, lentils, mushrooms, chickpeas, eggs, and cheese.
Patrons will be American omnivores.
Please output in same format as this example:
*Tofu curry ramen*
*Fried tofu, noodles, curry broth, pak choi, pickled onions.*
*Appealing description.*
The ingredients must be in order of usage, i.e the main ingredient must come first.
Very important: you must only use ingredients in the original menu or the list above. For every ingredient, there must be an exact match in the original menu or the list above.
Do not worsen $CO_2$ emissions, cost, nutrition, or preparation time. Emissions will be computed based on the main (first) ingredient.
Do not include any stars, asterisks, hashtags, underscores. Do not number the recipes. Do not include any text other than recipe information, e.g. do not say 'Here are the recipes'.

---

*Figure 10.* LLM prompt used in the recipe generation stage of the LLM+IQP approach.

## E. Supplementary Results

### E.1. Experimental Design

Figures 14 and 15 show an example output from o1-preview.

### E.2. Menu Design

Table 11 shows accuracy of the predicted preferences.

---

**Prompt**

Here are $<r>$ recipes. Please rate them on a scale of 1-10 based on standard American omnivore taste preferences, 1 being unappealing and 10 being appealing. Output only a comma-separated list of $<r>$ numbers, from 1 to 10. $<recipes>$

---

*Figure 11.* LLM prompt used in the rating stage of the LLM+IQP approach.

---

**Prompt**

You are an expert plant-based meat food scientist. Here are the ingredient lists of two $<category>$ products.
Product 1: $<ingredient\ list>$
Product 2: $<ingredient\ list>$
Additionally, here are the nutrition facts for product 1: $<nutrition\ facts>$.
And here are the nutrition facts for product 2: $<nutrition\ facts>$.
Now, suppose that a group of 100 omnivores eats both products in a blind taste test. Which do you predict would be ranked higher on the dimension of $<dimension>$? Please output a single character, either 1 or 2 on the first line.

---

*Figure 12.* LLM prompt for the sensory profile prediction task.

| | Claude 3.5 Sonnet | Gemini 1.5 Pro | GPT-3.5 Turbo | GPT-4o | Llama | o1-preview | Baseline |
|---|---|---|---|---|---|---|---|
| **Sensory Profile Prediction: LLM Prediction vs. Sensory Panel (Sustainable Protein)** | | | | | | | |
| All Dimensions | 495 | 495 | 495 | 495 | 495 | 495 | 495 |
| Overall Satisfaction | 85 | 85 | 85 | 85 | 85 | 85 | 85 |
| Meatiness | 61 | 61 | 61 | 61 | 61 | 61 | 61 |
| Greasiness | 103 | 103 | 103 | 103 | 103 | 103 | 103 |
| Juiciness | 73 | 73 | 73 | 73 | 73 | 73 | 73 |
| Sweetness | 28 | 28 | 28 | 28 | 28 | 28 | 28 |
| Saltiness | 63 | 63 | 63 | 63 | 63 | 63 | 63 |
| Purchase | 82 | 82 | 82 | 82 | 82 | 82 | 82 |
| **Recipe Preference Prediction (`Food.com`): LLM Prediction vs. Rating (Traditional Foods)** | | | | | | | |
| Rating | 500 | 500 | 500 | 500 | 500 | 500 | 500 |
| **Recipe Preference Prediction (Menu Design): LLM Prediction vs. Order Frequency (Traditional Foods)** | | | | | | | |
| Order Frequency | 1225 | 1275 | - | 1225 | 1326 | 1225 | 1225 |

*Table 9.* Sample sizes (number of pairs) for prediction tasks.

### E.3. Sensory Profile Prediction

The performance of an expert food scientist specifically for the 'Overall Satisfaction' dimension is shown in Table 14. Accuracies in each quartile of the ground truth preference gap are shown in Tables 15, 16, 17, 18.

### E.4. Recipe Preference Prediction

Tables 19 and 20 list supplementary analyses of performance on the recipe preference task. Table 21 shows performance on a smaller held-out set.

---

**Prompt**

You are an expert online recipe writer. Which online recipe would people prefer, recipe 1 or recipe 2?
Output a number only (1 or 2). You must choose one. If unsure, provide your best guess.
Recipe 1: *<recipe text>*
Recipe 2: *<recipe text>*
Answer:

---

*Figure 13.* LLM prompt for the recipe preference prediction task.

---

**Example Output**

1. Appearance Enhancement
Many participants found the appearance unappetizing, describing it as "looks like a dog treat," "raw," or "fake."
- Experiment: Test natural coloring agents to achieve a more authentic bacon look. Incorporate beet juice or paprika extract to enhance redness and use coconut oil for marbling effects.
- Variables: Vary concentrations of coloring agents and adjust cooking times and temperatures to achieve desired caramelization.
- Metrics: Conduct sensory evaluations focusing on appearance, measuring scores against the current mean of 2.47.
2. Texture Improvement
Issues with a mushy center and inconsistent crispiness were noted.
- Experiment: Adjust the protein blend ratios and incorporate texturizers like methylcellulose to enhance firmness and coherence.
- Variables: Test different ratios of soy and wheat proteins; vary levels of konjac powder and vegetable gums.
- Metrics: Use texture profile analysis to measure hardness and chewiness; collect sensory feedback on texture aiming to improve the mean from 3.43 towards 4 (neutral).
3. Flavor Enhancement
Participants desired more smokiness and found the taste bland.
- Experiment: Increase natural smoke flavoring and seasoning levels to enhance meatiness and smokiness.
- Variables: Experiment with different types and concentrations of smoke extracts (e.g., hickory, applewood) and adjust salt and sugar levels.
- Metrics: Sensory evaluations on flavor and smokiness, targeting improvements from means of 3.58 and 2.43 towards 4.
4. Reduce Greasiness
Some found the product too greasy.
- Experiment: Reduce sunflower oil content or replace it with a less greasy alternative like canola oil.
- Variables: Vary oil types and quantities.
- Metrics: Measure perceived greasiness in sensory tests, aiming to adjust the mean from 2.53 to 3 (moderately greasy).

---

*Figure 14.* Example o1-preview output for the experimental design task, after style standardization.

---

**Example Output**

1. Improve Texture and Crispiness. Issue: The product is perceived as too soft and lacking crispiness (Mean crispiness rating: 5.86; ideal is 4).
Changes:
- Adjust ingredient composition by incorporating crisping agents like rice flour or modifying the ratio of wheat gluten to reduce chewiness.
- Modify the process by slicing the product thinner and adjusting cooking temperatures and times to achieve a crisper texture.
Experiment:
- Design a factorial experiment testing different levels of rice flour (e.g., 1%, 2%, 3%) and cooking temperatures (e.g., 350°F, 375°F, 400°F).
- Measure the crispiness using texture analysis and conduct sensory evaluations.
2. Enhance Flavor Profile. Issue: Flavor is disliked, with complaints of excessive saltiness and artificial taste (Mean flavor rating: 2.22; saltiness rating skewed towards too salty). Changes:
- Reduce salt content by gradually decreasing salt levels by 10-20%.
- Increase umami and smokiness by adding natural umami enhancers like mushroom extract or increasing natural smoke flavors.
- Balance sweetness by introducing a small amount of natural sweeteners like maple syrup to mimic bacon's sweet-salty balance. Experiment:
- Create a series of formulations with varying salt levels (e.g., full, -10%, -20%) and added sweeteners.
- Conduct sensory tests focusing on flavor liking and saltiness perception.
3. Improve Appearance. Issue: Appearance is inconsistent; some find it artificial (Mixed feedback on color; Mean color rating: 4.42).
Changes:
- Introduce marbling by using natural colorants to create a marbled fat appearance, mimicking real bacon.
- Adjust color by modifying the hue using oleoresin paprika for red tones and annatto for lighter shades.
Experiment:
- Develop prototypes with different marbling techniques and colorant levels.
- Perform visual assessments with consumers to rate appearance authenticity.
4. Increase Greasiness. Issue: Product is not perceived as greasy enough (Mean greasiness rating: 1.39; ideal is 3).
Changes:
- Incorporate plant-based fats by increasing the amount of oils like coconut or sunflower oil.
- Use fat release technology by employing oil encapsulation that melts during cooking to simulate bacon fat.
Experiment:
- Test formulations with varying fat levels (e.g., 2%, 4%, 6% additional oil). - Evaluate greasiness through sensory panels and measure perceived oiliness.

*Figure 15.* Example o1-preview output for the experimental design task, after style standardization.

| Test | Accuracy | Complementarity | Specificity | Percent Time Saved |
|---|---|---|---|---|
| $t$-test | 0.121 | 0.350 | **0.003** | **0.0002** |
| Paired $t$-test | 0.125 | 0.368 | **0.003** | **0.0003** |
| Wilcoxon signed-rank test | 0.078 | 0.284 | **0.005** | **0.0007** |
| Linear mixed model with random effect for evaluator ID | 0.075 | 0.993 | **0.001** | **0.0002** |
| Permutation test | 0.140 | 0.392 | **0.003** | **0.0003** |
| OLS | 0.1812 | 0.416 | **0.001** | **0.0006** |

*Table 10.* Sensitivity of $p$-values in the experimental design task to choice of statistical test. The paired $t$-test takes into account the pairing across products. The Wilcoxon signed-rank test is exact, for paired samples, nonparametric, and does not rely on normality assumptions. The permutation is also exact and does not rely on normality assumptions. In the OLS, we control for product ID, evaluator ID, generator ID as fixed effects.

| LLM | Accuracy |
|---|---|
| **Claude 3.5 Sonnet** | **0.85** |
| **Gemini 1.5 Pro** | **0.72** |
| **GPT-3.5 Turbo** | - |
| **GPT-4o** | **0.82** |
| **Llama 3.1 70b Instruct** | **0.72** |
| o1-preview | 0.50 |
| **o1-preview, RD** | **0.84** |

*Table 11.* Accuracy in the fourth (highest) quartile of the preference gap, for predicted preferences in the menu design task vs. actual order frequency. The 25th and 50th percentile of the preference gap were both 1 (the minimum gap needed for inclusion in the analysis in Table 2), so we did not test those quartiles. Accuracies statistically significant according to chi-squared test are in bold. GPT-3.5 Turbo did not produce a valid output for this task and thus was not tested. We note that LLM predictions were based on the recipe title alone (not including the full description), which could explain the discrepancy between item-level prediction performance and menu-level performance for o1-preview. When its descriptions were replaced with a simple list of ingredients (in one of the ablations), o1-preview's accuracy in this quartile was 84% ($p$=2.40e-20).

| Outcome | Treatment Effect Estimate (p-value) | Percent Change |
|---|---|---|
| Emissions | 34.31 (1.37e-18) | 78.55 |
| Satisfaction with Set of Choices | -0.04 (0.93) | -0.58 |
| Likelihood of Visiting | -0.07 (0.89) | -0.95 |
| Likelihood of Recommending to a Friend | -0.38 (0.47) | -4.9 |
| Animal Usage | 0.00 (0.91) | -3.5 |

*Table 12.* Treatment Effect Estimates ($p$-value in parentheses) and percent change for o1-preview+IQP vs. the original menu, adjusting (via linear regression) for the covariates of age, gender, race, and dietary preference. Signs are flipped so that positive values are better in all cases. Compared to the original menu, o1-preview+IQP reduces average emissions by 79% while maintaining participants's satisfaction with their set of choices to within 1% of the original. Small reductions are observed in likelihood of visiting, likelihood of recommending to a friend, and animal welfare, but none are statistically significant.

| | Emissions | Satisfied | Visit | Recommend | Animals | PB | LLM Gen. (AO) | LLM Gen. (AC) |
|---|---|---|---|---|---|---|---|---|
| Original | 44.91 | 8.40 | 7.90 | 7.82 | 0.08 | 0.20 | 0.00 | 0.00 |
| Expert Chef | 41.28 | 8.20 | 7.76 | 7.49 | 0.08 | 0.27 | 0.00 | 0.00 |
| Vegetarian Subset | 19.87 | 6.82 | 6.76 | 6.36 | 0.00 | 1.00 | 0.00 | 0.00 |
| Remove Beef | 21.65 | 7.74 | 7.50 | 7.08 | 0.16 | 0.24 | 0.00 | 0.00 |
| Beef to Chicken | 17.57 | 8.04 | 8.00 | 7.98 | 0.15 | 0.36 | 0.00 | 0.00 |
| Vegetarian First | 37.37 | 8.20 | 7.74 | 7.62 | 0.13 | 0.28 | 0.00 | 0.00 |
| o1-preview+IQP | 8.70 | 8.44 | 7.80 | 7.46 | 0.08 | 0.54 | 0.50 | 0.44 |
| o1-preview | 9.64 | 6.51 | 6.29 | 6.22 | 0.00 | 1.00 | 1.00 | 1.00 |
| o1-preview+IQP, RD | 11.05 | 7.48 | 6.98 | 6.76 | 0.18 | 0.36 | 0.50 | 0.20 |
| o1-preview+IQP, RP | 9.44 | 7.84 | 7.63 | 7.33 | 0.12 | 0.55 | 0.36 | 0.39 |
| o1-preview+IQP, RDi | 9.75 | 8.16 | 7.96 | 7.66 | 0.13 | 0.50 | 0.47 | 0.34 |
| o1-preview+IQP, RPDi | 8.54 | 7.12 | 6.56 | 6.50 | 0.07 | 0.54 | 0.47 | 0.42 |
| Claude+IQP | 10.16 | 7.30 | 6.68 | 6.86 | 0.16 | 0.38 | 0.42 | 0.28 |
| Gemini+IQP | 11.71 | 7.59 | 7.35 | 7.02 | 0.19 | 0.31 | 0.44 | 0.10 |
| GPT-4o+IQP | 8.53 | 7.38 | 7.08 | 6.98 | 0.17 | 0.52 | 0.42 | 0.36 |
| Llama+IQP | 10.68 | 7.77 | 7.54 | 7.21 | 0.12 | 0.58 | 0.50 | 0.31 |

*Table 13.* Table of all results for the menu design task. "Emissions" is in kg of $CO_2$eq per kg of food item. "Animal Lives" is number of animals killed per kg of food item. "Satisfied", "Visit", and "Recommend" are on a ten point (1 to 10) scale. "PB" is the fraction of choices that were plant based (either vegan or vegetarian). "LLM Gen. (AO)" is the fraction of available options in the final optimized menu that were LLM generated (as opposed to recipes from the original menu). "LLM Gen. (AC)" is the fraction of actual choices that were LLM generated (as opposed to recipes from the original menu). In "o1-preview+IQP, RD", o1-preview's descriptions are replaced with a simple list of ingredients. In "o1-preview+IQP, RP", the preferences component of the objective (the first term in Problem 2) is removed. In "o1-preview+IQP, RDi", the diversity component of the objective is removed, i.e. $\lambda = 0$ in Problem 2. In "o1-preview+IQP, RPDi", both the preference and diversity components of the objective in Problem 2 are removed, i.e. it becomes a feasibility problem.

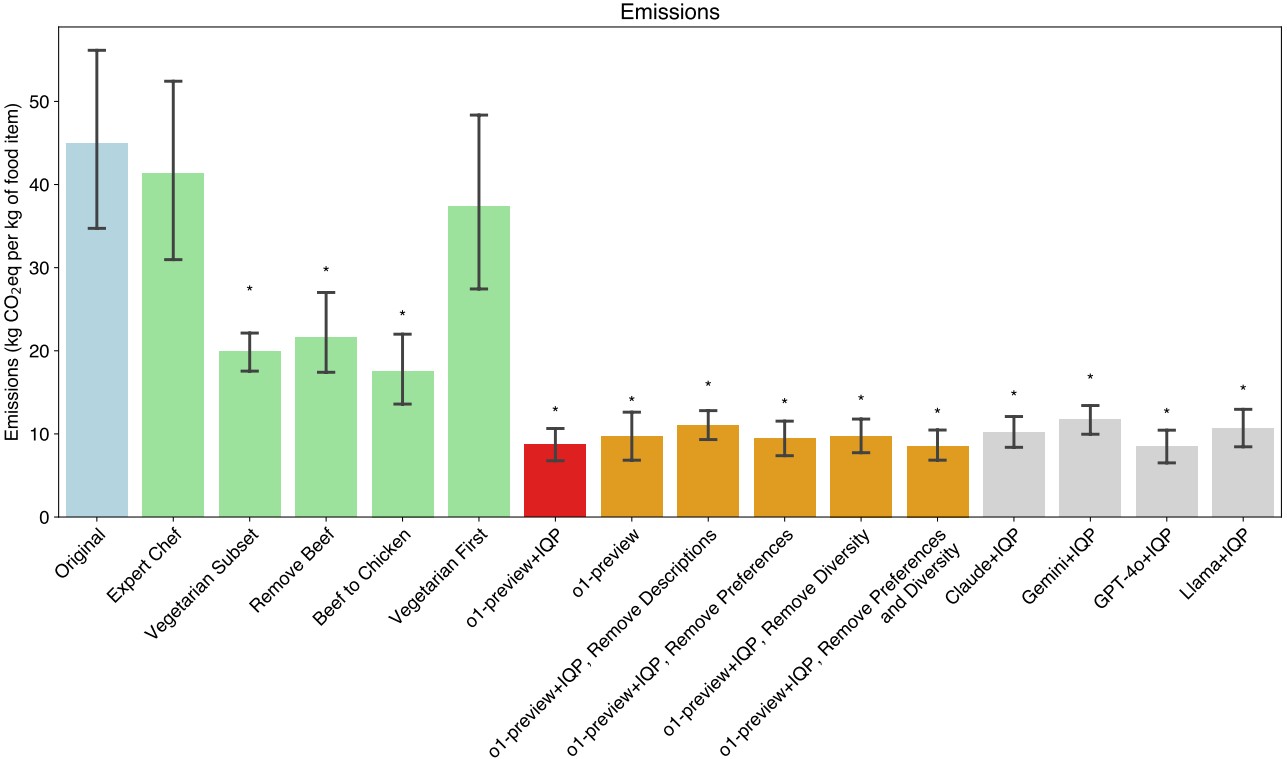

*Figure 16.* Average emissions for all arms. $n = 800$ across arms. Lower is better. Asterisks indicate a statistically significant difference ($t$-test with Bonferroni correction) compared to the original menu. Baselines are in green, ablations are in orange. In "o1-preview+IQP, Remove Descriptions", o1-preview's descriptions are replaced with a simple list of ingredients. In "o1-preview+IQP, Remove Preferences", the preferences component of the objective (the first term in Problem 2) is removed. In "o1-preview+IQP, Remove Diversity", the diversity component of the objective is removed, i.e. $\lambda = 0$ in Problem 2. In "o1-preview+IQP, Remove Preferences and Diversity", both the preference and diversity components of the objective in Problem 2 are removed, i.e. it becomes a feasibility problem.

| Data Subset | Expert | Claude 3.5 Sonnet | Gemini 1.5 Pro | GPT-3.5 Turbo | GPT-4o | Llama | o1-preview | Nutr. Baseline |
|---|---|---|---|---|---|---|---|---|
| All | 0.65 | 0.60 | 0.41 | 0.38 | 0.39 | 0.51 | 0.64 | 0.73 |
| Quartile 1 | 0.62 | 0.43 | 0.29 | 0.43 | 0.33 | 0.43 | 0.43 | 0.71 |
| Quartile 2 | 0.60 | 0.65 | 0.45 | 0.35 | 0.40 | 0.55 | 0.60 | 0.80 |
| Quartile 3 | 0.77 | 0.64 | 0.59 | 0.41 | 0.64 | 0.59 | 0.68 | 0.55 |
| Quartile 4 | 0.59 | 0.67 | 0.29 | 0.33 | **0.19** | 0.48 | **0.81** | **0.86** |

*Table 14.* Comparison of performance of the six LLMs on the sensory profile prediction task, specifically the "Overall Satisfaction" dimension, against an expert human food scientist. Statistically significant results (chi-squared test with Bonferroni correction) are in bold.

| | Claude 3.5 Sonnet | Gemini 1.5 Pro | GPT-3.5 Turbo | GPT-4o | Llama | o1-preview | Baseline |
|---|---|---|---|---|---|---|---|
| Overall Satisfaction | 0.43 | 0.29 | 0.43 | 0.33 | 0.43 | 0.43 | 0.71 |
| Meatiness | 0.27 | 0.33 | 0.40 | 0.47 | 0.27 | 0.40 | 0.47 |
| Greasiness | 0.65 | 0.69 | 0.46 | **0.77** | **0.77** | 0.65 | 0.69 |
| Juiciness | 0.44 | 0.67 | 0.67 | 0.67 | 0.33 | 0.61 | 0.50 |
| Sweetness | 0.29 | 0.71 | 0.86 | 0.71 | 0.71 | 0.71 | 0.86 |
| Saltiness | 0.75 | 0.81 | 0.62 | 0.81 | 0.69 | 0.75 | 0.56 |
| Purchase | 0.60 | 0.45 | 0.50 | 0.50 | 0.50 | 0.50 | 0.45 |
| All Dimensions | 0.52 | 0.55 | 0.53 | 0.60 | 0.53 | 0.57 | 0.59 |

*Table 15.* Accuracies of six LLMs on the sensory profile prediction task, quartile 1 of the preference gap. Statistically significant results (chi-squared test with Bonferroni correction) are in bold.

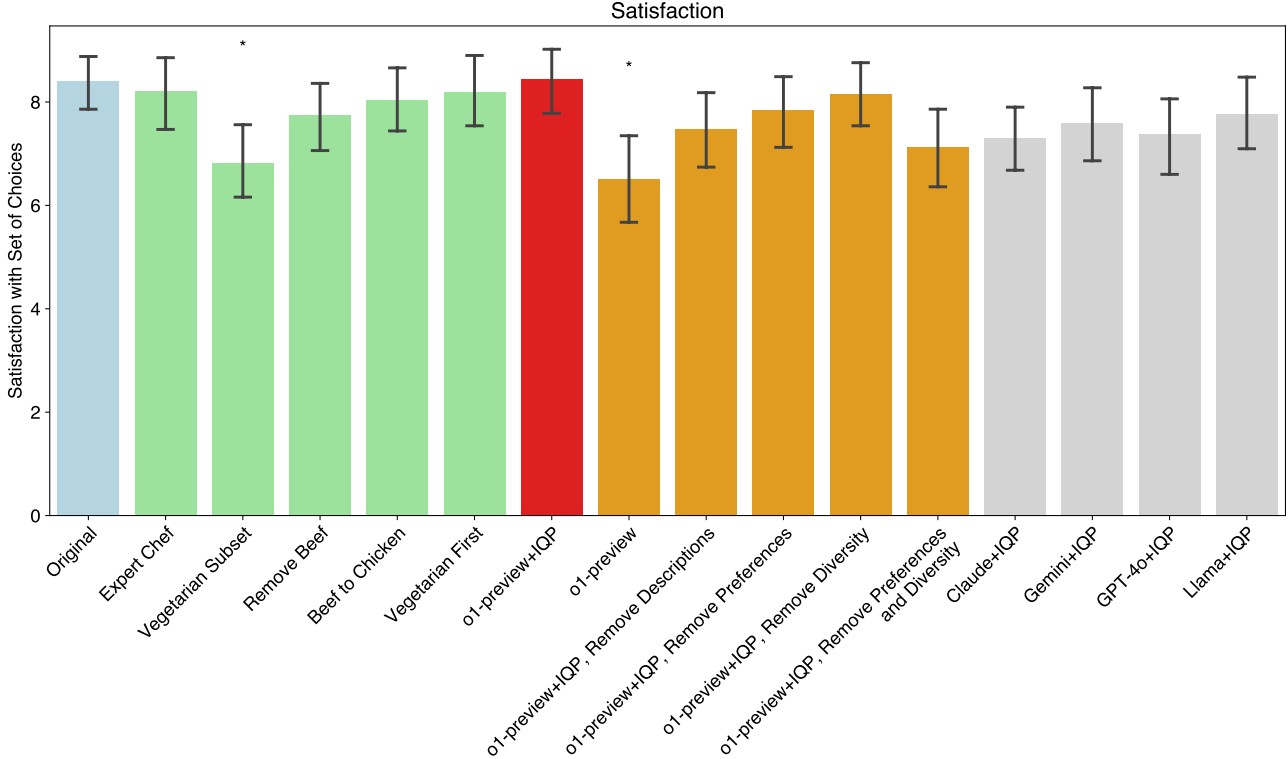

*Figure 17.* Average satisfaction for all arms. $n = 800$ across arms. Higher is better. Asterisks indicate a statistically significant difference ($t$-test with Bonferroni correction) compared to the original menu. Baselines are in green, ablations are in orange. In "o1-preview+IQP, Remove Descriptions", o1-preview's descriptions are replaced with a simple list of ingredients. In "o1-preview+IQP, Remove Preferences", the preferences component of the objective (the first term in Problem 2) is removed. In "o1-preview+IQP, Remove Diversity", the diversity component of the objective is removed, i.e. $\lambda = 0$ in Problem 2. In "o1-preview+IQP, Remove Preferences and Diversity", both the preference and diversity components of the objective in Problem 2 are removed, i.e. it becomes a feasibility problem.

|  | Claude 3.5 Sonnet | Gemini 1.5 Pro | GPT-3.5 Turbo | GPT-4o | Llama | o1-preview | Baseline |
|---|---|---|---|---|---|---|---|
| Overall Satisfaction | 0.65 | 0.45 | 0.35 | 0.40 | 0.55 | 0.60 | 0.80 |
| Meatiness | **0.87** | 0.80 | 0.60 | 0.67 | 0.60 | 0.80 | 0.53 |
| Greasiness | 0.52 | 0.60 | 0.72 | 0.72 | 0.64 | 0.56 | 0.68 |
| Juiciness | 0.65 | 0.71 | 0.71 | 0.53 | 0.59 | 0.59 | 0.71 |
| Sweetness | 0.29 | 0.43 | 0.57 | 0.29 | 0.14 | 0.43 | 0.43 |
| Saltiness | 0.67 | 0.60 | 0.67 | 0.60 | 0.73 | 0.53 | 0.60 |
| Purchase | 0.62 | **0.10** | 0.38 | 0.38 | 0.33 | 0.57 | 0.57 |
| All Dimensions | **0.62** | 0.52 | 0.57 | 0.53 | 0.54 | 0.59 | **0.64** |

*Table 16.* Accuracies of six LLMs on the sensory profile prediction task, quartile 2 of the preference gap. Statistically significant results (chi-squared test with Bonferroni correction) are in bold.

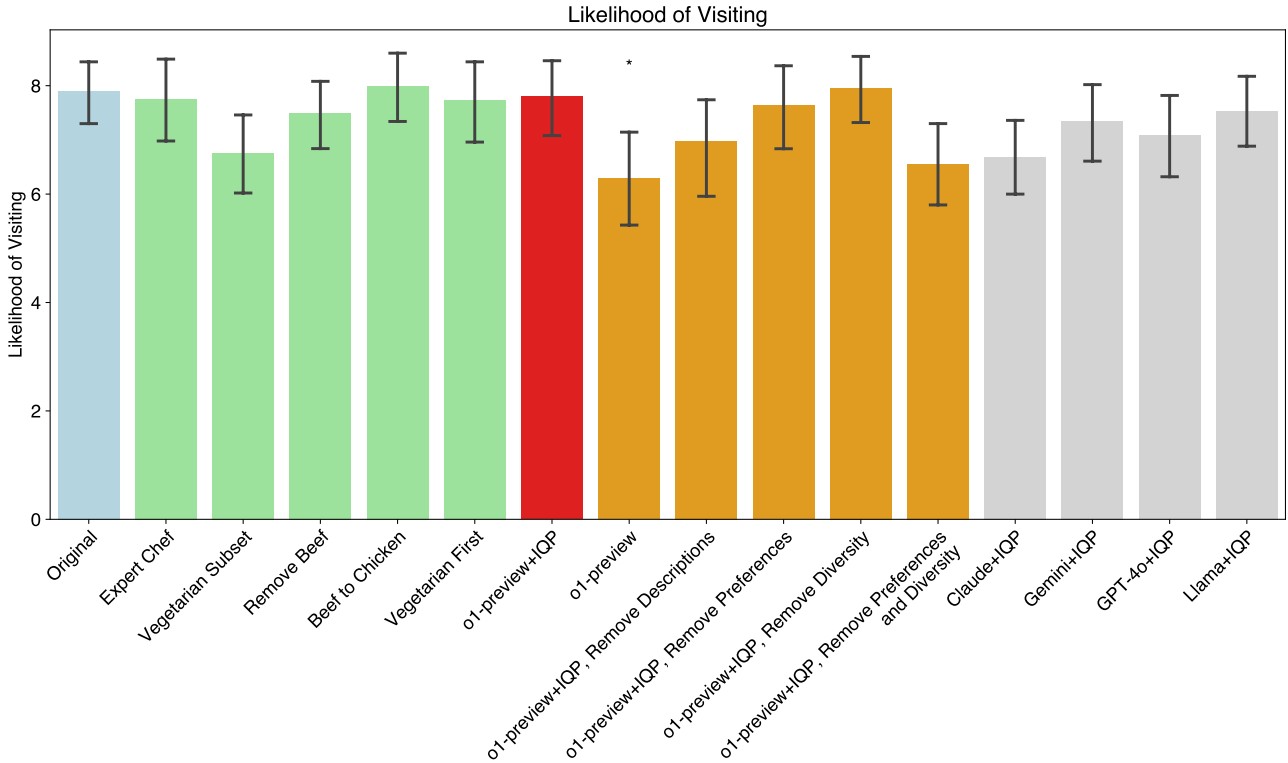

*Figure 18.* Average likelihood of visiting the hypothetical restaurant for all arms. $n = 800$ across arms. Higher is better. Asterisks indicate a statistically significant difference ($t$-test with Bonferroni correction) compared to the original menu. Baselines are in green, ablations are in orange. In "o1-preview+IQP, Remove Descriptions", o1-preview's descriptions are replaced with a simple list of ingredients. In "o1-preview+IQP, Remove Preferences", the preferences component of the objective (the first term in Problem 2) is removed. In "o1-preview+IQP, Remove Diversity", the diversity component of the objective is removed, i.e. $\lambda = 0$ in Problem 2. In "o1-preview+IQP, Remove Preferences and Diversity", both the preference and diversity components of the objective in Problem 2 are removed, i.e. it becomes a feasibility problem.

| | Claude 3.5 Sonnet | Gemini 1.5 Pro | GPT-3.5 Turbo | GPT-4o | Llama | o1-preview | Baseline |
|---|---|---|---|---|---|---|---|
| Overall Satisfaction | 0.64 | 0.59 | 0.41 | 0.64 | 0.59 | 0.68 | 0.55 |
| Meatiness | 0.60 | 0.60 | 0.60 | 0.67 | 0.60 | 0.60 | 0.53 |
| Greasiness | 0.65 | 0.69 | 0.65 | **0.85** | **0.81** | 0.73 | **0.77** |
| Juiciness | 0.68 | 0.53 | 0.58 | 0.68 | 0.53 | 0.79 | 0.74 |
| Sweetness | 0.43 | 0.43 | 0.43 | 0.57 | 0.43 | 0.43 | 0.71 |
| Saltiness | 0.69 | 0.69 | 0.50 | 0.81 | 0.62 | **0.88** | **0.88** |
| Purchase | 0.55 | 0.30 | 0.25 | 0.45 | 0.40 | 0.35 | 0.75 |
| All Dimensions | **0.62** | 0.56 | 0.50 | **0.68** | 0.59 | **0.66** | **0.70** |

*Table 17.* Accuracies of six LLMs on the sensory profile prediction task, quartile 3 of the preference gap. Statistically significant results (chi-squared test with Bonferroni correction) are in bold.

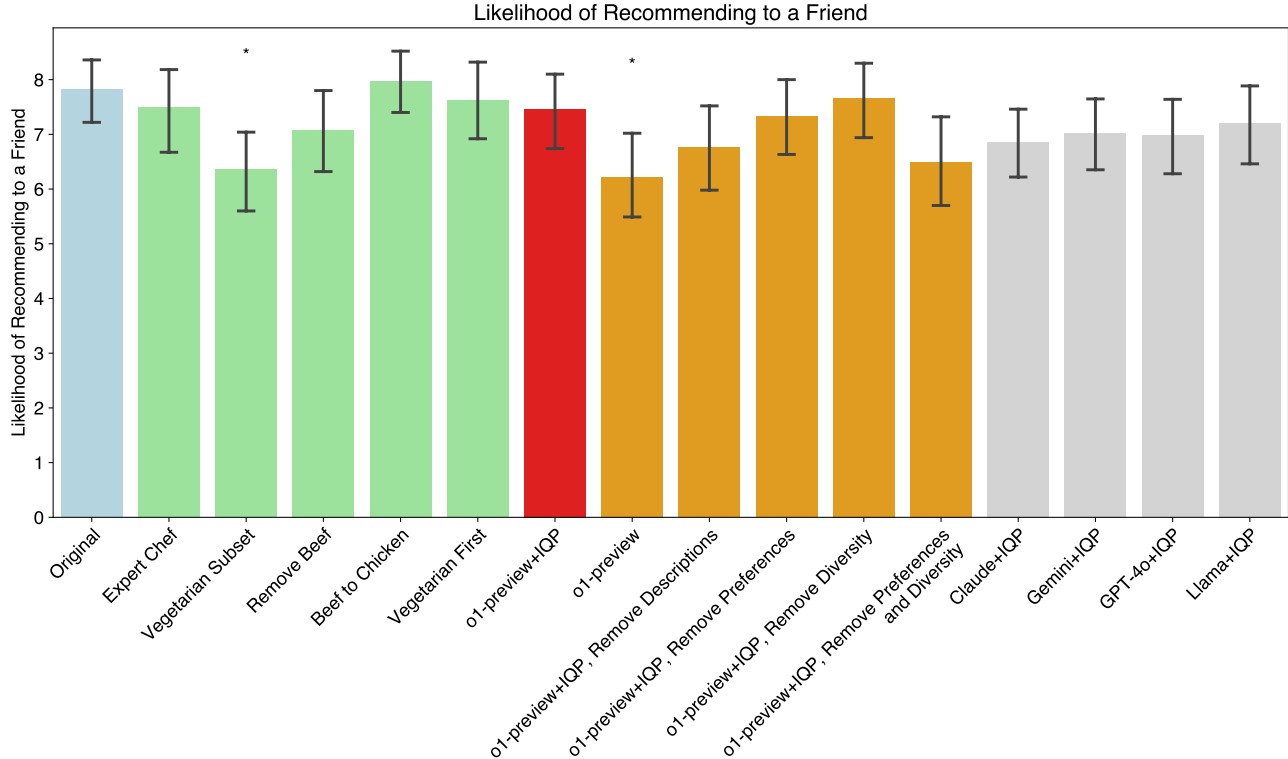

*Figure 19.* Average likelihood of recommending the hypothetical restaurant to a friend for all arms. $n = 800$ across arms. Higher is better. Asterisks indicate a statistically significant difference ($t$-test with Bonferroni correction) compared to the original menu. Baselines are in green, ablations are in orange. In "o1-preview+IQP, Remove Descriptions", o1-preview's descriptions are replaced with a simple list of ingredients. In "o1-preview+IQP, Remove Preferences", the preferences component of the objective (the first term in Problem 2) is removed. In "o1-preview+IQP, Remove Diversity", the diversity component of the objective is removed, i.e. $\lambda = 0$ in Problem 2. In "o1-preview+IQP, Remove Preferences and Diversity", both the preference and diversity components of the objective in Problem 2 are removed, i.e. it becomes a feasibility problem.

| | Claude 3.5 Sonnet | Gemini 1.5 Pro | GPT-3.5 Turbo | GPT-4o | Llama | o1-preview | Baseline |
|---|---|---|---|---|---|---|---|
| Overall Satisfaction | 0.67 | 0.29 | 0.33 | **0.19** | 0.48 | **0.81** | **0.86** |
| Meatiness | 0.67 | 0.53 | 0.40 | 0.73 | 0.60 | 0.80 | 0.40 |
| Greasiness | 0.50 | 0.58 | 0.69 | 0.62 | 0.69 | 0.62 | 0.62 |
| Juiciness | 0.67 | 0.50 | 0.56 | 0.67 | 0.61 | 0.56 | **0.89** |
| Sweetness | 0.43 | 0.86 | 0.57 | 0.86 | 0.71 | 1.00 | 1.00 |
| Saltiness | 0.69 | 0.56 | 0.50 | 0.69 | 0.75 | 0.81 | **0.88** |
| Purchase | 0.80 | 0.35 | 0.35 | 0.50 | 0.45 | 0.75 | 0.70 |
| All Dimensions | **0.64** | 0.49 | 0.49 | 0.57 | 0.60 | **0.73** | **0.74** |

*Table 18.* Accuracies of six LLMs on the sensory profile prediction task, quartile 4 of the preference gap. Statistically significant results (chi-squared test with Bonferroni correction) are in bold.

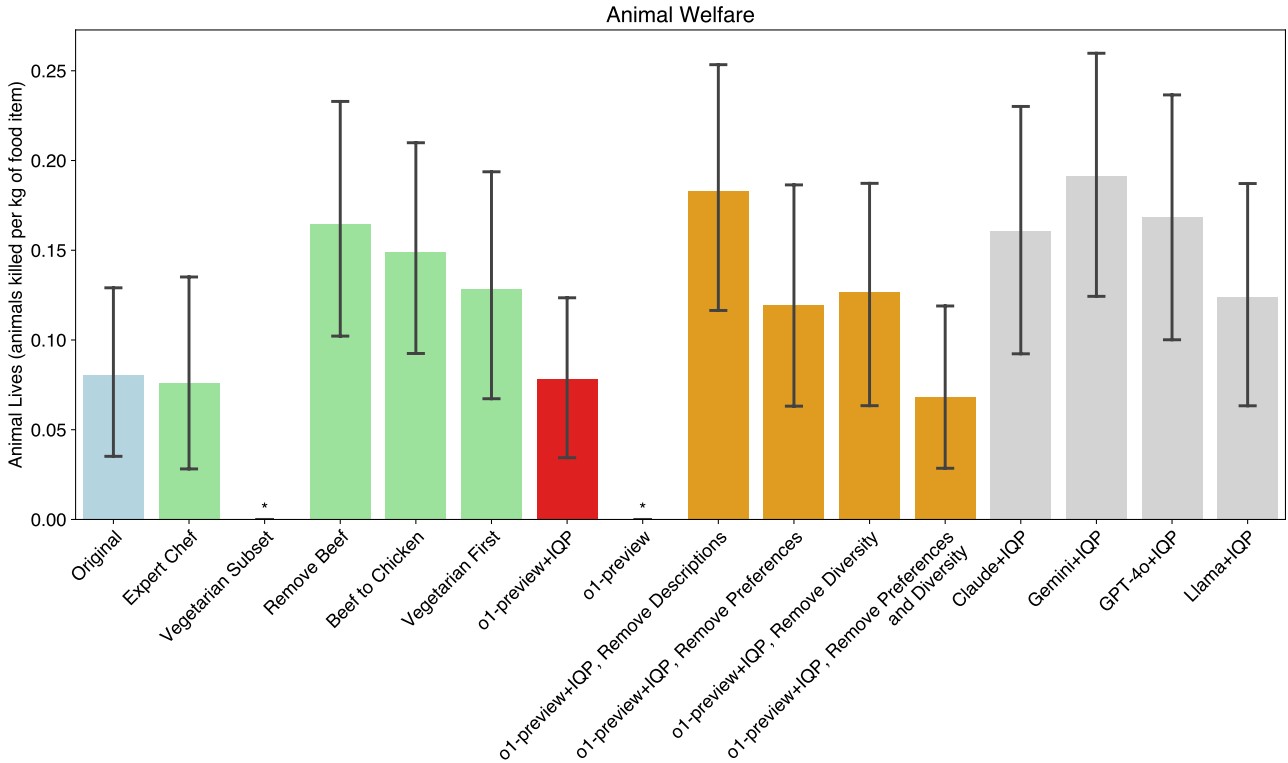

*Figure 20.* Average animal usage for all arms. $n = 800$ across arms. Lower is better for animal welfare. Asterisks indicate a statistically significant difference ($t$-test with Bonferroni correction) compared to the original menu. Baselines are in green, ablations are in orange. In "o1-preview+IQP, Remove Descriptions", o1-preview's descriptions are replaced with a simple list of ingredients. In "o1-preview+IQP, Remove Preferences", the preferences component of the objective (the first term in Problem 2) is removed. In "o1-preview+IQP, Remove Diversity", the diversity component of the objective is removed, i.e. $\lambda = 0$ in Problem 2. In "o1-preview+IQP, Remove Preferences and Diversity", both the preference and diversity components of the objective in Problem 2 are removed, i.e. it becomes a feasibility problem.

| LLM | Condition | | | | Overall |
|---|---|---|---|---|---|
| | **Q1** | **Q2** | **Q3** | **Q4** | |
| **GPT-4o** | **62.32%** | 65.31% | 67.74% | 71.43% | **64.00%** |
| **Claude 3.5 Sonnet** | **64.79%** | 61.22% | 58.06% | 85.71% | **63.20%** |
| **o1-preview** | 60.92% | 57.14% | 59.68% | 57.14% | **59.60%** |
| **GPT-3.5 Turbo** | 55.99% | 55.78% | 58.06% | 85.71% | 56.60% |
| **Llama 3.1 70b Instruct** | 53.17% | 48.3% | 43.55% | 57.14% | 50.60% |
| **Gemini 1.5 Pro** | 50.00% | 50.34% | 33.87% | 28.57% | 47.80% |

*Table 19.* Accuracies of six LLMs in the recipe preference prediction task, stratified by the quartiles of the difference in review scores. Statistically significant results (chi-squared test with Bonferroni correction) are in bold.

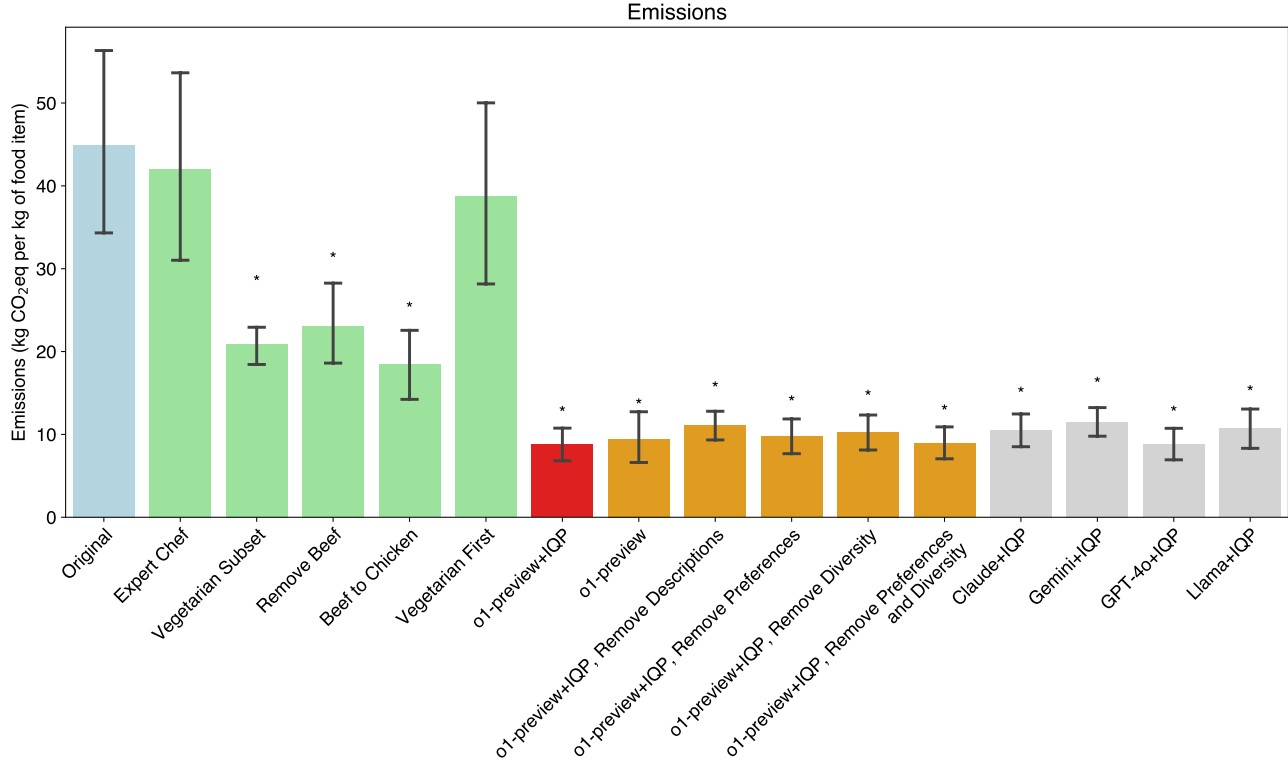

*Figure 21.* Average emissions for all arms, excluding vegan and vegetarian participants. $n = 740$ across arms. Lower is better. Asterisks indicate a statistically significant difference ($t$-test with Bonferroni correction) compared to the original menu. Baselines are in green, ablations are in orange. In "o1-preview+IQP, Remove Descriptions", o1-preview's descriptions are replaced with a simple list of ingredients. In "o1-preview+IQP, Remove Preferences", the preferences component of the objective (the first term in Problem 2) is removed. In "o1-preview+IQP, Remove Diversity", the diversity component of the objective is removed, i.e. $\lambda = 0$ in Problem 2. In "o1-preview+IQP, Remove Preferences and Diversity", both the preference and diversity components of the objective in Problem 2 are removed, i.e. it becomes a feasibility problem.

| LLM | Overall | ¬V vs. ¬V | V vs. ¬V | V vs. V | V > ¬V | V < ¬V |
|---|---|---|---|---|---|---|
| | | | **Condition** | | | |
| **GPT-4o** | **64.00%** | **70.81%** | 50.77% | 37.5% | 53.66% | **66.03%** |
| **Claude 3.5 Sonnet** | **63.20%** | **67.34%** | 56.15% | 41.67% | 43.9% | **66.99%** |
| **o1-preview** | **59.60%** | **65.32%** | 46.15% | 50.0% | 48.78% | **61.72%** |
| **GPT-3.5 Turbo** | 56.60% | **63.87%** | 42.31% | 29.17% | 40.24% | **59.81%** |
| **Llama 3.1 70b Instruct** | 50.60% | 54.91% | 42.31% | 33.33% | 43.9% | 51.91% |
| **Gemini 1.5 Pro** | 47.80% | 50.58% | 43.08% | 33.33% | 45.12% | 48.33% |

*Table 20.* Accuracies of six LLMs in the recipe preference prediction task, stratified by the ground truth comparison type: non-vegetarian vs. non-vegetarian (¬V vs. ¬V), vegetarian vs. non-vegetarian (V vs. ¬V), and vegetarian vs. vegetarian comparisons (V vs. V). Within the vegetarian vs. non-vegetarian comparisons, performance is displayed separately when vegetarian is preferred (V > ¬V) and when non-vegetarian is preferred (V < ¬V). Statistically significant results (chi-squared test with Bonferroni correction) are in bold.

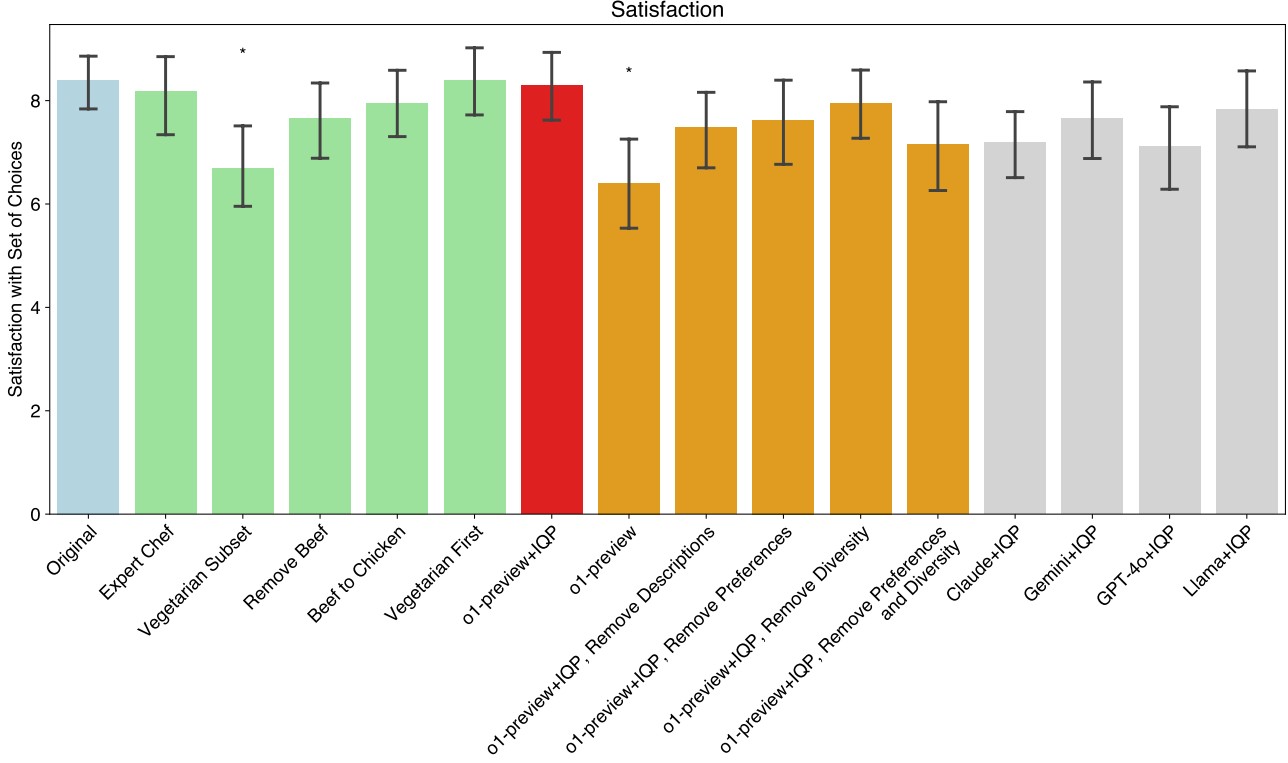

*Figure 22.* Average satisfaction for all arms, excluding vegan and vegetarian participants. $n = 740$ across arms. Higher is better. Asterisks indicate a statistically significant difference ($t$-test with Bonferroni correction) compared to the original menu. Baselines are in green, ablations are in orange. In "o1-preview+IQP, Remove Descriptions", o1-preview's descriptions are replaced with a simple list of ingredients. In "o1-preview+IQP, Remove Preferences", the preferences component of the objective (the first term in Problem 2) is removed. In "o1-preview+IQP, Remove Diversity", the diversity component of the objective is removed, i.e. $\lambda = 0$ in Problem 2. In "o1-preview+IQP, Remove Preferences and Diversity", both the preference and diversity components of the objective in Problem 2 are removed, i.e. it becomes a feasibility problem.

| LLM | Accuracy |
|---|---|
| **Claude 3.5 Sonnet** | 56.90% |
| **Gemini 1.5 Pro** | 41.38% |
| **GPT-3.5 Turbo** | 55.17% |
| **GPT-4o** | 55.17% |
| **Llama 3.1 70b Instruct** | 74.14% |
| **o1-preview** | 68.97% |

*Table 21.* **Recipe preference prediction: Post-cut-off dataset.** Overall accuracy on the post-cut-off dataset.

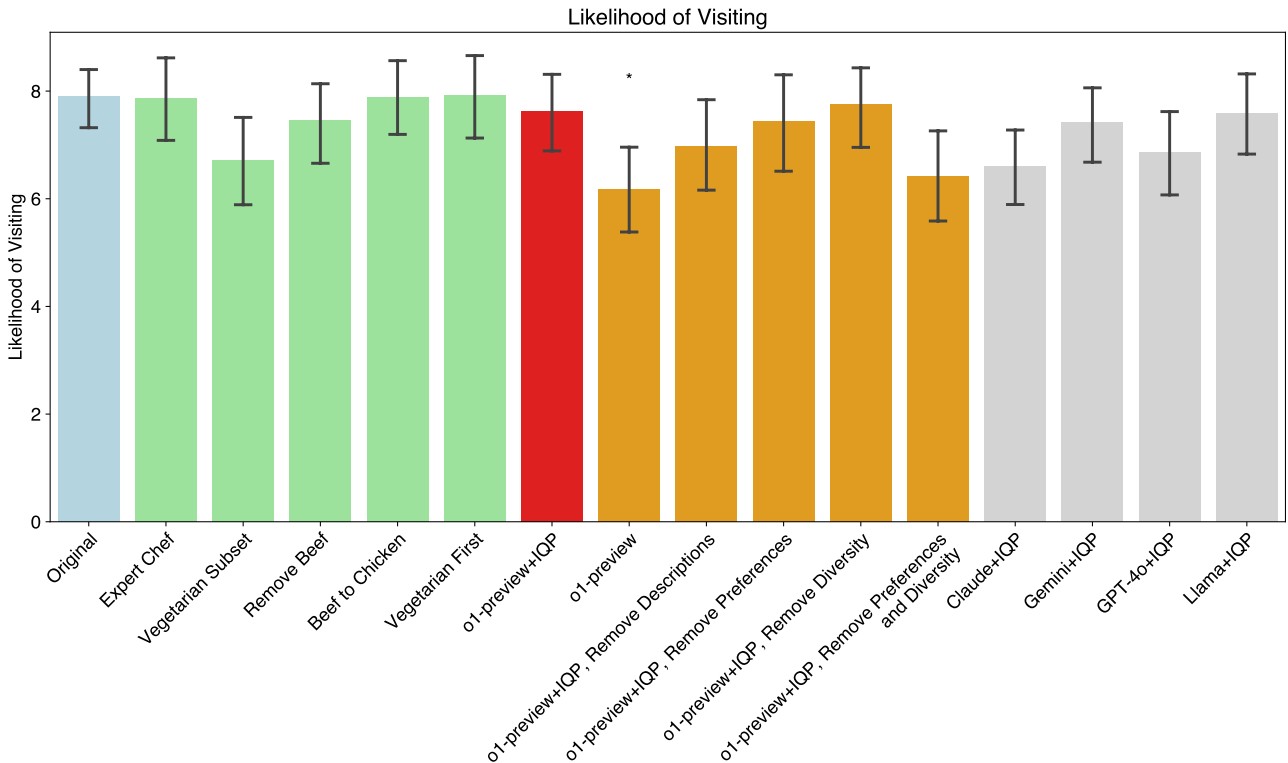

*Figure 23.* Average likelihood of visiting the hypothetical restaurant for all arms, excluding vegan and vegetarian participants. $n = 740$ across arms. Higher is better. Asterisks indicate a statistically significant difference ($t$-test with Bonferroni correction) compared to the original menu. Baselines are in green, ablations are in orange. In "o1-preview+IQP, Remove Descriptions", o1-preview's descriptions are replaced with a simple list of ingredients. In "o1-preview+IQP, Remove Preferences", the preferences component of the objective (the first term in Problem 2) is removed. In "o1-preview+IQP, Remove Diversity", the diversity component of the objective is removed, i.e. $\lambda = 0$ in Problem 2. In "o1-preview+IQP, Remove Preferences and Diversity", both the preference and diversity components of the objective in Problem 2 are removed, i.e. it becomes a feasibility problem.

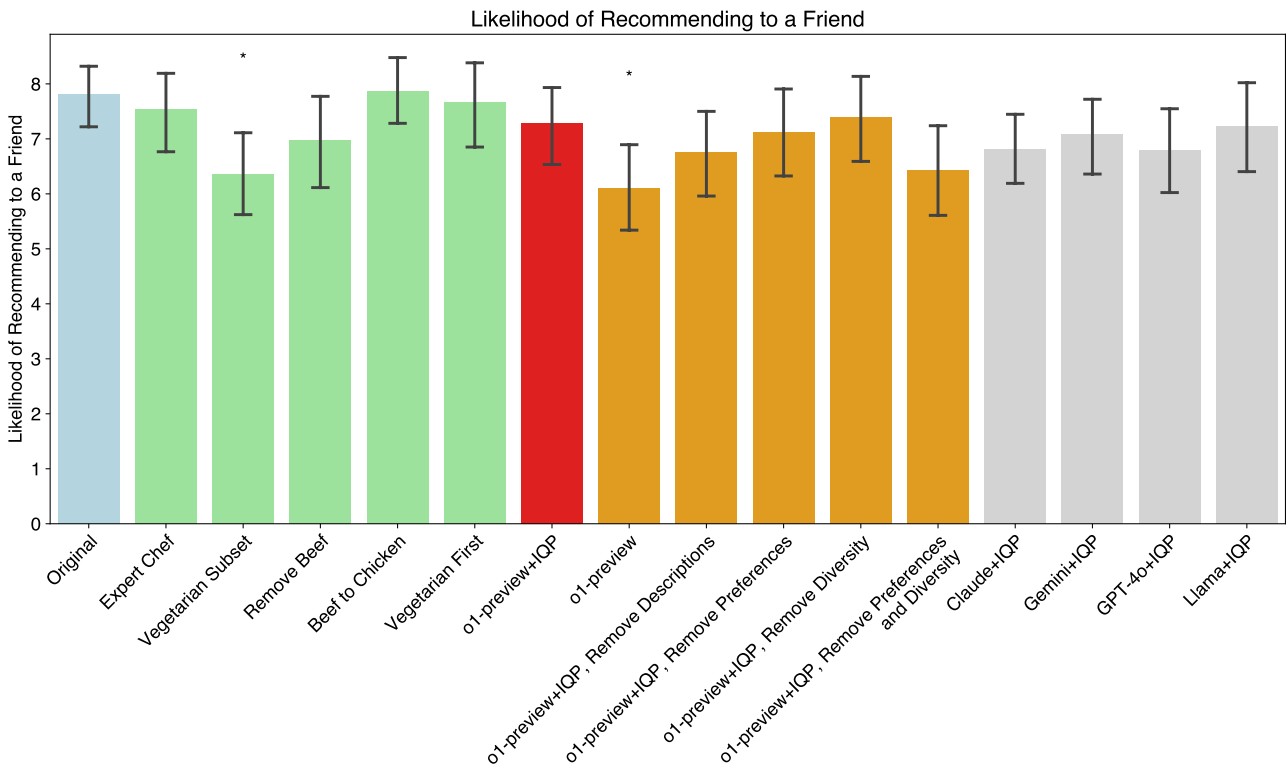

*Figure 24.* Average likelihood of recommending the hypothetical restaurant to a friend for all arms, excluding vegan and vegetarian participants. $n = 740$ across arms. Higher is better. Asterisks indicate a statistically significant difference ($t$-test with Bonferroni correction) compared to the original menu. Baselines are in green, ablations are in orange. In "o1-preview+IQP, Remove Descriptions", o1-preview's descriptions are replaced with a simple list of ingredients. In "o1-preview+IQP, Remove Preferences", the preferences component of the objective (the first term in Problem 2) is removed. In "o1-preview+IQP, Remove Diversity", the diversity component of the objective is removed, i.e. $\lambda = 0$ in Problem 2. In "o1-preview+IQP, Remove Preferences and Diversity", both the preference and diversity components of the objective in Problem 2 are removed, i.e. it becomes a feasibility problem.

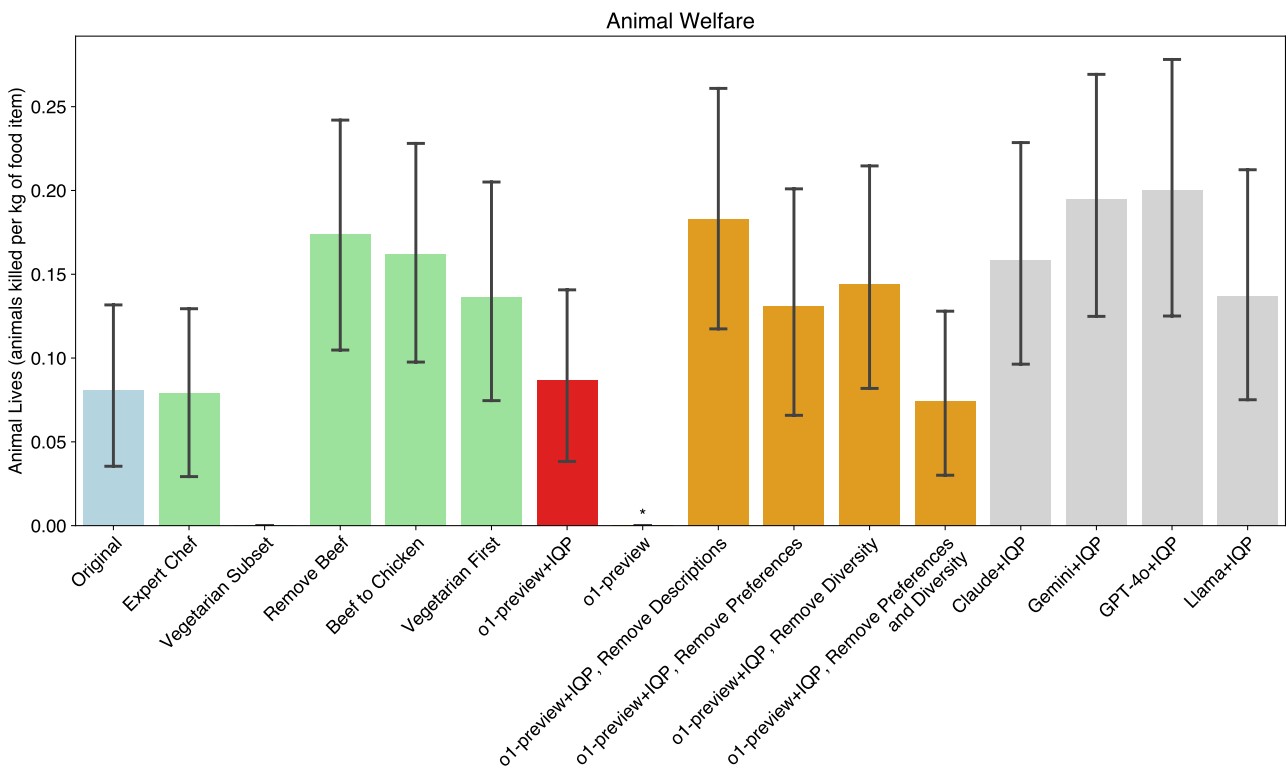

*Figure 25.* Average animal usage for all arms, excluding vegan and vegetarian participants. $n = 740$ across arms. Lower is better. Asterisks indicate a statistically significant difference ($t$-test with Bonferroni correction) compared to the original menu. Baselines are in green, ablations are in orange. In "o1-preview+IQP, Remove Descriptions", o1-preview's descriptions are replaced with a simple list of ingredients. In "o1-preview+IQP, Remove Preferences", the preferences component of the objective (the first term in Problem 2) is removed. In "o1-preview+IQP, Remove Diversity", the diversity component of the objective is removed, i.e. $\lambda = 0$ in Problem 2. In "o1-preview+IQP, Remove Preferences and Diversity", both the preference and diversity components of the objective in Problem 2 are removed, i.e. it becomes a feasibility problem.

## F. Menus

**Original Menu (Banerjee et al., 2023)**

1. Chicken Curry Ramen. Japanese fried chicken & noodles in a delicious curry broth.

2. Pork Ribs. Pork Ribs smothered with Kentucky style BBQ sauce.

3. Beef Bourguignon. 6oz Black Angus beef burger topped with crispy bacon, red wine braised onions, Raclette cheese.

4. Chicken Katsu Curry. Succulent chicken in a crispy Japanese panko crumb with mild curry sauce, pickles and steamed rice.

5. Pepperoni Melt. This one's got our triple cheese blend, tomato sauce and pepperoni.

6. Lamb Meatballs with Spaghetti. Handmade lamb patties grilled and topped with Greek yoghurt, tomato sauce, and onions.

7. Aromatic Duck Rolls. Aromatic roast duck, hoisin sauce, cucumber, spring onion, rolled in rice paper.

8. Slow Cooked Beef in Chianti Sauce. Beef shin and shallots slow cooked in rich Chianti red wine sauce with a cheesy dumpling.

9. Pork Porchetta. Slow-roasted pork belly in herbed red wine sauce, served with roasted new potatoes and broccoli.

10. Cured Salmon Sashimi. Dazzling beetroot cured slices of salmon served with lemon.

11. Beef Brisket and Venison Tagliatelle. A beef, brisket, venison, red wine and beef dripping ragu with tagliatelle, fresh bufala mozzarella and crispy sage.

12. Rustica Chorizo Pizza. Chorizo salami, torn wild boar and pork meatballs, smoked mozzarella and baby sunblush tomatoes.

13. Butterfly Chicken Burger. Two succulent chicken breasts joined by crispy skin, packaged in a Portuguese roll and topped with tomato and lettuce.

14. American Hot Pizza. Pepperoni, mozzarella and tomato, with your choice of hot green, Roquito or jalapeno peppers.

15. Beef Angus Burger. 6oz beef burger with baby gem lettuce and plum tomato in a chargrilled brioche bun with Dijon mayonnaise.

16. Beer Battered Fish with Chips. with frites and pea & mint puree.

17. Wild Boar Polpette. Oven-baked herby wild boar and pork meatballs in a pomodoro sauce with melted smoked mozzarella.

18. Steak. Sliced Black Angus rump steak with red onion chutney, watercress, and Dijon mayonnaise.

19. Canelloni. pasta with béchamel, mozzarella and Gran Milano cheese.

20. Supreme Pizza. Cheese and tomato, onion, mushroom, fresh basil, olive and garlic oil.

21. Goat's Cheese Calzone. Goats cheese, grilled aubergines, roasted peppers, oven roasted tomatoes.

22. Vegan Spaghetti Lentil Ragu. A hearty ragu of green lentils and mixed vegetables in a rich tomato and fennel sauce.

23. Jack Fruit Arrabiata. Vegan jackfruit peppers, spicy harissa, roquito chilis & vegan Mozzarella cheese, finished with basil.

24. Halloumi Sticks with Mayo Avocado Dip. Chunky sticks of grilled halloumi cheese with a chilli jam and mayo avocado dip.

25. Cheese Salad. Smoked cheddar, cheddar and sage stuffing.

26. Falafel with Tahini. Our signature recipe, served with a tahini dip.

27. Four Cheese Margherita. Mozzarella, mascarpone, fontal and grana cheese on a tomato base.

28. Vegan Meatballs. Vegan meatballs in a rich tomato and fennel sauce.

29. Bufala Caprese. Specialty tomatoes and drizzle of roasted garlic in extra virgin olive oil with Buffalo mozzarella.

30. Butternut Squash Salad. Roasted butternut squash with feta cheese, pomegranate seeds, mixed leaf salad and watercress.

31. Panchetta Carbonara. crispy pancetta and asparagus in a velvety sauce made with mascarpone, pecorino and Grana Padano cheese.

32. Lentil Linguine Ragu. Rich Italian lentil ragu with baby spinach, tomatoes, basil & pecorino cheese.

33. Aegean Slaw. Thinly shredded cabbage, carrot and red onion peppers, with an olive oil dressing.

34. Goat's Cheese Salad. Goat's cheese and balsamic onion crostinis on winter baby kale, red pepper, cucumber, plum tomatoes.

35. Harusami Aubergine. Fried slices of aubergine with garlic & ginger sesame soy.

36. Inari Taco. Golden tofu pockets filled with sticky sushi rice, avocado salsa & vegan sriracha mayo.

**Expert Chef**

1. Crispy Miso Eggplant Ramen. Miso marinated eggplant in a crispy panko crust served with a shitake rich curry broth and hand pulled ramen noodles.

2. Pork Ribs. Pork Ribs smothered with Kentucky style BBQ sauce.

3. Beef Bourguignon. 6oz Black Angus beef burger topped with crispy bacon, red wine braised onions, Raclette cheese.

4. Chicken Katsu Curry. Succulent chicken in a crispy Japanese panko crumb with mild curry sauce, pickles and steamed rice.

5. Pepperoni Melt. This one's got our triple cheese blend, tomato sauce and pepperoni.

6. Lamb Meatballs with Spaghetti. Handmade lamb patties grilled and topped with Greek yoghurt, tomato sauce, and onions.

7. Star Anise Black Bean Tempeh Rolls. Black bean tempeh with a lacquered star anise hoisin sauce layered with pickled cucumber, shredded carrots, mint and maple toasted sesame seeds wrapped in rice paper.

8. Basque Celeriac and Gigante Bean Ragout. Celeriac, Gigante beans, nantes carrots, baby leeks, spring garlic, thyme, bay leaves and a lacto fermented tomato sauce creates an orchestra of flavors and textures.

9. Pork Porchetta. Slow-roasted pork belly in herbed red wine sauce, served with roasted new potatoes and broccoli.

10. Beet and Tamari Infused "Tuna". Watermelon infused with Tamari and beet juice, dehydrated and served on a beet horseradish puree with baby watercress, toasted tamari sunflower seeds and a yuzu dressing.

11. Beef Brisket and Venison Tagliatelle. A beef, brisket, venison, red wine and beef dripping ragu with tagliatelle, fresh bufala mozzarella and crispy sage.

12. Rustica Soyrizo Flatbread. Smoked paprika Soyrizo, toasted pinenut basil pesto slathered on extra virgin olive oil herb brushed house flatbread with confit sun dried tomato and garlic , spicy arugula.

13. Butterfly Chicken Burger. Two succulent chicken breasts joined by crispy skin, packaged in a Portuguese roll and topped with tomato and lettuce.

14. American Hot Pizza. Pepperoni, mozzarella and tomato, with your choice of hot green, Roquito or jalapeno peppers.

15. Beef Angus Burger. 6oz beef burger with baby gem lettuce and plum tomato in a chargrilled brioche bun with Dijon mayonnaise.

16. Beer Battered Shio Koji Marinated Enoki Mushroom. Shio koji marinated enoki mushroom, Okinawa sweet potatoes and Kabocha squash tempura with a fermented garlic dipping sauce.

17. Wild Boar Polpette. Oven-baked herby wild boar and pork meatballs in a pomodoro sauce with melted smoked mozzarella.

18. Steak. Sliced Black Angus rump steak with red onion chutney, watercress, and Dijon mayonnaise.

19. Pasta e olio e ceci. cast iron sauteed garlic, chilli flakes and local pressed extra virgin olive oil tossed with al dente spaghetti and finished with pasta water and smashed ceci [garbanzo beans] then garnished with heaps of chopped Italian parsley and ground black pepper and a squeeze of Meyer lemon.

20. Supreme Pizza. Hand thrown sourdough pizza with fermented tomato basil sauce , wood grilled mushrooms and herbed cashew cheese.

21. Goat's Cheese Calzone. Goats cheese, grilled aubergines, roasted peppers, oven roasted tomatoes.

22. Vegan Spaghetti Lentil Ragu. A hearty ragu of green lentils and mixed vegetables in a rich tomato and fennel sauce.

23. Jack Fruit Arrabiata. Vegan jackfruit peppers, spicy harissa, roquito chilis & vegan Mozzarella cheese, finished with basil.

24. Halloumi Sticks with Mayo Avocado Dip. Chunky sticks of grilled halloumi cheese with a chilli jam and mayo avocado dip.

25. Cheese Salad. Smoked cheddar, cheddar and sage stuffing.

26. Falafel with Tahini. Our signature recipe, served with a tahini dip.

27. Four Cheese Margherita. Mozzarella, mascarpone, fontal and grana cheese on a tomato base.

28. Vegan Meatballs. Vegan meatballs in a rich tomato and fennel sauce.

29. Bufala Caprese. Specialty tomatoes and drizzle of roasted garlic in extra virgin olive oil with Buffalo mozzarella.

30. Butternut Squash Salad. Roasted pomegranate molasses rubbed butternut squash tossed with pomegranate arils, toasted pumpkin seeds, herbed macadamia nut cheese with mushroom flat bread crumble.

31. Panchetta Carbonara. crispy pancetta and asparagus in a velvety sauce made with mascarpone, pecorino and Grana Padano cheese.

32. Pasta e lenticchie. Beluga lentil ragout slow cooked with Italian plum tomatoes ,tossed with bronze cut linguine and finished with smoky Shiitake bacon.

33. Aegean Slaw. Thinly shredded cabbage, carrot and red onion peppers, with an olive oil dressing.

34. Goat's Cheese Salad. Goat's cheese and balsamic onion crostinis on winter baby kale, red pepper, cucumber, plum tomatoes.

35. Harusami Aubergine. Fried slices of aubergine with garlic & ginger sesame soy.

36. Inari Taco. Golden tofu pockets filled with sticky sushi rice, avocado salsa & vegan sriracha mayo.

## Remove Beef

1. Chicken Curry Ramen. Japanese fried chicken & noodles in a delicious curry broth.

2. Pork Ribs. Pork Ribs smothered with Kentucky style BBQ sauce.

3. Chicken Katsu Curry. Succulent chicken in a crispy Japanese panko crumb with mild curry sauce, pickles and steamed rice.

4. Pepperoni Melt. This one's got our triple cheese blend, tomato sauce and pepperoni.

5. Lamb Meatballs with Spaghetti. Handmade lamb patties grilled and topped with Greek yoghurt, tomato sauce, and onions.

6. Aromatic Duck Rolls. Aromatic roast duck, hoisin sauce, cucumber, spring onion, rolled in rice paper.

7. Pork Porchetta. Slow-roasted pork belly in herbed red wine sauce, served with roasted new potatoes and broccoli.

8. Cured Salmon Sashimi. Dazzling beetroot cured slices of salmon served with lemon.

9. Rustica Chorizo Pizza. Chorizo salami, torn wild boar and pork meatballs, smoked mozzarella and baby sunblush tomatoes.

10. Butterfly Chicken Burger. Two succulent chicken breasts joined by crispy skin, packaged in a Portuguese roll and topped with tomato and lettuce.

11. American Hot Pizza. Pepperoni, mozzarella and tomato, with your choice of hot green, Roquito or jalapeno peppers.

12. Beer Battered Fish with Chips. with frites and pea & mint puree.

13. Wild Boar Polpette. Oven-baked herby wild boar and pork meatballs in a pomodoro sauce with melted smoked mozzarella.

14. Canelloni. pasta with béchamel, mozzarella and Gran Milano cheese.

15. Supreme Pizza. Cheese and tomato, onion, mushroom, fresh basil, olive and garlic oil.

16. Goat's Cheese Calzone. Goats cheese, grilled aubergines, roasted peppers, oven roasted tomatoes.

17. Vegan Spaghetti Lentil Ragu. A hearty ragu of green lentils and mixed vegetables in a rich tomato and fennel sauce.

18. Jack Fruit Arrabiata. Vegan jackfruit peppers, spicy harissa, roquito chilis & vegan Mozzarella cheese, finished with basil.

19. Halloumi Sticks with Mayo Avocado Dip. Chunky sticks of grilled halloumi cheese with a chilli jam and mayo avocado dip.

20. Cheese Salad. Smoked cheddar, cheddar and sage stuffing.

21. Falafel with Tahini. Our signature recipe, served with a tahini dip.

22. Four Cheese Margherita. Mozzarella, mascarpone, fontal and grana cheese on a tomato base.

23. Vegan Meatballs. Vegan meatballs in a rich tomato and fennel sauce.

24. Bufala Caprese. Specialty tomatoes and drizzle of roasted garlic in extra virgin olive oil with Buffalo mozzarella.

25. Butternut Squash Salad. Roasted butternut squash with feta cheese, pomegranate seeds, mixed leaf salad and watercress.

26. Panchetta Carbonara. crispy pancetta and asparagus in a velvety sauce made with mascarpone, pecorino and Grana Padano cheese.

27. Lentil Linguine Ragu. Rich Italian lentil ragu with baby spinach, tomatoes, basil & pecorino cheese.

28. Aegean Slaw. Thinly shredded cabbage, carrot and red onion peppers, with an olive oil dressing.

29. Goat's Cheese Salad. Goat's cheese and balsamic onion crostinis on winter baby kale, red pepper, cucumber, plum tomatoes.

30. Harusami Aubergine. Fried slices of aubergine with garlic & ginger sesame soy.

31. Inari Taco. Golden tofu pockets filled with sticky sushi rice, avocado salsa & vegan sriracha mayo.

**Vegetarian**

1. Canelloni. pasta with béchamel, mozzarella and Gran Milano cheese.

2. Supreme Pizza. Cheese and tomato, onion, mushroom, fresh basil, olive and garlic oil.

3. Goat's Cheese Calzone. Goats cheese, grilled aubergines, roasted peppers, oven roasted tomatoes.

4. Vegan Spaghetti Lentil Ragu. A hearty ragu of green lentils and mixed vegetables in a rich tomato and fennel sauce.

5. Jack Fruit Arrabiata. Vegan jackfruit peppers, spicy harissa, roquito chilis & vegan Mozzarella cheese, finished with basil.

6. Halloumi Sticks with Mayo Avocado Dip. Chunky sticks of grilled halloumi cheese with a chilli jam and mayo avocado dip.

7. Cheese Salad. Smoked cheddar, cheddar and sage stuffing.

8. Falafel with Tahini. Our signature recipe, served with a tahini dip.

9. Four Cheese Margherita. Mozzarella, mascarpone, fontal and grana cheese on a tomato base.

10. Vegan Meatballs. Vegan meatballs in a rich tomato and fennel sauce.

11. Bufala Caprese. Specialty tomatoes and drizzle of roasted garlic in extra virgin olive oil with Buffalo mozzarella.

12. Butternut Squash Salad. Roasted butternut squash with feta cheese, pomegranate seeds, mixed leaf salad and watercress.

13. Lentil Linguine Ragu. Rich Italian lentil ragu with baby spinach, tomatoes, basil & pecorino cheese.

14. Aegean Slaw. Thinly shredded cabbage, carrot and red onion peppers, with an olive oil dressing.

15. Goat's Cheese Salad. Goat's cheese and balsamic onion crostinis on winter baby kale, red pepper, cucumber, plum tomatoes.

16. Harusami Aubergine. Fried slices of aubergine with garlic & ginger sesame soy.

17. Inari Taco. Golden tofu pockets filled with sticky sushi rice, avocado salsa & vegan sriracha mayo.

**Beef to Chicken**

1. Chicken Curry Ramen. Japanese fried chicken & noodles in a delicious curry broth.

2. Pork Ribs. Pork Ribs smothered with Kentucky style BBQ sauce.

3. Chicken Bourguignon. 6oz chicken burger topped with crispy bacon, red wine braised onions, Raclette cheese.

4. Chicken Katsu Curry. Succulent chicken in a crispy Japanese panko crumb with mild curry sauce, pickles and steamed rice.

5. Pepperoni Melt. This one's got our triple cheese blend, tomato sauce and pepperoni.

6. Lamb Meatballs with Spaghetti. Handmade lamb patties grilled and topped with Greek yoghurt, tomato sauce, and onions.

7. Aromatic Duck Rolls. Aromatic roast duck, hoisin sauce, cucumber, spring onion, rolled in rice paper.

8. Slow Cooked Chicken in Chianti Sauce. Chicken and shallots slow cooked in rich Chianti red wine sauce with a cheesy dumpling.

9. Pork Porchetta. Slow-roasted pork belly in herbed red wine sauce, served with roasted new potatoes and broccoli.

10. Cured Salmon Sashimi. Dazzling beetroot cured slices of salmon served with lemon.

11. Chicken and Venison Tagliatelle. A chicken breast, venison, red wine and beef dripping ragu with tagliatelle, fresh bufala mozzarella and crispy sage.

12. Rustica Chorizo Pizza. Chorizo salami, torn wild boar and pork meatballs, smoked mozzarella and baby sunblush tomatoes.

13. Butterfly Chicken Burger. Two succulent chicken breasts joined by crispy skin, packaged in a Portuguese roll and topped with tomato and lettuce.

14. American Hot Pizza. Pepperoni, mozzarella and tomato, with your choice of hot green, Roquito or jalapeno peppers.

15. Chicken Burger. 6oz chicken burger with baby gem lettuce and plum tomato in a chargrilled brioche bun with Dijon mayonnaise.

16. Beer Battered Fish with Chips. with frites and pea & mint puree.

17. Wild Boar Polpette. Oven-baked herby wild boar and pork meatballs in a pomodoro sauce with melted smoked mozzarella.

18. Baked Chicken. Baked chicken with red onion chutney, watercress, and Dijon mayonnaise.

19. Canelloni. pasta with béchamel, mozzarella and Gran Milano cheese.

20. Supreme Pizza. Cheese and tomato, onion, mushroom, fresh basil, olive and garlic oil.

21. Goat's Cheese Calzone. Goats cheese, grilled aubergines, roasted peppers, oven roasted tomatoes.

22. Vegan Spaghetti Lentil Ragu. A hearty ragu of green lentils and mixed vegetables in a rich tomato and fennel sauce.

23. Jack Fruit Arrabiata. Vegan jackfruit peppers, spicy harissa, roquito chilis & vegan Mozzarella cheese, finished with basil.

24. Halloumi Sticks with Mayo Avocado Dip. Chunky sticks of grilled halloumi cheese with a chilli jam and mayo avocado dip.

25. Cheese Salad. Smoked cheddar, cheddar and sage stuffing.

26. Falafel with Tahini. Our signature recipe, served with a tahini dip.

27. Four Cheese Margherita. Mozzarella, mascarpone, fontal and grana cheese on a tomato base.

28. Vegan Meatballs. Vegan meatballs in a rich tomato and fennel sauce.

29. Bufala Caprese. Specialty tomatoes and drizzle of roasted garlic in extra virgin olive oil with Buffalo mozzarella.

30. Butternut Squash Salad. Roasted butternut squash with feta cheese, pomegranate seeds, mixed leaf salad and watercress.

31. Panchetta Carbonara. crispy pancetta and asparagus in a velvety sauce made with mascarpone, pecorino and Grana Padano cheese.

32. Lentil Linguine Ragu. Rich Italian lentil ragu with baby spinach, tomatoes, basil & pecorino cheese.

33. Aegean Slaw. Thinly shredded cabbage, carrot and red onion peppers, with an olive oil dressing.

34. Goat's Cheese Salad. Goat's cheese and balsamic onion crostinis on winter baby kale, red pepper, cucumber, plum tomatoes.

35. Harusami Aubergine. Fried slices of aubergine with garlic & ginger sesame soy.

36. Inari Taco. Golden tofu pockets filled with sticky sushi rice, avocado salsa & vegan sriracha mayo.

**Vegetarian First**

1. Canelloni. Pasta with béchamel, mozzarella and Gran Milano cheese.

2. Supreme Pizza. Cheese and tomato, onion, mushroom, fresh basil, olive and garlic oil.

3. Goat's Cheese Calzone. Goats cheese, grilled aubergines, roasted peppers, oven roasted tomatoes.

4. Vegan Spaghetti Lentil Ragu. A hearty ragu of green lentils and mixed vegetables in a rich tomato and fennel sauce.

5. Jack Fruit Arrabiata. Vegan jackfruit peppers, spicy harissa, roquito chilis & vegan Mozzarella cheese, finished with basil.

6. Halloumi Sticks with Mayo Avocado Dip. Chunky sticks of grilled halloumi cheese with a chilli jam and mayo avocado dip.

7. Cheese Salad. Smoked cheddar, cheddar and sage stuffing.

8. Falafel with Tahini. Our signature recipe, served with a tahini dip.

9. Four Cheese Margherita. Mozzarella, mascarpone, fontal and grana cheese on a tomato base.

10. Vegan Meatballs. Vegan meatballs in a rich tomato and fennel sauce.

11. Bufala Caprese. Specialty tomatoes and drizzle of roasted garlic in extra virgin olive oil with Buffalo mozzarella.

12. Butternut Squash Salad. Roasted butternut squash with feta cheese, pomegranate seeds, mixed leaf salad and watercress.

13. Lentil Linguine Ragu. Rich Italian lentil ragu with baby spinach, tomatoes, basil & pecorino cheese.

14. Aegean Slaw. Thinly shredded cabbage, carrot and red onion peppers, with an olive oil dressing.

15. Goat's Cheese Salad. Goat's cheese and balsamic onion crostinis on winter baby kale, red pepper, cucumber, plum tomatoes.

16. Harusami Aubergine. Fried slices of aubergine with garlic & ginger sesame soy.

17. Inari Taco. Golden tofu pockets filled with sticky sushi rice, avocado salsa & vegan sriracha mayo.

18. Chicken Curry Ramen. Japanese fried chicken & noodles in a delicious curry broth.

19. Pork Ribs. Pork Ribs smothered with Kentucky style BBQ sauce.

20. Beef Bourguignon. 6oz Black Angus beef burger topped with crispy bacon, red wine braised onions, Raclette cheese.

21. Chicken Katsu Curry. Succulent chicken in a crispy Japanese panko crumb with mild curry sauce, pickles and steamed rice.

22. Pepperoni Melt. This one's got our triple cheese blend, tomato sauce and pepperoni.

23. Lamb Meatballs with Spaghetti. Handmade lamb patties grilled and topped with Greek yoghurt, tomato sauce, and onions.

24. Aromatic Duck Rolls. Aromatic roast duck, hoisin sauce, cucumber, spring onion, rolled in rice paper.

25. Slow Cooked Beef in Chianti Sauce. Beef shin and shallots slow cooked in rich Chianti red wine sauce with a cheesy dumpling.

26. Pork Porchetta. Slow-roasted pork belly in herbed red wine sauce, served with roasted new potatoes and broccoli.

27. Cured Salmon Sashimi. Dazzling beetroot cured slices of salmon served with lemon.

28. Beef Brisket and Venison Tagliatelle. A beef, brisket, venison, red wine and beef dripping ragu with tagliatelle, fresh bufala mozzarella and crispy sage.

29. Rustica Chorizo Pizza. Chorizo salami, torn wild boar and pork meatballs, smoked mozzarella and baby sunblush tomatoes.

30. Butterfly Chicken Burger. Two succulent chicken breasts joined by crispy skin, packaged in a Portuguese roll and topped with tomato and lettuce.

31. American Hot Pizza. Pepperoni, mozzarella and tomato, with your choice of hot green, Roquito or jalapeno peppers.

32. Beef Angus Burger. 6oz beef burger with baby gem lettuce and plum tomato in a chargrilled brioche bun with Dijon mayonnaise.

33. Beer Battered Fish with Chips. With frites and pea & mint puree.

34. Wild Boar Polpette. Oven-baked herby wild boar and pork meatballs in a pomodoro sauce with melted smoked mozzarella.

35. Steak. Sliced Black Angus rump steak with red onion chutney, watercress, and Dijon mayonnaise.

36. Panchetta Carbonara. Crispy pancetta and asparagus in a velvety sauce made with mascarpone, pecorino and Grana Padano cheese.

**o1-preview+IQP**

1. Creamy Mushroom Tagliatelle. An indulgent pasta dish featuring sautéed mushrooms and baby spinach in a creamy mascarpone sauce, tossed with tagliatelle and seasoned with garlic oil and fresh basil.

2. Vegetable Delight Pizza. A delicious crispy pizza topped with mozzarella, tomato sauce, and a medley of grilled vegetables, finished with fresh basil.

3. Three Cheese Omelette. A fluffy omelette loaded with smoked cheddar, mozzarella, grana cheese, and sautéed red onions.

4. Eggplant Parmesan. Layers of tender aubergine baked with rich tomato sauce, melted mozzarella, and Grana Padano cheese, garnished with fresh basil.

5. Mushroom and Goat's Cheese Omelette. A fluffy omelette filled with sautéed mushrooms and creamy goat's cheese, infused with garlic oil and fresh basil.

6. Chickpea Curry with Rice. A flavorful and hearty chickpea curry served with steamed rice and accompanied by tangy pickles.

7. Spinach and Feta Stuffed Mushrooms. Large mushrooms stuffed with sautéed baby spinach and creamy feta cheese, drizzled with garlic oil and baked to perfection.

8. Lentil Veggie Burger. A hearty lentil-based veggie burger served on a toasted brioche bun with fresh lettuce, tomato, and tangy Dijon mayonnaise.

9. Falafel Salad. Our signature falafel served over a fresh mixed salad, with crunchy cucumbers, juicy tomatoes, red onions, and a creamy tahini dressing.

10. Egg Shakshuka. Poached eggs simmered in a spiced tomato sauce with peppers and onions, garnished with fresh basil.

11. Vegetable and Tofu Stir-Fry. A vibrant stir-fry of crispy tofu and fresh vegetables tossed with noodles in a savory garlic and ginger sesame soy sauce.

12. Tofu Katsu Curry. Succulent tofu coated in crispy panko crumbs, served with mild curry sauce, tangy pickles, and steamed rice.

13. Lentil Stuffed Peppers. Roasted bell peppers stuffed with hearty lentils, tomato sauce, and fresh baby spinach, topped with creamy goat's cheese.

14. Butternut Squash and Feta Salad. Sweet roasted butternut squash and tangy feta cheese on a bed of fresh mixed greens, sprinkled with pomegranate seeds and watercress.

15. Vegan Meatball Sub. Hearty vegan meatballs simmered in pomodoro sauce, topped with melted smoked mozzarella, served in a toasted Portuguese roll.

16. Chickpea and Spinach Curry. A nourishing curry of chickpeas and baby spinach simmered in a mild curry sauce, served with steamed rice.

17. Tofu Curry Ramen. Japanese-style ramen with fried tofu and noodles in a delicious curry broth, topped with pak choi and pickled onions.

18. Mushroom and Lentil Bolognese. Hearty mushrooms and lentils cooked in a rich tomato sauce, served over tagliatelle pasta and garnished with fresh basil.

19. Supreme Pizza. Cheese and tomato, onion, mushroom, fresh basil, olive and garlic oil.

20. Chicken Curry Ramen. Japanese fried chicken & noodles in a delicious curry broth.

21. Pork Ribs. Pork Ribs smothered with Kentucky style BBQ sauce.

22. Butterfly Chicken Burger. Two succulent chicken breasts joined by crispy skin, packaged in a Portuguese roll and topped with tomato and lettuce.

23. Four Cheese Margherita. Mozzarella, mascarpone, fontal and grana cheese on a tomato base.

24. Panchetta Carbonara. crispy pancetta and asparagus in a velvety sauce made with mascarpone, pecorino and Grana Padano cheese.

25. Chicken Katsu Curry. Succulent chicken in a crispy Japanese panko crumb with mild curry sauce, pickles and steamed rice.

26. Rustica Chorizo Pizza. Chorizo salami, torn wild boar and pork meatballs, smoked mozzarella and baby sunblush tomatoes.

27. Beer Battered Fish with Chips. with frites and pea & mint puree.

28. Pork Porchetta. Slow-roasted pork belly in herbed red wine sauce, served with roasted new potatoes and broccoli.

29. Butternut Squash Salad. Roasted butternut squash with feta cheese, pomegranate seeds, mixed leaf salad and watercress.

30. Aegean Slaw. Thinly shredded cabbage, carrot and red onion peppers, with an olive oil dressing.

31. Cured Salmon Sashimi. Dazzling beetroot cured slices of salmon served with lemon.

32. Falafel with Tahini. Our signature recipe, served with a tahini dip.

33. Lentil Linguine Ragu. Rich Italian lentil ragu with baby spinach, tomatoes, basil & pecorino cheese.

34. Aromatic Duck Rolls. Aromatic roast duck, hoisin sauce, cucumber, spring onion, rolled in rice paper.

35. Vegan Spaghetti Lentil Ragu. A hearty ragu of green lentils and mixed vegetables in a rich tomato and fennel sauce.

36. Harusami Aubergine. Fried slices of aubergine with garlic & ginger sesame soy.

**GPT-4o+IQP**

1. Falafel Wrap. A flavorful and nutritious wrap perfect for any time of day.

2. Lentil Meatball Spaghetti. A comforting combination of lentil meatballs and rich tomato sauce with pasta.

3. Eggs Florentine Pizza. A delightful pizza featuring fresh eggs and vibrant toppings.

4. Spinach and Asparagus Carbonara. A creamy and delicious pasta loaded with greens.

5. Butternut Squash and Feta Cannelloni. Rich, creamy, and packed with flavor, perfect for vegetarians.

6. Roasted Beetroot and Goat's Cheese Salad. A vibrant salad with the earthy sweetness of beetroot and creamy goat's cheese.

7. BBQ Jackfruit Ribs. Tender jackfruit smothered in a tangy BBQ sauce, served with sides.

8. Eggplant Ragu Tagliatelle. A hearty and delicious pasta with a rich eggplant ragu.

9. Vegan Stuffed Bell Peppers. Hearty stuffed peppers, packed with delicious and healthy fillings.

10. Spicy Lentil and Jackfruit Tacos. A fiery taco loaded with protein and bold flavors.

11. Mushroom Bourguignon. A rich and savory dish, perfect for a comforting meal.

12. Tofu Porchetta. Herb-crusted tofu and flavorful sides for a vegan take on porchetta.

13. Chickpea Katsu Burger. A satisfying burger with a crunchy chickpea patty and fresh toppings.

14. Panko Crusted Tofu Fingers. Crunchy tofu sticks with a deliciously creamy dip.

15. Tofu and Veggie Stir-fry. A nutritious stir-fry with tofu and a mix of fresh vegetables.

16. Pork Ribs. Pork Ribs smothered with Kentucky style BBQ sauce.

17. Beer Battered Fish with Chips. with frites and pea & mint puree.

18. Supreme Pizza. Cheese and tomato, onion, mushroom, fresh basil, olive and garlic oil.

19. Four Cheese Margherita. Mozzarella, mascarpone, fontal and grana cheese on a tomato base.

20. Panchetta Carbonara. crispy pancetta and asparagus in a velvety sauce made with mascarpone, pecorino and Grana Padano cheese.

21. Chicken Curry Ramen. Japanese fried chicken & noodles in a delicious curry broth.

22. Pork Porchetta. Slow-roasted pork belly in herbed red wine sauce, served with roasted new potatoes and broccoli.

23. Rustica Chorizo Pizza. Chorizo salami, torn wild boar and pork meatballs, smoked mozzarella and baby sunblush tomatoes.

24. Goat's Cheese Salad. Goat's cheese and balsamic onion crostinis on winter baby kale, red pepper, cucumber, plum tomatoes.

25. Chicken Katsu Curry. Succulent chicken in a crispy Japanese panko crumb with mild curry sauce, pickles and steamed rice.

26. Aromatic Duck Rolls. Aromatic roast duck, hoisin sauce, cucumber, spring onion, rolled in rice paper.

27. Wild Boar Polpette. Oven-baked herby wild boar and pork meatballs in a pomodoro sauce with melted smoked mozzarella.

28. Aegean Slaw. Thinly shredded cabbage, carrot and red onion peppers, with an olive oil dressing.

29. Cured Salmon Sashimi. Dazzling beetroot cured slices of salmon served with lemon.

30. Jack Fruit Arrabiata. Vegan jackfruit peppers, spicy harissa, roquito chilis & vegan Mozzarella cheese, finished with basil.

31. Butternut Squash Salad. Roasted butternut squash with feta cheese, pomegranate seeds, mixed leaf salad and watercress.

32. Lentil Linguine Ragu. Rich Italian lentil ragu with baby spinach, tomatoes, basil & pecorino cheese.

33. Vegan Spaghetti Lentil Ragu. A hearty ragu of green lentils and mixed vegetables in a rich tomato and fennel sauce.

34. Vegan Meatballs. Vegan meatballs in a rich tomato and fennel sauce.

35. Falafel with Tahini. Our signature recipe, served with a tahini dip.

36. Harusami Aubergine. Fried slices of aubergine with garlic & ginger sesame soy.

## Claude 3.5 Sonnet+IQP

1. Cheese Melt Pizza. A gooey, cheesy pizza topped with savory mushrooms and tangy tomato sauce.

2. Rustica Vegetable Pizza. A hearty vegetarian pizza loaded with savory mushrooms, smoky cheese, and sweet tomatoes.

3. Mushroom Tagliatelle. Ribbon pasta tossed with a rich mushroom and red wine sauce, topped with creamy mozzarella and crispy sage leaves.

4. Vegetable Hot Pizza. A spicy vegetarian pizza featuring a medley of peppers and savory mushrooms.

5. Mushroom Bourguignon. A hearty vegan twist on the classic French dish, featuring meaty mushrooms and savory red wine sauce.

6. Butternut Squash Carbonara. Creamy pasta featuring roasted butternut squash and crisp asparagus in a rich cheese sauce.

7. Tofu Katsu Curry. Crispy breaded tofu cutlet served with a mild and aromatic curry sauce.

8. Mushroom Rolls. Savory mushrooms and crisp vegetables wrapped in delicate rice paper with a sweet hoisin glaze.

9. Slow Cooked Lentils in Chianti Sauce. Hearty lentils slow-simmered in a rich red wine sauce, topped with a fluffy cheese dumpling.

10. Wild Mushroom Polpette. Oven-baked mushroom "meatballs" in a rich tomato sauce with melted smoky cheese.

11. Lentil Linguine Ragu. Al dente linguine tossed with a robust lentil ragu and finished with fresh herbs and cheese.

12. Lentil Meatballs with Spaghetti. Tender lentil meatballs served over spaghetti with a rich tomato sauce and creamy yoghurt.

13. Lentil Ragu Spaghetti. A hearty lentil and vegetable ragu served over tender spaghetti with aromatic fennel.

14. Jackfruit Arrabiata. Spicy jackfruit "pulled pork" with peppers and vegan cheese in a fiery arrabiata sauce.

15. Tofu Fish and Chips. Crispy battered tofu fillets served with golden fries and a refreshing pea puree.

16. Pork Ribs. Pork Ribs smothered with Kentucky style BBQ sauce.

17. Rustica Chorizo Pizza. Chorizo salami, torn wild boar and pork meatballs, smoked mozzarella and baby sunblush tomatoes.

18. Four Cheese Margherita. Mozzarella, mascarpone, fontal and grana cheese on a tomato base.

19. Panchetta Carbonara. crispy pancetta and asparagus in a velvety sauce made with mascarpone, pecorino and Grana Padano cheese.

20. Chicken Curry Ramen. Japanese fried chicken & noodles in a delicious curry broth.

21. Chicken Katsu Curry. Succulent chicken in a crispy Japanese panko crumb with mild curry sauce, pickles and steamed rice.

22. Pork Porchetta. Slow-roasted pork belly in herbed red wine sauce, served with roasted new potatoes and broccoli.

23. Beer Battered Fish with Chips. with frites and pea & mint puree.

24. Goat's Cheese Calzone. Goats cheese, grilled aubergines, roasted peppers, oven roasted tomatoes.

25. Aromatic Duck Rolls. Aromatic roast duck, hoisin sauce, cucumber, spring onion, rolled in rice paper.

26. Butterfly Chicken Burger. Two succulent chicken breasts joined by crispy skin, packaged in a Portuguese roll and topped with tomato and lettuce.

27. Halloumi Sticks with Mayo Avocado Dip. Chunky sticks of grilled halloumi cheese with a chilli jam and mayo avocado dip.

28. Falafel with Tahini. Our signature recipe, served with a tahini dip.

29. Lentil Linguine Ragu. Rich Italian lentil ragu with baby spinach, tomatoes, basil & pecorino cheese.

30. Wild Boar Polpette. Oven-baked herby wild boar and pork meatballs in a pomodoro sauce with melted smoked mozzarella.

31. Vegan Spaghetti Lentil Ragu. A hearty ragu of green lentils and mixed vegetables in a rich tomato and fennel sauce.

32. Butternut Squash Salad. Roasted butternut squash with feta cheese, pomegranate seeds, mixed leaf salad and watercress.

33. Aegean Slaw. Thinly shredded cabbage, carrot and red onion peppers, with an olive oil dressing.

34. Jack Fruit Arrabiata. Vegan jackfruit peppers, spicy harissa, roquito chilis & vegan Mozzarella cheese, finished with basil.

35. Vegan Meatballs. Vegan meatballs in a rich tomato and fennel sauce.

36. Harusami Aubergine. Fried slices of aubergine with garlic & ginger sesame soy.

**Gemini 1.5 Pro+IQP**

1. Pepperoni Melt with Tofu. Enjoy this twist on our classic Pepperoni Melt by swapping meat for tofu.

2. Four Cheese and Mushroom Pizza. A classic pizza with an added layer of flavor.

3. Chickpea and Spinach Curry. Warm up with this hearty and flavorful chickpea and spinach curry.

4. Rustica Veggie Pizza. This meat-free pizza is packed with flavor and will satisfy any pizza lover.

5. Veggie Supreme Pizza. A classic pizza, loaded with fresh veggies and bursting with flavor.

6. Spinach and Mushroom Carbonara. This creamy and flavorful pasta dish is sure to please everyone at the table.

7. Mushroom Bourguignon. Enjoy the rich flavors of our classic Beef Bourguignon with a meat-free twist.

8. Spicy Jackfruit Pizza. This spicy and delicious pizza is sure to tantalize your taste buds.

9. Mushroom and Spinach Meatballs. These flavorful and healthy meatballs are a great alternative to the classic.

10. Tofu Curry Ramen. A vegan alternative to our classic Chicken Curry Ramen, with crispy fried tofu.

11. Tofu Katsu Curry. All the goodness of our Chicken Katsu Curry in a satisfying plant-based alternative.

12. Mushroom and Lentil Ragu with Tagliatelle. A hearty and satisfying vegetarian take on our Beef Brisket and Venison Tagliatelle.

13. Mushroom Calzone. Savory and satisfying, our mushroom calzone is packed full of fresh ingredients.

14. Butternut Squash and Chickpea Salad . Enjoy all the current flavors of Fall in this colorful salad.

15. Lentil Meatballs with Spaghetti. Enjoy this lighter version of our Lamb Meatballs with Spaghetti.

16. Lentil Bolognese. This lentil bolognese is a hearty and flavorful alternative to our classic spaghetti dish.

17. Chicken Curry Ramen. Japanese fried chicken & noodles in a delicious curry broth.

18. Pork Ribs. Pork Ribs smothered with Kentucky style BBQ sauce.

19. Chicken Katsu Curry. Succulent chicken in a crispy Japanese panko crumb with mild curry sauce, pickles and steamed rice.

20. Beer Battered Fish with Chips. with frites and pea & mint puree.

21. Halloumi Sticks with Mayo Avocado Dip. Chunky sticks of grilled halloumi cheese with a chilli jam and mayo avocado dip.

22. Four Cheese Margherita. Mozzarella, mascarpone, fontal and grana cheese on a tomato base.

23. Panchetta Carbonara. crispy pancetta and asparagus in a velvety sauce made with mascarpone, pecorino and Grana Padano cheese.

24. Rustica Chorizo Pizza. Chorizo salami, torn wild boar and pork meatballs, smoked mozzarella and baby sunblush tomatoes.

25. Supreme Pizza. Cheese and tomato, onion, mushroom, fresh basil, olive and garlic oil.

26. Goat's Cheese Calzone. Goats cheese, grilled aubergines, roasted peppers, oven roasted tomatoes.

27. Aromatic Duck Rolls. Aromatic roast duck, hoisin sauce, cucumber, spring onion, rolled in rice paper.

28. Butterfly Chicken Burger. Two succulent chicken breasts joined by crispy skin, packaged in a Portuguese roll and topped with tomato and lettuce.

29. Canelloni. pasta with béchamel, mozzarella and Gran Milano cheese.

30. Jack Fruit Arrabiata. Vegan jackfruit peppers, spicy harissa, roquito chilis & vegan Mozzarella cheese, finished with basil.

31. Falafel with Tahini. Our signature recipe, served with a tahini dip.

32. Aegean Slaw. Thinly shredded cabbage, carrot and red onion peppers, with an olive oil dressing.

33. Pork Porchetta. Slow-roasted pork belly in herbed red wine sauce, served with roasted new potatoes and broccoli.

34. Butternut Squash Salad. Roasted butternut squash with feta cheese, pomegranate seeds, mixed leaf salad and watercress.

35. Lentil Linguine Ragu. Rich Italian lentil ragu with baby spinach, tomatoes, basil & pecorino cheese.

36. Vegan Spaghetti Lentil Ragu. A hearty ragu of green lentils and mixed vegetables in a rich tomato and fennel sauce.

**Llama 3.1 70b Instruct+IQP**

1. Grilled Aubergine and Goat's Cheese Salad. A flavorful and refreshing salad perfect for any time of the year.

2. Stuffed Bell Peppers. A colorful and flavorful vegetarian main course.

3. Grilled Halloumi and Vegetable Skewers. A simple yet impressive vegetarian appetizer or snack.

4. Grilled Portobello Mushroom and Goat's Cheese Salad. A flavorful and refreshing salad perfect for any time of the year.

5. Roasted Butternut Squash Risotto. A creamy and comforting dish featuring roasted butternut squash.

6. Spaghetti with Roasted Vegetable Ragu. A vibrant and satisfying vegan pasta dish.

7. Pan-Seared Portobello Mushroom Burger. A savory and filling vegetarian burger option.

8. Chickpea and Spinach Curry. A nutritious and aromatic curry perfect for a weeknight dinner.

9. Spaghetti with Grilled Aubergine and Tomato Sauce. A classic and comforting vegetarian pasta dish.

10. Roasted Butternut Squash and Sage Risotto. A creamy and comforting vegetarian main course.

11. Lentil and Mushroom Bolognese. A hearty and rich vegetarian take on the classic Bolognese.

12. Roasted Vegetable and Lentil Tagliatelle. A creative and delicious vegetarian take on the classic pasta dish.

13. Lentil and Vegetable Stew. A hearty and comforting vegetarian stew perfect for a cold winter's day.

14. Spaghetti with Lentil and Mushroom Bolognese. A hearty and rich vegetarian take on the classic Bolognese.

15. Vegan Lentil and Mushroom Meatballs. A tasty and satisfying vegan alternative to traditional meatballs.

16. Roasted Vegetable and Chickpea Wrap. A healthy and convenient vegetarian wrap option.

17. Grilled Aubergine and Red Pepper Salad. A flavorful and refreshing salad perfect for any time of the year.

18. Roasted Vegetable and Lentil Polpette. A creative and delicious vegan alternative to traditional meatballs.

19. Pork Ribs. Pork Ribs smothered with Kentucky style BBQ sauce.

20. Rustica Chorizo Pizza. Chorizo salami, torn wild boar and pork meatballs, smoked mozzarella and baby sunblush tomatoes.

21. Supreme Pizza. Cheese and tomato, onion, mushroom, fresh basil, olive and garlic oil.

22. Halloumi Sticks with Mayo Avocado Dip. Chunky sticks of grilled halloumi cheese with a chilli jam and mayo avocado dip.

23. Four Cheese Margherita. Mozzarella, mascarpone, fontal and grana cheese on a tomato base.

24. Bufala Caprese. Specialty tomatoes and drizzle of roasted garlic in extra virgin olive oil with Buffalo mozzarella.

25. Chicken Curry Ramen. Japanese fried chicken & noodles in a delicious curry broth.

26. Chicken Katsu Curry. Succulent chicken in a crispy Japanese panko crumb with mild curry sauce, pickles and steamed rice.

27. Pork Porchetta. Slow-roasted pork belly in herbed red wine sauce, served with roasted new potatoes and broccoli.

28. Butterfly Chicken Burger. Two succulent chicken breasts joined by crispy skin, packaged in a Portuguese roll and topped with tomato and lettuce.

29. Beer Battered Fish with Chips. with frites and pea & mint puree.

30. Falafel with Tahini. Our signature recipe, served with a tahini dip.

31. Panchetta Carbonara. crispy pancetta and asparagus in a velvety sauce made with mascarpone, pecorino and Grana Padano cheese.

32. Butternut Squash Salad. Roasted butternut squash with feta cheese, pomegranate seeds, mixed leaf salad and watercress.

33. Aegean Slaw. Thinly shredded cabbage, carrot and red onion peppers, with an olive oil dressing.

34. Jack Fruit Arrabiata. Vegan jackfruit peppers, spicy harissa, roquito chilis & vegan Mozzarella cheese, finished with basil.

35. Vegan Meatballs. Vegan meatballs in a rich tomato and fennel sauce.

36. Lentil Linguine Ragu. Rich Italian lentil ragu with baby spinach, tomatoes, basil & pecorino cheese.

**Gemini 1.5 Pro**

1. Tofu Curry Ramen. A vegan take on our classic ramen, with crispy fried tofu in a fragrant curry broth.

2. Lentil Meatballs with Spaghetti . A lighter, plant-based version of our classic meatballs, packed with flavor.

3. Mushroom Bourguignon. A flavorful twist on a classic, featuring tender mushrooms in a rich red wine sauce.

4. Chicken Katsu Curry. Succulent chicken in a crispy Japanese panko crumb with mild curry sauce, pickles and steamed rice.

5. Pepperoni Melt. This one's got our triple cheese blend, tomato sauce and pepperoni.

6. Chickpea Meatballs with Spaghetti. A hearty and flavorful vegetarian option, featuring tender chickpea meatballs.

7. Aromatic Duck Rolls. Aromatic roast duck, hoisin sauce, cucumber, spring onion, rolled in rice paper.

8. Mushroom and Lentil Chianti. A hearty and flavorful vegetarian stew featuring mushrooms and lentils in a rich Chianti red wine sauce.

9. Mushroom Porchetta. Savory mushrooms roasted to perfection in an herbed red wine sauce, served with roasted new potatoes and broccoli.

10. Cured Salmon Sashimi. Dazzling beetroot cured slices of salmon served with lemon. .

11. Mushroom and Tofu Tagliatelle. A vegetarian-friendly twist on our classic ragu: mushrooms and tofu in a rich red wine and beef dripping ragu.

12. Rustica Chorizo Pizza. Rustica Chorizo Pizza. Chorizo salami, torn wild boar and pork meatballs, smoked mozzarella and baby sunblush tomatoes. Topped with chilli threads and riserva cheese.

13. Butterfly Chicken Burger. Two succulent chicken breasts joined by crispy skin, packaged in a Portuguese roll and topped with tomato and lettuce.

14. Mushroom and Pepperoni Pizza. A classic pizza with your chocie of hot green, Roquito or jalapeno peppers.

15. Tofu Burger. A plant-based take on the classic burger, with flavorful tofu, fresh veggies, and tangy Dijon mayonnaise.

16. Beer Battered Fish with Chips. with frites and pea & mint puree.

17. Lentil Polpette. Oven-baked lentil patties in a pomodoro sauce with melted smoked mozzarella.

18. Steak. Sliced Black Angus rump steak with red onion chutney, watercress, and Dijon mayonnaise.

19. Cheese and Tomato Cannelloni. Pasta with béchamel, mozzarella, tomato sauce and Gran Milano cheese.

20. Supreme Pizza. Cheese and tomato, onion, mushroom, fresh basil, olive and garlic oil.

21. Goat's Cheese Calzone. Goats cheese, grilled aubergines, roasted peppers, oven roasted tomatoes.

22. Vegan Spaghetti Lentil Ragu. A hearty ragu of green lentils and mixed vegetables in a rich tomato and fennel sauce.

23. Jack Fruit Arrabiata. Vegan jackfruit peppers, spicy harissa, roquito chilis & vegan Mozzarella cheese, finished with basil.

24. Halloumi Sticks with Mayo Avocado Dip. Chunky sticks of grilled halloumi cheese with a chilli jam and mayo avocado dip.

25. Cheese Salad. Smoked cheddar, cheddar and sage stuffing.

26. Falafel with Tahini. Our signature recipe, served with a tahini dip.

27. Four Cheese Margherita. Mozzarella, mascarpone, fontal and grana cheese on a tomato base.

28. Vegan Meatballs. Vegan meatballs in a rich tomato and fennel sauce.

29. Bufala Caprese. Specialty tomatoes and drizzle of roasted garlic in extra virgin olive oil with Buffalo mozzarella.

30. Butternut Squash Salad. Roasted butternut squash with feta cheese, pomegranate seeds, mixed leaf salad and watercress.

31. Panchetta Carbonara. Crispy pancetta and asparagus in a velvety sauce made with mascarpone, pecorino and Grana Padano cheese.

32. Lentil Linguine Ragu. Rich Italian lentil ragu with baby spinach, tomatoes, basil & pecorino cheese.

33. Aegean Slaw. Thinly shredded cabbage, carrot and red onion peppers, with an olive oil dressing.

34. Goat's Cheese Salad. Goat's cheese and balsamic onion crostinis on winter baby kale, red pepper, cucumber, plum tomatoes.

35. Harusami Aubergine. Fried slices of aubergine with garlic & ginger sesame soy.

36. Inari Taco. Golden tofu pockets filled with sticky sushi rice, avocado salsa & vegan sriracha mayo.

**Claude 3.5 Sonnet**

1. Tofu curry ramen. Crispy fried tofu and noodles in a rich curry broth, topped with fresh pak choi and tangy pickled onions.

2. Mushroom ribs. Meaty mushrooms smothered in Kentucky style BBQ sauce, offering a plant-based twist on a classic.

3. Lentil bourguignon burger. Hearty lentil patty topped with crispy bacon, red wine braised onions, and melted Raclette cheese.

4. Tofu katsu curry. Crispy panko-crusted tofu with mild curry sauce, tangy pickles, and steamed rice.

5. Cheese and mushroom melt. Triple cheese blend melted over savory mushrooms and tangy tomato sauce.

6. Lentil meatballs with spaghetti. Handmade lentil patties grilled and topped with creamy Greek yoghurt, tomato sauce, and onions.

7. Mushroom spring rolls. Aromatic roasted mushrooms with hoisin sauce, cucumber, and spring onion, rolled in delicate rice paper.

8. Slow cooked lentils in Chianti sauce. Lentils and shallots slow cooked in rich Chianti red wine sauce with a cheesy dumpling.

9. Mushroom porchetta. Slow-roasted mushrooms in herbed red wine sauce, served with roasted new potatoes and broccoli.

10. Tofu sashimi. Dazzling beetroot cured slices of tofu served with zesty lemon.

11. Lentil and mushroom tagliatelle. A lentil, mushroom, and red wine ragu with tagliatelle, fresh bufala mozzarella and crispy sage.

12. Rustica mushroom pizza. Savory mushrooms, lentil meatballs, smoked mozzarella and baby sunblush tomatoes, topped with chilli threads and riserva cheese.

13. Tofu burger. Two succulent tofu fillets in a crispy coating, packaged in a Portuguese roll and topped with tomato and lettuce.

14. Mushroom hot pizza. Savory mushrooms, mozzarella and tomato, with your choice of hot green, Roquito or jalapeno peppers.

15. Lentil burger. Hearty lentil patty with baby gem lettuce and plum tomato in a chargrilled brioche bun with Dijon mayonnaise.

16. Beer battered tofu with chips. Crispy beer-battered tofu with frites and refreshing pea & mint puree.

17. Lentil polpette. Oven-baked herby lentil meatballs in a pomodoro sauce with melted smoked mozzarella.

18. Grilled mushroom steak. Sliced grilled mushroom steak with red onion chutney, watercress, and Dijon mayonnaise.

19. Mushroom cannelloni. Pasta rolls filled with savory mushrooms, topped with béchamel, mozzarella and Gran Milano cheese.

20. Supreme veggie pizza. A medley of cheese, tomato, onion, mushroom, fresh basil, olive and garlic oil.

21. Goat's cheese calzone. Goat's cheese, grilled aubergines, roasted peppers, and oven roasted tomatoes in a folded pizza pocket.

22. Lentil spaghetti ragu. A hearty ragu of green lentils and mixed vegetables in a rich tomato and fennel sauce.

23. Jackfruit arrabiata. Spicy jackfruit and peppers with harissa, roquito chilis & vegan mozzarella cheese, finished with basil.

24. Halloumi sticks with avocado dip. Chunky sticks of grilled halloumi cheese with a chilli jam and creamy mayo avocado dip.

25. Cheese salad. A variety of cheeses including smoked cheddar and cheddar, with aromatic sage stuffing.

26. Falafel with tahini. Our signature chickpea falafel recipe, served with a creamy tahini dip.

27. Four cheese margherita. A blend of mozzarella, mascarpone, fontal and grana cheese on a classic tomato base.

28. Lentil meatballs. Savory lentil meatballs in a rich tomato and fennel sauce.

29. Tofu caprese. Marinated tofu with specialty tomatoes and a drizzle of roasted garlic in extra virgin olive oil with Buffalo mozzarella.

30. Butternut squash salad. Roasted butternut squash with tangy feta cheese, pomegranate seeds, mixed leaf salad and peppery watercress.

31. Mushroom carbonara. Savory mushrooms and crisp asparagus in a velvety sauce made with mascarpone, pecorino and Grana Padano cheese.

32. Lentil linguine ragu. Rich Italian lentil ragu with baby spinach, tomatoes, basil & pecorino cheese.

33. Aegean slaw. Thinly shredded cabbage, carrot and red onion peppers, with a light olive oil dressing.

34. Goat's cheese salad. Goat's cheese and balsamic onion crostinis on winter baby kale, red pepper, cucumber, and plum tomatoes.

35. Harusami aubergine. Fried slices of aubergine with aromatic garlic & ginger sesame soy sauce.

36. Inari taco. Golden tofu pockets filled with sticky sushi rice, avocado salsa & spicy vegan sriracha mayo.

**GPT-4o**

1. Tofu Curry Ramen. Japanese fried tofu & noodles in a delicious curry broth. Topped with pak choi & pickled onions.

2. Vegetarian Ribs. Grilled tofu slathered in rich, Kentucky-style BBQ sauce. .

3. Vegetarian Bourguignon Burger. Lentil patty burger topped with crispy mushrooms, red wine braised onions, Raclette cheese.

4. Tofu Katsu Curry. Succulent tofu in a crispy Japanese panko crumb with mild curry sauce, pickles, and steamed rice.

5. Mushroom Melt. This one's got our triple cheese blend, tomato sauce, and grilled mushrooms.

6. Chickpea Meatballs with Spaghetti. Handmade chickpea patties grilled and topped with Greek yoghurt, tomato sauce, and onions.

7. Aromatic Tofu Rolls. Aromatic roast tofu, hoisin sauce, cucumber, spring onion, rolled in rice paper.

8. Slow Cooked Mushroom in Chianti Sauce. Portobello mushrooms and shallots slow cooked in rich Chianti red wine sauce with a cheesy dumpling.

9. Lentil Porchetta. Slow-roasted lentil loaf in herbed red wine sauce, served with roasted new potatoes and broccoli.

10. Beetroot Cured Tofu Sashimi. Dazzling beetroot cured slices of tofu served with lemon.

11. Mushroom and Lentil Tagliatelle. A mushroom and lentil ragu with tagliatelle, fresh bufala mozzarella, and crispy sage.

12. Rustica Tofu Pizza. Grilled tofu, smoked mozzarella, baby sunblush tomatoes, topped with chilli threads and riserva cheese.

13. Grilled Tofu Burger. Two succulent pieces of grilled tofu joined by a crispy skin, packaged in a Portuguese roll and topped with tomato and lettuce.

14. Vegetarian Hot Pizza. Grilled tofu, mozzarella, and tomato, with your choice of hot green, Roquito, or jalapeno peppers.

15. Lentil Burger. 6oz lentil patty with baby gem lettuce and plum tomato in a chargrilled brioche bun with Dijon mayonnaise.

16. Beer Battered Tofu with Chips. Beer battered tofu with frites and pea & mint puree.

17. Wild Mushroom Polpette. Oven-baked herby wild mushroom and lentil meatballs in pomodoro sauce with melted smoked mozzarella.

18. Vegetarian Steak. Grilled portobello mushrooms with red onion chutney, watercress, and Dijon mayonnaise.

19. Cheese Cannelloni. Pasta with béchamel, mozzarella, and Gran Milano cheese.

20. Supreme Vegetable Pizza. Cheese and tomato, onion, mushroom, fresh basil, olive oil, and garlic oil.

21. Goat's Cheese Calzone. Goat's cheese, grilled aubergines, roasted peppers, oven roasted tomatoes in a folded pizza.

22. Lentil Spaghetti Ragu. A hearty ragu of green lentils and mixed vegetables in a rich tomato and fennel sauce.

23. Jackfruit Arrabbiata. Vegan jackfruit, spicy harissa, roquito chilis, and vegan mozzarella cheese, finished with basil.

24. Halloumi Sticks with Mayo Avocado Dip. Chunky sticks of grilled halloumi cheese with a chilli jam and mayo avocado dip.

25. Cheese Salad. Smoked cheddar, cheddar, and sage stuffing served as a salad.

26. Falafel with Tahini. Our signature chickpea falafel served with a tahini dip.

27. Four Cheese Margherita Pizza. Mozzarella, mascarpone, fontal, and grana cheese on a tomato base.

28. Vegetarian Meatballs. Lentil meatballs in a rich tomato and fennel sauce.

29. Bufala Caprese. Specialty tomatoes and roasted garlic in extra virgin olive oil with Buffalo mozzarella.

30. Butternut Squash Salad. Roasted butternut squash with feta cheese, pomegranate seeds, mixed leaf salad, and watercress.

31. Egg Carbonara. Velvety carbonara sauce made with egg, crispy pancetta, and asparagus. .

32. Lentil Linguine. Green lentils, baby spinach, tomatoes, basil, and pecorino cheese with linguine.

33. Aegean Slaw. Thinly shredded cabbage, carrot and red onion peppers with an olive oil dressing.

34. Goat's Cheese Salad. Goat's cheese and balsamic onion crostinis on winter baby kale, red pepper, cucumber, plum tomatoes.

35. Harusami Mushroom. Fried mushrooms with garlic, ginger, and sesame soy.

36. Inari Taco. Golden tofu pockets filled with sticky sushi rice, avocado salsa, and vegan sriracha mayo.

**o1-preview**

1. Tofu Curry Ramen. A flavorful bowl of ramen with crispy fried tofu and aromatic curry broth.

2. Grilled Mushroom BBQ Skewers. Juicy and tender mushrooms glazed with smoky barbecue sauce.

3. Lentil Bourguignon. A hearty twist on a classic, featuring tender lentils in rich red wine sauce topped with melted Raclette cheese.

4. Egg Katsu Curry. Delight in a crispy egg coated in panko crumbs, served with a mild curry sauce.

5. Mushroom Melt. This melt features savory mushrooms with melted cheese and rich tomato sauce.

6. Chickpea Meatballs with Spaghetti. Handmade chickpea patties grilled and served with spaghetti and tangy tomato sauce.

7. Aromatic Tofu Rolls. Fresh rolls filled with aromatic tofu and crisp vegetables.

8. Slow Cooked Mushrooms in Chianti Sauce. Earthy mushrooms slow-cooked in rich Chianti wine sauce with a cheesy dumpling.

9. Stuffed Aubergine Porchetta. Roasted aubergine stuffed with aromatic herbs and served with hearty sides.

10. Tofu Sashimi. Delicate slices of tofu served with a squeeze of lemon.

11. Mushroom and Lentil Tagliatelle. A rich and hearty pasta with mushrooms and lentils in a savory ragu.

12. Rustica Veggie Pizza. A delightful pizza loaded with vegetables and smoky cheeses.

13. Grilled Halloumi Burger. A savory burger featuring grilled halloumi and fresh toppings.

14. Spicy Veggie Pizza. A fiery pizza topped with spicy peppers and melted cheese.

15. Gourmet Mushroom Burger. A hearty burger featuring grilled mushrooms and classic fixings.

16. Beer Battered Tofu with Chips. Crispy beer-battered tofu served with traditional sides.

17. Vegan Polpette. Herby vegan meatballs baked in flavorful tomato sauce with melted cheese.

18. Grilled Aubergine Stack. Layers of grilled aubergine with tangy red onion chutney.

19. Spinach and Cheese Cannelloni. Classic Italian dish with creamy spinach and cheese filling.

20. Supreme Veggie Pizza. A classic pizza loaded with fresh vegetables.

21. Goat's Cheese Calzone. A folded pizza filled with Mediterranean vegetables and goat's cheese.

22. Vegan Spaghetti Lentil Ragu. A hearty vegan spaghetti with rich lentil ragu.

23. Jackfruit Arrabiata. A spicy and flavorful dish with tender jackfruit.

24. Halloumi Sticks with Mayo Avocado Dip. Chunky grilled halloumi sticks served with a delicious dip.

25. Cheese Salad. A satisfying salad featuring a variety of cheeses.

26. Falafel with Tahini. Crispy falafel made from chickpeas served with creamy tahini.

27. Four Cheese Margherita. An indulgent pizza for cheese lovers.

28. Vegan Meatballs. Savory plant-based meatballs in a rich sauce.

29. Bufala Caprese. A fresh and delicious Italian classic.

30. Butternut Squash Salad. A vibrant salad with sweet and tangy flavors.

31. Asparagus Carbonara. A creamy pasta dish featuring tender asparagus.

32. Lentil Linguine Ragu. A comforting pasta with rich lentil sauce.

33. Aegean Slaw. A crisp and refreshing slaw with Mediterranean flavors.

34. Goat's Cheese Salad. A hearty salad with tangy goat's cheese and fresh vegetables.

35. Harusami Aubergine. Fried slices of aubergine in a flavorful sauce.

36. Inari Taco. Golden tofu pockets filled with rice and zesty salsa.

**o1-preview, Remove Descriptions**

1. Creamy Mushroom Tagliatelle. Mushrooms, tagliatelle pasta, mascarpone cheese, baby spinach, garlic oil, basil.

2. Vegetable Delight Pizza. Mozzarella, tomato sauce, grilled aubergines, roasted peppers, oven-roasted tomatoes, fresh basil.

3. Three Cheese Omelette. Eggs, smoked cheddar, mozzarella, grana cheese, red onions.

4. Eggplant Parmesan. Aubergine, tomato sauce, mozzarella, Grana Padano cheese, basil.

5. Mushroom and Goat's Cheese Omelette. Eggs, mushrooms, goat's cheese, garlic oil, basil.

6. Chickpea Curry with Rice. Chickpeas, mild curry sauce, pickles, steamed rice.

7. Spinach and Feta Stuffed Mushrooms. Mushrooms, baby spinach, feta cheese, garlic oil.

8. Lentil Veggie Burger. Lentils, chargrilled brioche bun, baby gem lettuce, plum tomato, Dijon mayonnaise.

9. Falafel Salad. Falafel, mixed leaf salad, cucumber, plum tomatoes, red onion, tahini dip.

10. Egg Shakshuka. Eggs, tomato sauce, red peppers, red onions, garlic oil, basil.

11. Vegetable and Tofu Stir-Fry. Tofu, noodles, pak choi, red peppers, red onions, garlic & ginger sesame soy.

12. Tofu Katsu Curry. Tofu in crispy Japanese panko crumb, mild curry sauce, pickles, steamed rice.

13. Lentil Stuffed Peppers. Bell peppers, lentils, tomato sauce, baby spinach, goat's cheese.

14. Butternut Squash and Feta Salad. Roasted butternut squash, feta cheese, pomegranate seeds, mixed leaf salad, watercress.

15. Vegan Meatball Sub. Vegan meatballs, pomodoro sauce, smoked mozzarella, Portuguese roll.

16. Chickpea and Spinach Curry. Chickpeas, baby spinach, mild curry sauce, steamed rice.

17. Tofu Curry Ramen. Fried tofu, noodles, curry broth, pak choi, pickled onions.

18. Mushroom and Lentil Bolognese. Mushrooms, lentils, tagliatelle pasta, tomato sauce, basil.

19. Supreme Pizza. Cheese and tomato, onion, mushroom, fresh basil, olive and garlic oil.

20. Chicken Curry Ramen. Japanese fried chicken & noodles in a delicious curry broth.

21. Pork Ribs. Pork Ribs smothered with Kentucky style BBQ sauce.

22. Butterfly Chicken Burger. Two succulent chicken breasts joined by crispy skin, packaged in a Portuguese roll and topped with tomato and lettuce.

23. Four Cheese Margherita. Mozzarella, mascarpone, fontal and grana cheese on a tomato base.

24. Panchetta Carbonara. crispy pancetta and asparagus in a velvety sauce made with mascarpone, pecorino and Grana Padano cheese.

25. Chicken Katsu Curry. Succulent chicken in a crispy Japanese panko crumb with mild curry sauce, pickles and steamed rice.

26. Rustica Chorizo Pizza. Chorizo salami, torn wild boar and pork meatballs, smoked mozzarella and baby sunblush tomatoes.

27. Beer Battered Fish with Chips. with frites and pea & mint puree.

28. Pork Porchetta. Slow-roasted pork belly in herbed red wine sauce, served with roasted new potatoes and broccoli.

29. Butternut Squash Salad. Roasted butternut squash with feta cheese, pomegranate seeds, mixed leaf salad and watercress.

30. Aegean Slaw. Thinly shredded cabbage, carrot and red onion peppers, with an olive oil dressing.

31. Cured Salmon Sashimi. Dazzling beetroot cured slices of salmon served with lemon.

32. Falafel with Tahini. Our signature recipe, served with a tahini dip.

33. Lentil Linguine Ragu. Rich Italian lentil ragu with baby spinach, tomatoes, basil & pecorino cheese.

34. Aromatic Duck Rolls. Aromatic roast duck, hoisin sauce, cucumber, spring onion, rolled in rice paper.

35. Vegan Spaghetti Lentil Ragu. A hearty ragu of green lentils and mixed vegetables in a rich tomato and fennel sauce.

36. Harusami Aubergine. Fried slices of aubergine with garlic & ginger sesame soy.

**o1-preview, Remove Preferences**

1. Tofu Curry Ramen. Japanese-style ramen with fried tofu and noodles in a delicious curry broth, topped with pak choi and pickled onions.

2. Creamy Mushroom Tagliatelle. An indulgent pasta dish featuring sautéed mushrooms and baby spinach in a creamy mascarpone sauce, tossed with tagliatelle and seasoned with garlic oil and fresh basil.

3. Falafel Salad. Our signature falafel served over a fresh mixed salad, with crunchy cucumbers, juicy tomatoes, red onions, and a creamy tahini dressing.

4. Lentil Stuffed Peppers. Roasted bell peppers stuffed with hearty lentils, tomato sauce, and fresh baby spinach, topped with creamy goat's cheese.

5. Mushroom and Goat's Cheese Omelette. A fluffy omelette filled with sautéed mushrooms and creamy goat's cheese, infused with garlic oil and fresh basil.

6. Eggplant Parmesan. Layers of tender aubergine baked with rich tomato sauce, melted mozzarella, and Grana Padano cheese, garnished with fresh basil.

7. Egg Shakshuka. Poached eggs simmered in a spiced tomato sauce with peppers and onions, garnished with fresh basil.

8. Butternut Squash and Feta Salad. Sweet roasted butternut squash and tangy feta cheese on a bed of fresh mixed greens, sprinkled with pomegranate seeds and watercress.

9. Lentil Veggie Burger. A hearty lentil-based veggie burger served on a toasted brioche bun with fresh lettuce, tomato, and tangy Dijon mayonnaise.

10. Vegetable and Tofu Stir-Fry. A vibrant stir-fry of crispy tofu and fresh vegetables tossed with noodles in a savory garlic and ginger sesame soy sauce.

11. Three Cheese Omelette. A fluffy omelette loaded with smoked cheddar, mozzarella, grana cheese, and sautéed red onions.

12. Tofu Katsu Curry. Succulent tofu coated in crispy panko crumbs, served with mild curry sauce, tangy pickles, and steamed rice.

13. Spinach and Feta Stuffed Mushrooms. Large mushrooms stuffed with sautéed baby spinach and creamy feta cheese, drizzled with garlic oil and baked to perfection.

14. Supreme Pizza. Cheese and tomato, onion, mushroom, fresh basil, olive and garlic oil.

15. Canelloni. pasta with béchamel, mozzarella and Gran Milano cheese.

16. Beer Battered Fish with Chips. with frites and pea & mint puree.

17. Harusami Aubergine. Fried slices of aubergine with garlic & ginger sesame soy.

18. Chicken Katsu Curry. Succulent chicken in a crispy Japanese panko crumb with mild curry sauce, pickles and steamed rice.

19. Vegan Meatballs. Vegan meatballs in a rich tomato and fennel sauce.

20. Inari Taco. Golden tofu pockets filled with sticky sushi rice, avocado salsa & vegan sriracha mayo.

21. Pork Ribs. Pork Ribs smothered with Kentucky style BBQ sauce.

22. Aegean Slaw. Thinly shredded cabbage, carrot and red onion peppers, with an olive oil dressing.

23. Vegan Spaghetti Lentil Ragu. A hearty ragu of green lentils and mixed vegetables in a rich tomato and fennel sauce.

24. Falafel with Tahini. Our signature recipe, served with a tahini dip.

25. Halloumi Sticks with Mayo Avocado Dip. Chunky sticks of grilled halloumi cheese with a chilli jam and mayo avocado dip.

26. Wild Boar Polpette. Oven-baked herby wild boar and pork meatballs in a pomodoro sauce with melted smoked mozzarella.

27. Chicken Curry Ramen. Japanese fried chicken & noodles in a delicious curry broth.

28. Butterfly Chicken Burger. Two succulent chicken breasts joined by crispy skin, packaged in a Portuguese roll and topped with tomato and lettuce.

29. Bufala Caprese. Specialty tomatoes and drizzle of roasted garlic in extra virgin olive oil with Buffalo mozzarella.

30. Butternut Squash Salad. Roasted butternut squash with feta cheese, pomegranate seeds, mixed leaf salad and watercress.

31. Panchetta Carbonara. crispy pancetta and asparagus in a velvety sauce made with mascarpone, pecorino and Grana Padano cheese.

32. Lentil Linguine Ragu. Rich Italian lentil ragu with baby spinach, tomatoes, basil & pecorino cheese.

33. Jack Fruit Arrabiata. Vegan jackfruit peppers, spicy harissa, roquito chilis & vegan Mozzarella cheese, finished with basil.

34. Aromatic Duck Rolls. Aromatic roast duck, hoisin sauce, cucumber, spring onion, rolled in rice paper.

35. Rustica Chorizo Pizza. Chorizo salami, torn wild boar and pork meatballs, smoked mozzarella and baby sunblush tomatoes.

36. Pork Porchetta. Slow-roasted pork belly in herbed red wine sauce, served with roasted new potatoes and broccoli.

**o1-preview, Remove Diversity**

1. Creamy Mushroom Tagliatelle. An indulgent pasta dish featuring sautéed mushrooms and baby spinach in a creamy mascarpone sauce, tossed with tagliatelle and seasoned with garlic oil and fresh basil.

2. Vegetable Delight Pizza. A delicious crispy pizza topped with mozzarella, tomato sauce, and a medley of grilled vegetables, finished with fresh basil.

3. Three Cheese Omelette. A fluffy omelette loaded with smoked cheddar, mozzarella, grana cheese, and sautéed red onions.

4. Eggplant Parmesan. Layers of tender aubergine baked with rich tomato sauce, melted mozzarella, and Grana Padano cheese, garnished with fresh basil.

5. Mushroom and Goat's Cheese Omelette. A fluffy omelette filled with sautéed mushrooms and creamy goat's cheese, infused with garlic oil and fresh basil.

6. Chickpea Curry with Rice. A flavorful and hearty chickpea curry served with steamed rice and accompanied by tangy pickles.

7. Spinach and Feta Stuffed Mushrooms. Large mushrooms stuffed with sautéed baby spinach and creamy feta cheese, drizzled with garlic oil and baked to perfection.

8.  Lentil Veggie Burger. A hearty lentil-based veggie burger served on a toasted brioche bun with fresh lettuce, tomato, and tangy Dijon mayonnaise.

9.  Falafel Salad. Our signature falafel served over a fresh mixed salad, with crunchy cucumbers, juicy tomatoes, red onions, and a creamy tahini dressing.

10. Egg Shakshuka. Poached eggs simmered in a spiced tomato sauce with peppers and onions, garnished with fresh basil.

11. Vegetable and Tofu Stir-Fry. A vibrant stir-fry of crispy tofu and fresh vegetables tossed with noodles in a savory garlic and ginger sesame soy sauce.

12. Tofu Katsu Curry. Succulent tofu coated in crispy panko crumbs, served with mild curry sauce, tangy pickles, and steamed rice.

13. Lentil Stuffed Peppers. Roasted bell peppers stuffed with hearty lentils, tomato sauce, and fresh baby spinach, topped with creamy goat's cheese.

14. Butternut Squash and Feta Salad. Sweet roasted butternut squash and tangy feta cheese on a bed of fresh mixed greens, sprinkled with pomegranate seeds and watercress.

15. Vegan Meatball Sub. Hearty vegan meatballs simmered in pomodoro sauce, topped with melted smoked mozzarella, served in a toasted Portuguese roll.

16. Chickpea and Spinach Curry. A nourishing curry of chickpeas and baby spinach simmered in a mild curry sauce, served with steamed rice.

17. Tofu Curry Ramen. Japanese-style ramen with fried tofu and noodles in a delicious curry broth, topped with pak choi and pickled onions.

18. Supreme Pizza. Cheese and tomato, onion, mushroom, fresh basil, olive and garlic oil.

19. Chicken Curry Ramen. Japanese fried chicken & noodles in a delicious curry broth.

20. Pork Ribs. Pork Ribs smothered with Kentucky style BBQ sauce.

21. Butterfly Chicken Burger. Two succulent chicken breasts joined by crispy skin, packaged in a Portuguese roll and topped with tomato and lettuce.

22. Four Cheese Margherita. Mozzarella, mascarpone, fontal and grana cheese on a tomato base.

23. Panchetta Carbonara. crispy pancetta and asparagus in a velvety sauce made with mascarpone, pecorino and Grana Padano cheese.

24. Chicken Katsu Curry. Succulent chicken in a crispy Japanese panko crumb with mild curry sauce, pickles and steamed rice.

25. Rustica Chorizo Pizza. Chorizo salami, torn wild boar and pork meatballs, smoked mozzarella and baby sunblush tomatoes.

26. Beer Battered Fish with Chips. with frites and pea & mint puree.

27. Pork Porchetta. Slow-roasted pork belly in herbed red wine sauce, served with roasted new potatoes and broccoli.

28. Butternut Squash Salad. Roasted butternut squash with feta cheese, pomegranate seeds, mixed leaf salad and watercress.

29. Aegean Slaw. Thinly shredded cabbage, carrot and red onion peppers, with an olive oil dressing.

30. Goat's Cheese Calzone. Goats cheese, grilled aubergines, roasted peppers, oven roasted tomatoes.

31. Inari Taco. Golden tofu pockets filled with sticky sushi rice, avocado salsa & vegan sriracha mayo.

32. Falafel with Tahini. Our signature recipe, served with a tahini dip.

33. Lentil Linguine Ragu. Rich Italian lentil ragu with baby spinach, tomatoes, basil & pecorino cheese.

34. Vegan Spaghetti Lentil Ragu. A hearty ragu of green lentils and mixed vegetables in a rich tomato and fennel sauce.

35. Harusami Aubergine. Fried slices of aubergine with garlic & ginger sesame soy.

36. Vegan Meatballs. Vegan meatballs in a rich tomato and fennel sauce.

**o1-preview, Remove Preferences and Diversity**

1. Tofu Curry Ramen. Japanese-style ramen with fried tofu and noodles in a delicious curry broth, topped with pak choi and pickled onions.

2. Chickpea Curry with Rice. A flavorful and hearty chickpea curry served with steamed rice and accompanied by tangy pickles.

3. Eggplant Parmesan. Layers of tender aubergine baked with rich tomato sauce, melted mozzarella, and Grana Padano cheese, garnished with fresh basil.

4. Mushroom and Lentil Bolognese. Hearty mushrooms and lentils cooked in a rich tomato sauce, served over tagliatelle pasta and garnished with fresh basil.

5. Vegan Meatball Sub. Hearty vegan meatballs simmered in pomodoro sauce, topped with melted smoked mozzarella, served in a toasted Portuguese roll.

6. Egg Shakshuka. Poached eggs simmered in a spiced tomato sauce with peppers and onions, garnished with fresh basil.

7. Lentil Veggie Burger. A hearty lentil-based veggie burger served on a toasted brioche bun with fresh lettuce, tomato, and tangy Dijon mayonnaise.

8. Mushroom and Goat's Cheese Omelette. A fluffy omelette filled with sautéed mushrooms and creamy goat's cheese, infused with garlic oil and fresh basil.

9. Chickpea and Spinach Curry. A nourishing curry of chickpeas and baby spinach simmered in a mild curry sauce, served with steamed rice.

10. Tofu Katsu Curry. Succulent tofu coated in crispy panko crumbs, served with mild curry sauce, tangy pickles, and steamed rice.

11. Falafel Salad. Our signature falafel served over a fresh mixed salad, with crunchy cucumbers, juicy tomatoes, red onions, and a creamy tahini dressing.

12. Lentil Stuffed Peppers. Roasted bell peppers stuffed with hearty lentils, tomato sauce, and fresh baby spinach, topped with creamy goat's cheese.

13. Vegetable Delight Pizza. A delicious crispy pizza topped with mozzarella, tomato sauce, and a medley of grilled vegetables, finished with fresh basil.

14. Butternut Squash and Feta Salad. Sweet roasted butternut squash and tangy feta cheese on a bed of fresh mixed greens, sprinkled with pomegranate seeds and watercress.

15. Vegetable and Tofu Stir-Fry. A vibrant stir-fry of crispy tofu and fresh vegetables tossed with noodles in a savory garlic and ginger sesame soy sauce.

16. Creamy Mushroom Tagliatelle. An indulgent pasta dish featuring sautéed mushrooms and baby spinach in a creamy mascarpone sauce, tossed with tagliatelle and seasoned with garlic oil and fresh basil.

17. Three Cheese Omelette. A fluffy omelette loaded with smoked cheddar, mozzarella, grana cheese, and sautéed red onions.

18. Pork Porchetta. Slow-roasted pork belly in herbed red wine sauce, served with roasted new potatoes and broccoli.

19. Beer Battered Fish with Chips. with frites and pea & mint puree.

20. Inari Taco. Golden tofu pockets filled with sticky sushi rice, avocado salsa & vegan sriracha mayo.

21. Halloumi Sticks with Mayo Avocado Dip. Chunky sticks of grilled halloumi cheese with a chilli jam and mayo avocado dip.

22. Falafel with Tahini. Our signature recipe, served with a tahini dip.

23. Four Cheese Margherita. Mozzarella, mascarpone, fontal and grana cheese on a tomato base.

24. Wild Boar Polpette. Oven-baked herby wild boar and pork meatballs in a pomodoro sauce with melted smoked mozzarella.

25. Aromatic Duck Rolls. Aromatic roast duck, hoisin sauce, cucumber, spring onion, rolled in rice paper.

26. Rustica Chorizo Pizza. Chorizo salami, torn wild boar and pork meatballs, smoked mozzarella and baby sunblush tomatoes.

27. Lentil Linguine Ragu. Rich Italian lentil ragu with baby spinach, tomatoes, basil & pecorino cheese.

28. Pork Ribs. Pork Ribs smothered with Kentucky style BBQ sauce.

29. Canelloni. pasta with béchamel, mozzarella and Gran Milano cheese.

30. Butternut Squash Salad. Roasted butternut squash with feta cheese, pomegranate seeds, mixed leaf salad and watercress.

31. Jack Fruit Arrabiata. Vegan jackfruit peppers, spicy harissa, roquito chilis & vegan Mozzarella cheese, finished with basil.

32. Vegan Spaghetti Lentil Ragu. A hearty ragu of green lentils and mixed vegetables in a rich tomato and fennel sauce.

33. Harusami Aubergine. Fried slices of aubergine with garlic & ginger sesame soy.

34. Butterfly Chicken Burger. Two succulent chicken breasts joined by crispy skin, packaged in a Portuguese roll and topped with tomato and lettuce.

35. Vegan Meatballs. Vegan meatballs in a rich tomato and fennel sauce.

36. Aegean Slaw. Thinly shredded cabbage, carrot and red onion peppers, with an olive oil dressing.

