# OpenReview forum: "What can large language models do for sustainable food?"
_ICML.cc/2025/Conference — ICML 2025 poster_

### Official Review · Reviewer_VYkF · 2025-03-02

**Overall Recommendation:** 3

**Summary:**

This paper explores the potential of large language models for sustainable food science. Specifically, this paper evaluates LLMs on four tasks, including experimental design, menu design, sensory profile prediction and recipe preference prediction. Then, this paper equips LLMs with combinatorial optimization to overcome the challenge (i.e., fail to balance emission target and satisfaction) in menu design tasks. Experimental results show initial success of LLMs for sustainable food science.

## update after rebuttal
I keep my scores unchanged, but I belive this is a good starting point for LLM in sustainable food science.

**Claims And Evidence:**

Yes.

**Essential References Not Discussed:**

None.

**Experimental Designs Or Analyses:**

Yes.

**Methods And Evaluation Criteria:**

Yes.

**Other Comments Or Suggestions:**

None.

**Other Strengths And Weaknesses:**

Overall, this is a solid paper with clear motivation and novel empirical findings. However, in my view, it may not be a strong fit for ICML. A substantial portion of the paper focuses on sustainable food science, which may not align closely with the core interests of the machine learning community. Additionally, the machine learning component primarily involves the application of large language models, serving more as a tool than contributing ML methodology. Given this, I believe the paper might be better suited for journals focused on food science, where the audience has deeper domain expertise, and where the work may have a greater impact.

That said, since ICML also encourages application-driven machine learning, I am inclined to recommend a borderline accept for this paper.

**Questions For Authors:**

None.

**Relation To Broader Scientific Literature:**

This paper explore novel perspectives of LLM application, i.e., sustainable food science.

**Theoretical Claims:**

N/A.

---

> ### Author Rebuttal · Authors · 2025-03-31
>
> We greatly appreciate your feedback.
> # Fit for ICML
> Thank you for raising this point. We believe that this paper is a strong fit for the Application-Driven Machine Learning track of ICML, defined as papers that “introduce novel methods, datasets, tasks, and/or metrics according to the needs of a real-world use case”.  We have made contributions along all of these dimensions. We have combined LLMs and optimization to address the sustainable menu design task (outperforming an expert human chef and several other baselines), introduced two datasets not previously studied by the ML community, and introduced four novel tasks and associated evaluation metrics.
>
> We believe that publication of our work in ICML would be mutually beneficial for the ML community, serving to raise awareness of this important testbed for ML algorithms, and the sustainable food community. Climate Change AI was founded in 2019 by AI researchers to encourage more research at the intersection of AI and sustainability, and has held 12 workshops at NeurIPS, ICML, and ICLR over the past 6 years, suggesting that the intersection of AI/ML and climate change mitigation is indeed of interest to the ML community. Additionally, in recent years, sustainability organizations such as the Good Food Institute, Food Systems Innovation, New Harvest, and the Bezos Earth Fund have called for research at the intersection of AI/ML and sustainable food, particularly novel sustainable protein sources. Food systems are responsible for one third of human-caused greenhouse-gas emissions, and thus represent an important part of AI/ML for climate change mitigation efforts. Despite this, no prior work on sustainable food, to our knowledge, has been published in an AI/ML conference. Our work is a step in this direction. We plan to publish followup work in domain journals to also ensure impact in the sustainable food community.
>
> # ML contribution
>
> We would like to clarify our ML contribution. For the recipe preference prediction, sensory profile prediction, and experimental design tasks, yes, we apply LLMs and analyze their performance, without contributing new methods. However, for the menu design task, we find that LLMs on their own do not adequately balance multiple constraints, and present a framework for combining LLMs with optimization techniques, which we find reduces emissions by 79% while maintaining patron satisfaction. We also provide an initial theoretical analysis of our framework (Proposition 6.1), showing that the error of our approach can be bounded as a function of the number of items selected and the maximum item-level prediction error of the LLM. While we currently only apply this framework to sustainable menu design, we believe it can also inspire future work. Besides the sustainable menu design task we study, examples of applications we are motivated by include health coaching (“Generate a diet and exercise plan for achieving my goals while meeting constraints on ingredients, cost, preparation time, and current injuries”), curriculum design (“Generate a curriculum that will maximize student engagement while meeting constraints on topics covered and class time”), and travel planning (“Generate a travel plan that meets my preferences and covers the following locations”). In these applications, both background knowledge (of nutrition, fitness, education, travel, human preferences, etc.) and optimization (to meet constraints) are necessary, and we show how to combine the background knowledge of LLMs with optimization tools.
>
> Our framework contributes to two strands of prior work in the ML and optimization communities. The first is the literature on LLMs for optimization modeling [1,2,3], which focuses on reducing barriers to the use of specialized optimization software via allowing users to specify optimization problems in natural language. Our work builds upon this literature, which assumes that the optimization problems are precisely specified. In our framework, LLMs are used to provide background knowledge on topics such as human preferences, to convert an imprecisely specified optimization problem (as is the case with many real-world planning, scheduling, decision making, etc. problems, such as the three mentioned above) to a precise optimization problem.  The second is the predict-then-optimize framework [4,5], as pointed out by reviewer XTjM. Here, our contribution is to add a generation step, in which an LLM generates the elements of the ground set (e.g. recipes, exercises, travel destinations) from which a solution is produced, as well as to apply LLMs for the prediction step.
>
> [1] AhmadiTeshnizi, A., et al. “OptiMUS.” ICML 2024.
>
> [2] Jiang, C., et al. “LLMOPT.” ICLR 2025.
>
> [3] Huang, C., et al. "ORLM." arXiv 2024.
>
> [4] Elmachtoub, A. et al. "Smart `Predict, then Optimize.’” Management Science 68.1 (2022).
>
> [5] Bertsimas, D. et al. "From Predictive to Prescriptive Analytics." Management Science 66.3 (2020).
>
> Thank you for your review!

---

> > ### Comment · Reviewer_VYkF · 2025-04-01
> >
> > Thanks for your response. I decide to keep my score unchanged.

---

> > > ### Author Response · Authors · 2025-04-03
> > >
> > > Thank you, VYkF, for reviewing our response. We greatly appreciate it.

---

### Official Review · Reviewer_YVD5 · 2025-03-06

**Overall Recommendation:** 4

**Summary:**

This paper explores the capabilities of Large Language Models (LLMs) in a set design of prediction tasks associated with sustainable diets (mainly plant-based) that were based on sustainable food literature and collaboration with domain experts. The overall objective of the tasks consists of generating low-emission menu designs (based on the associated emissions induced by the ingredients used in the recipes) that preserve human satisfaction. The authors' main contribution is a framework that evaluates how good LLMs are in a zero-shot setup and a novel approach that involves defining the problem as an LLM-guided combinatorial optimization for the task of menu design. They tested the framework with six different state-of-the-art LLMs. Their framework uses two food-related datasets: NECTAR (about the sensory evaluation of food products) and Food.com (about recipes); they also include a list of recipes from delivery applications. They included twenty plant-based food scientists in evaluating their method based on four metrics (accuracy, specificity, complementarity, and time saved) in a blinded randomized test in which the scientist collaborated with an anonymous scientist (the LLM or a peer). The answers were homogenized in style to avoid detecting the use of a model.

**Claims And Evidence:**

Most of the claims in the paper are relatively well supported; they remain conservative, but that's natural given the nature of the setup: LLMs have biases towards omnivore diets, and satisfaction score depends on a proxy evaluation through the LLMs.

The results are analyzed statistically using a confidence interval of 95% confidence for different tests. However, those relying on human evaluators ($n = 20$) might hinder the validity of their statistical tests, considering the rule of thumb of having at least $n = 30$ for the central limit theorem to hold.

**Essential References Not Discussed:**

Not to my understanding.

**Experimental Designs Or Analyses:**

Experiments and procedures are rigorous and follow a methodic evaluation process. Their experiments are Menu Design, Sensory Profile Prediction, and Recipe Preference Prediction. I don't identify any concern with the way they performed the experiments other than the confusion that could exist between metrics like accuracy (which has two definitions depending on the task).

**Methods And Evaluation Criteria:**

The datasets used in the paper are relevant to the proposed tasks.

The paper introduces a novel evaluation criteria that could have some caveats. It considers four metrics (accuracy, specificity, complementarity, and time saved) that rely on the open questions to a group of food scientists with moderate experience in plant-based food science and meat. This evaluation method seems reasonable given the nature of the tasks, but it could introduce biases given that it is entirely human-reliant.

Something similar happens with the satisfaction metric, which relies on a proxy obtained by the LLMs, which the paper points out as biased towards omnivore options.

There is confusion when the paper talks about accuracy; as for the task *Sensory Profile Prediction*, the definition corresponds to the ability of the LLM to describe the same sensory profile defined in the NECTAR dataset. For *Recipe Preference Prediction*, the accuracy corresponds to correctly estimating the recipe preference.

**Other Comments Or Suggestions:**

No additional comments.

**Other Strengths And Weaknesses:**

*Strengths*:

- The paper is well-written and easy to follow
- It is novel to consider LLMs in the field of sustainable food; it could enable the adoption of low-emission alternatives that preserve human satisfaction
- The inclusion of domain experts gives solidity to the proposed framework and experiments

*Weaknesses*:

- The theoretical bound in section 6 is not used; it could have contributed to some analysis of the traceability of the LLM-assisted optimization problem
- The results based on humans could be statically invalid given the number of participants they included
- The supplementary material offers resources to corroborate their framework, but it's hard to follow and doesn't provide any instruction

**Questions For Authors:**

1. What is the primary purpose of the theoretical bound introduced in 6?
2. Do you agree that the sample size of the expert-assisted metrics impacts the applicability of the traditional methods to create confidence intervals?

**Relation To Broader Scientific Literature:**

The topic of interest is addressed by prior work in the food science domain without a relationship to the domain of artificial intelligence. The paper proposes a novelty based on introducing LLMs in food science-related tasks of their design. The tasks were designed around the data availability, and datasets like NECTAR and Food.com played an important role. They also relied on prior work on how human preferences can be distilled from LLMs.

**Theoretical Claims:**

There is just one theoretical claim related to the LLM-guided combinatorial optimization problem in section 6.1. with a proof in the Appendix A. I believe that the theoretical derivation has a problem, overestimating the bound of $|f(\hat{x}^*) - f(x^*)|$, following their justification, the right bound should be: $|f(\hat{x}^*) - f(x^*)| \leq 2 K \epsilon$ the diversity term cancels out.

However, this claim does not seem particularly important in their methodology. The selection of K is based on matching the length of a menu from another work (line 365).

The explanation of equation 2 has a typo (Line 361) when referring to $C_i$ instead of $C_j$, which is important to highlight because index $i$ and $j$ are used to refer to selection and constraint, respectively. This confuses the understanding.

---

> ### Author Rebuttal · Authors · 2025-03-31
>
> We greatly appreciate your feedback.
> # Statistical analysis
> Thank you for raising this point. We note that we could have analyzed this data in alternative ways, given both the small sample size ($n=60$ ratings across 30 products) and clustering in the data (namely, pairs of products that were evaluated by the same food scientist, or where their associated experimental design was generated by the same food scientist). We ran additional statistical tests and found that the statistical significance of our findings are robust to all analysis variations we tested, which includes methods that do not rely on the central limit theorem.  We will include this in the revised paper. The table of $p$-values across dimensions and methods is below.  For all dimensions the mean value was higher (better) for o1-preview than the human food scientists.
>
> | Test | Accuracy | Complementarity | Specificity | Percent Time Saved |
> |-|-|-|-|-|
> | $t$-test (as in submitted paper) | 0.121 | 0.350 | **0.003** | **0.0002** |
> | Paired $t$-test (taking into account the pairing across products) | 0.125 | 0.368 | **0.003** | **0.0003** |
> | Wilcoxon signed-rank test (exact, for paired samples, nonparametric, does not rely on normality assumptions) | 0.078 | 0.284 | **0.005** | **0.0007** |
> | Linear mixed model with random effect for evaluator ID  | 0.075 | 0.993 | **0.001** | **0.0002** |
> | Permutation test (nonparametric, exact, does not rely on normality assumptions) | 0.140 | 0.392 | **0.003** | **0.0003** |
> | OLS where we control for product ID, evaluator ID, generator ID as fixed effects | 0.1812 | 0.416 | **0.001** | **0.0006** |
>
> Additionally, we would like to clarify our sample size for this task. Each data point was a rating of a human or LLM experimental design to improve a product. 30 products total were evaluated, for each of the two groups (human and LLM), yielding **$n=60$** ratings total. The total number of human evaluators was 20. As stated in Appendix D.1, in Phase 1 (generation phase), 15 food scientists were recruited to generate experimental designs for 2 products each, yielding 30 products with associated experimental designs. o1-preview was also prompted to generate experimental designs for each of the 30 products. Then, in Phase 2 (evaluation phase), another 15 food scientists (with overlap from Phase 1, but ensuring that no one evaluated their own designs) evaluated both human and LLM designs for 2 products each, on the four dimensions of accuracy, complementarity, specificity, and percent time saved. Then, for each of the four dimensions, a $t$-test was performed on the human vs. LLM scores, though as we show above the result is robust to alternative tests. We will make this more clear in the final version.
> # Evaluation criteria and satisfaction metric
> Regarding the comment “Something similar happens with the satisfaction metric, which relies on a proxy obtained by the LLMs, which the paper points out as biased towards omnivore options”, we clarify that for the menu design task, the satisfaction metric is based on responses of actual human participants, in response to the question, “How satisfied are you with your set of choices?” Please let us know if we misunderstood your comment; we will be happy to respond further.
>
> Across tasks, we use a combination of automated metrics and those that rely on human judgment. In the preference prediction tasks, we use automated metrics (accuracy relative to ground truth sensory panel or recipe rating data). In the menu design task, the emissions computation is automated.
>
> We agree that the human-reliant evaluation in the experimental design task could introduce bias, but we recruited 20 distinct expert human raters to minimize systematic bias.
> # Definitions of accuracy
> We appreciate this point and will make the distinction clear in our revision.
>
> # Theoretical claim and typo
> You are correct that the upper bound was overestimated, and should be $2Kϵ$ - thank you! We will fix this as well as the typo in the updated version.
>
> # Purpose of theoretical bound
> The theoretical bound provides the insight that the maximum error of our approach can be bounded based on the number of items selected (rather than e.g. the size of the ground set) and the maximum item-level error of the LLM. We agree that future work could assess how downstream performance varies depending on the quality of the preference prediction method, and compare empirical results with the theoretical bound.
>
> # Supplementary material
> We apologize for the insufficient documentation. Though we are not allowed to modify our submission at this stage, we commit to releasing a public GitHub repository that is well documented and provides full instructions for reproducing our results, other than information we cannot release due to IRB restrictions or the policy of the data provider (for the sensory panel data).
>
> Thank you for your thoughtful review!

---

> > ### Comment · Reviewer_YVD5 · 2025-04-08
> >
> > I appreciate the clarification provided by the authors regarding the statistical validity of their tests and the extra detail about how they were implemented. That allowed me to understand your original draft better. Thank you for considering the rest of the suggestions about form and clarity of content. Assuming you include these in the camera-ready version, I will update my score.

---

> > > ### Author Response · Authors · 2025-04-08
> > >
> > > Thank you, YVD5, for reviewing our response and updating your assessment. We greatly appreciate it. Yes, we will definitely incorporate all of your suggestions in the camera-ready version.

---

### Official Review · Reviewer_XTjM · 2025-03-12

**Overall Recommendation:** 2

**Summary:**

The paper investigates how LLMs can help reduce environmental impacts associated with food production. It establishes a typology of tasks relevant to sustainable food development, specifically focusing on design and prediction tasks at various levels (ingredients, recipes, and food systems). Evaluations of various LLMs across tasks reveal that LLMs allow to reduce the time required to generate experimental designs in sustainable protein formulation, outperforming expert human scientists across different metrics. However, they perform poorly in fine-grained tasks like menu design when simultaneously addressing climate impacts and human satisfaction. To overcome this limitation, the authors integrate LLMs with combinatorial optimization, achieving a substantial 79% emissions reduction in hypothetical restaurant scenarios without compromising customer satisfaction. The results underscore LLMs' strong potential, especially when complemented by optimization techniques, to accelerate sustainable food development and adoption.

**Claims And Evidence:**

The paper provides substantial empirical evidence that LLMs can effectively help reduce the time required by food scientists in sustainable protein experimental design tasks, while improving specificity scores. In addition, combining LLM predictions with combinatorial optimization can successfully reduce emissions (79%) with respect to the baseline while maintaining consumer satisfaction, as shown in their human subject experiments. These findings are supported by concrete experimental results involving expert evaluations and online surveys.

**Essential References Not Discussed:**

The paper's contributions align with the broader scientific literature on the "predict-then-optimize" method. However, the paper does not explicitly cite or discuss critical works related to this steam of literature in Operations Research and related journals, such as [Smart “Predict, then Optimize”, Elmachtoub and Grigas, 2021] and [From Predictive to Prescriptive Analytics, Bertsimas and Kallus, 2020].

**Experimental Designs Or Analyses:**

n/a

**Methods And Evaluation Criteria:**

The proposed methods and evaluation criteria make sense for the application. The integration of LLM predictions with combinatorial optimization addresses realistic trade-offs encountered in sustainable food design. The chosen datasets (NECTAR sensory panel and Food.com recipes) are appropriate. However, stronger baselines for menu design could further improve the evaluation, such as menus using lower-emission equivalent of the original menu (e.g., substituting beef with chicken) rather than only considering vegetarian or beef-free alternatives.

**Other Comments Or Suggestions:**

n/a

**Other Strengths And Weaknesses:**

The paper is well-written and reads well. The experiments involving human feedback (particularly the evaluation with 20 expert food scientists and the online surveys with a total of 552 participants) represent significant empirical contributions. However, the optimization approach introduced in Section 6 (combining LLM-based predictions with combinatorial optimization) is not particularly innovative, as it essentially adopts the well-known predict-then-optimize methodology where the prediction model is an LLM. Furthermore, given that the quantities to be predicted are highly uncertain quantities, such as human preferences and ratings, the predict-then-optimize framework is known to perform poorly with point prediction compared to other more robust approaches accounting for prediction uncertainty (see [From Predictive to Prescriptive Analytics, Bertsimas and Kallus, 2020]). Additionally, the paper does not include significant theoretical contributions.

**Questions For Authors:**

n/a

**Relation To Broader Scientific Literature:**

The paper's key contributions are (1) the application of LLMs for scientific discovery and human preferences modeling to the sustainable food domain, and (2) combining LLMs with optimization methods.
For (1), the paper extends prior work on LLM-based modeling of human behavior to the sustainable food domain, not currently explored in the literature. However, as discussed in the results sections, the observations align with the existing food science literature, resulting in few novel findings.
Regarding (2), the paper cites [Yang et al., 2024] and [AhmadiTeshnizi et al., 2024] using LLMs for mathematical optimization problems. However, the coupling of LLM and integer quadratic programming is more closely related to the stream of data-driven optimization, in which the framework introduced in this paper has already been thoroughly studied.

**Theoretical Claims:**

The proof of Proposition 6.1 is correct. This proposition is the only theoretical claim.

---

> ### Author Rebuttal · Authors · 2025-03-31
>
> We greatly appreciate your feedback.
> # Baselines
> We have added two baselines (expert chef, and the beef to chicken substitution you suggest), both of which we outperform, and three ablations (remove preferences component, remove diversity component, and remove both) for the menu design task. The chef was given the same set of instructions as the LLMs, but with more time: one hour. (Our o1-preview+IQP procedure takes ~3.5 minutes to run). We found that the chef did not always meet the ingredient availability constraint. Additionally, while the chef generated creative recipes, they lacked a strategy for reducing emissions, choosing to preserve several high-emissions dishes.
>
> Below are our new results, compared with the original menu and our proposed method o1-preview+IQP. SD is in parentheses.
>
> |                                                           | Emissions     | Satisfaction |
> |-|-|-|
> | Original                                                  | 44.91 (38.83) | 8.40 (1.86)  |
> | o1-preview+IQP ($\lambda=100$)                               | 8.70 (7.06)   | 8.44 (2.20)  |
> | o1-preview+IQP ($\lambda=0$)                                 | 9.75 (7.50)   | 8.16 (2.25)  |
> | o1-preview+IQP ($\lambda=100$, remove preferences) | 9.44 (7.26)   | 7.83 (2.56)  |
> | o1-preview+IQP ($\lambda=0$, remove preferences)   | 8.54 (6.72)   | 7.12 (2.88)  |
> | Expert chef                              | 41.28 (40.38) | 8.20 (2.51)  |
> | Replace beef with chicken                                 | 17.57 (15.07) | 8.04 (2.22)  |
> # Novelty of empirical findings
> We would like to clarify that we do have several novel findings, most notably that 1) LLMs can outperform expert food scientists in the generation of experimental designs for improving sustainable protein formulations and 2) our LLM+IQP algorithm outperforms an expert human chef and several baselines in the design of sustainable menus. Additionally, we characterize performance of LLMs on preference prediction tasks, finding that they can be useful for coarse-grained prediction but that further work is needed for fine-grained prediction. This is in addition to our novel task formulations.
> # Related work
> Thank you so much for pointing us to the “predict-then-optimize” literature. We will cite it in the updated version. The ML contribution of our paper can be viewed as extending the predict-then-optimize framework by incorporating a component where the elements of the ground set (e.g. recipes) are generated, and applying LLMs to both the generation and prediction steps. Incorporating measures of LLM uncertainty and conducting additional theoretical analysis of this **generate**-predict-then-optimize framework is an ongoing area of work.
>
> Here we also address Vdt3’s references. Regarding [1] and [2] from their review, our framework differs in that the LLM is used to provide background knowledge on topics such as human preferences, to convert an imprecisely specified optimization problem (as is the case with many real-world planning, scheduling, decision making, etc. problems) to a precise optimization problem. Besides the sustainable menu design task we study, examples of applications include health coaching (“Generate a diet and exercise plan for achieving my goals while meeting constraints on ingredients and current injuries”), curriculum design (“Generate a curriculum that will maximize student engagement while meeting constraints on topics covered and class time”), and travel planning (“Generate a travel plan that meets my preferences and covers the following locations”). These applications cannot be readily addressed by the work in [1] and [2], which assume that the provided optimization problem is fully specified. Moreover, our work adds to the set of applications in this literature. Regarding Vdt3’s reference [3], we appreciate the pointer to this comprehensive survey. Our work differs from this literature due to our focus on leveraging background knowledge of LLMs to expand the class of reasoning problems that can easily be addressed with optimization tools.
> # Our contributions
> The Application-Driven ML reviewer instructions state, “Originality need not mean wholly novel methods. It may mean a novel combination of existing methods to solve the task at hand, a novel dataset, or a new way of framing tasks or evaluating performance so as to match the needs of the user.” We have made contributions along all of these dimensions. We have combined LLMs and optimization to address the sustainable menu design task (outperforming an expert human chef and several other baselines), introduced two datasets not previously studied by the ML community, and introduced four novel tasks and associated evaluation metrics.
>
> We agree that we do not have significant theoretical contributions. However, we do not believe that this is required for the Application-Driven ML track, which notes that the form of the contribution may differ from papers in the main track.
>
> Thanks again!

---

### Official Review · Reviewer_VdT3 · 2025-03-14

**Overall Recommendation:** 3

**Summary:**

This paper explores the potential of Large Language Models (LLMs) in addressing sustainability challenges in food systems. The authors define a typology of tasks related to sustainable food, including design and prediction tasks at the ingredient, recipe, and system levels. They evaluate six LLMs on four specific tasks: sustainable protein design, menu design, sensory profile prediction, and recipe preference prediction. The study finds that LLMs can significantly reduce the time spent on experimental design tasks compared to human experts but struggle with tasks requiring the balancing of multiple constraints, such as menu design. To address this, the authors propose a framework that integrates LLMs with combinatorial optimization, demonstrating a notable reduction in emissions.

**Claims And Evidence:**

Yes

**Essential References Not Discussed:**

On the related topic of LLM for optimization. The paper EoH [1] published in ICML 2024 integrates LLM in a search framework for optimization including combinatorial optimization. LLMOPT [2] adopts LLM for optimization modeling. The survey paper [3] also provides a more systematic discussion of this topic.

[1]Evolution of heuristics: Towards efficient automatic algorithm design using large language model, ICML 2024
[2]LLMOPT: Learning to Define and Solve General Optimization Problems from Scratch, ICLR 2025
[3]A systematic survey on large language models for algorithm design, 2024

**Experimental Designs Or Analyses:**

Yes. It lacks real-world testing.

**Methods And Evaluation Criteria:**

Yes

**Other Comments Or Suggestions:**

Refer to questions

**Other Strengths And Weaknesses:**

Strengths:
The paper pioneers the application of LLMs to sustainable food systems.
The authors evaluate multiple LLMs across diverse tasks, providing a robust comparison of their capabilities.
The proposed framework combining LLMs with combinatorial optimization is an interesting approach that addresses LLMs' limitations in mathematical reasoning and multi-constraint optimization.


Weaknesses:
The datasets used, particularly for sensory profile prediction, are relatively small and may not fully capture the complexity of human taste preferences.
The menu design task is evaluated in a hypothetical setting, and the proposed solutions have not been tested in real-world environments.
The authors acknowledge that they did not extensively explore prompt engineering, which could potentially improve LLM performance.

**Questions For Authors:**

How did you determine the optimal hyperparameters (e.g., λ for diversity) in the combinatorial optimization framework, and what sensitivity analysis was performed to ensure robustness?

Could you provide more details on the Ratcliff/Obershelp sequence matching algorithm used for recipe similarity, and why it was chosen over other similarity metrics?

What specific techniques or architectures were used to standardize the style of LLM and human responses in the experimental design task, and how did this affect the evaluation outcomes?

How did you handle the trade-off between computational complexity and solution quality in the integer quadratic programming (IQP) formulation for menu design?

How do you design the prompts? Can other advanced prompt engineering techniques improve the performance?

**Relation To Broader Scientific Literature:**

I think the paper is application-orientated. The method and results are mainly for food systems.

**Theoretical Claims:**

No proof

---

> ### Author Rebuttal · Authors · 2025-03-31
>
> We greatly appreciate your feedback.
> # Theoretical claims
> A proof is in Appendix A. YVD5 noted that the bound can be improved, which we will incorporate into the final version.
> # Related work
> Thank you, we will cite these works. Please see our response to XTjM, where we discuss these papers; we did not have space to include it here.
> # Results mainly for food systems
> Please see our response to VYkF explaining why we believe our paper is a strong fit for the Application-Driven ML track of ICML.
> # Dataset size and complexity of taste preferences
> The Food.com dataset contains 522,517 recipes and 1,401,982 reviews. The recipes contain at least 50, and on average 121.34 reviews. The mean SD of ratings across the recipes is 1.21 (on a 5 pt scale), suggesting diversity of ratings. The NECTAR dataset consists of 47 products, with at least 100 human sensory evaluations per product along 21 dimensions, with 1150 distinct human taste testers. The subjects were restricted to American omnivores, and we will add to the Limitations section that our results may not generalize outside of this population. However, the sample was designed to be representative of American omnivores as a whole, e.g. was diverse along dimensions of age, gender, and race. Please see the NECTAR 2024 report for exact statistics (we will include this in the final version). The mean SD of the ratings across products for the dimensions we study in our sensory profile prediction task ranged from 0.89 (Greasiness) to 1.84 (Overall Satisfaction) - both on a 7 pt scale - suggesting diversity of ratings.
>
> Finally, we note that the NECTAR dataset is expanding, and will reach 500 products and 50,000 sensory evaluations by 2026, further increasing the potential impact of introducing this dataset to the ML community. It has already expanded by 126 products (each with at least 100 sensory evaluations) since the submission of this paper. We plan to re-run our experiments on this expanded dataset.
> # Hypothetical setting
> Please see our response to 7evN.
> # Optimal $\lambda$, and sensitivity analysis
> We placed the maximum weight on diversity (for our input data $\lambda=100$ achieves the same result as $\lambda=\infty$) to ensure a diverse set of options for online participants, who may have allergies or other constraints. Assuming a high quality ground set, we think this is a reasonable default choice in general.
> We studied the sensitivity of the optimal menu to the value of $\lambda$. When decreasing $\lambda$ with a step size of 1, the generated menus are identical until $\lambda=3$, at which point the optimal menu changes by 2 recipes (out of 36), suggesting robustness to the exact choice of $\lambda$. Future work could use the LLM-as-judge framework to optimize the value of $\lambda$ without running human subjects experiments.
>
> | $\lambda$ | Set Difference (Num. Recipes That Differ from $\hat{S}*$ for $\lambda=100$) |
> |-----------|-----------|
> | 3 | 2 |
> | 2 | 2|
> | 1 | 2 |
> | 0.5 | 6|
> | 0 | 12|
> # Ratcliff/Obershelp
> This algorithm computes similarity between $S_1$ and $S_2$ as $\frac{2K_m}{|S_1| + |S_2|}$. $K_m$ is the number of matching characters, defined as the length of the longest common substring (LCS) plus recursively the number of matching characters on both sides of the LCS. More details can be found in the difflib documentation. We used this algorithm for recipe similarity because it is a simple and widely used algorithm that allows for partial matches. As a sensitivity analysis, we tested the Levenshtein distance, another commonly used method for text similarity, and found that the optimal menu changed by 4 recipes (out of 36), suggesting that that performance boost, if any, would be limited. We will include this and other similarity metrics (e.g. those based on semantic similarity) in the final version.
> # Style standardization
> We used o1-preview to standardize the style. Our prompt is in Appendix D.1, Figure 5. As in [1], we did not assess how standardization affected the outcomes; in general, responses remained similar after the standardization step, with a few exceptions, e.g. where the humans referred to their personal experiences.
> # Trade-off between computational complexity and solution quality
> Our problem sizes are relatively small, with a ground set of size 56. We were thus able to solve our problem instances optimally in less than a second. Essentially, there was no tradeoff between computational complexity and solution quality for our use case.
> # Prompt engineering
> For style standardization, we adapted a prompt from prior work [1]. For the other tasks, the ML and culinary/food science members of our team collaborated to iteratively design the prompt.  We aimed to test LLMs’ potential off the shelf, which is why we didn't do extensive prompt engineering, as in [2].
>
> [1] Si, C. et al. "Can LLMs Generate Novel Research Ideas?" ICLR 2025.
>
> [2] Bulian, J., et al. “Assessing LLMs on Climate Information.” ICML 2025.
>
> Thank you for your review!

---

> > ### Comment · Reviewer_VdT3 · 2025-04-02
> >
> > Thank you for your responses. Most of my concerns have been addressed. I will adjust my score accordingly.

---

> > > ### Author Response · Authors · 2025-04-03
> > >
> > > Thank you, VdT3, for reviewing our response and updating your assessment. We greatly appreciate it.

---

### Official Review · Reviewer_7evN · 2025-03-14

**Overall Recommendation:** 4

**Summary:**

This paper investigates how LLMs can contribute to developing sustainable food options (e.g., reducing greenhouse gas emissions). The authors define a typology of design and prediction tasks for sustainable food at three resolutions (ingredients, recipes, and food systems ). The paper focuses on four tasks: 1) Experimental Design, 2) Menu Design, 3) Sensory Profile Prediction, 4) Recipe Preference Prediction.

Method-wise, the main contribution is the proposed integration of the LLMs’ background knowledge (especially about human preferences) with traditional combinatorial optimization to tackle real-world constraints.

**Claims And Evidence:**

The paper's main claim is that LLMs can reduce the time and effort needed to design more sustainable plant-based protein formulations. This is backed by experiments which show that the LLM outperformed or equaled the human baseline on specificity and time saved, evaluated by food scientists.

LLMs alone handle multiple constraints poorly when designing menus (tend to produce fully vegan menus) and combining LLM-based preference estimates with a combinatorial optimization approach can yield large emissions cuts while keeping customer satisfaction. This is backed by Figure 2.

In general, the claims are adequately backed by the experiments. However, I would like to see more ablations and analysis on the proposed method (i.e., o1-review-IQP).

**Essential References Not Discussed:**

N/A

**Experimental Designs Or Analyses:**

I would like to see more analysis/ablations on the proposed method (i.e, o1-preview-IQP).

**Methods And Evaluation Criteria:**

Strengths:
- The methods and evaluation are straightforward and makes intuitive sense

Weaknesses:
- No ground-truth validation for menu-based emissions reductions (what if actual diners select differently in practice?)
- LLM’s generated recipes are not tested for real taste, only inferred preferences
- Fine-grained sensory predictions lack stronger benchmarks (e.g., simpler regression models based on molecular food science)

**Other Comments Or Suggestions:**

N/A

**Other Strengths And Weaknesses:**

See Methods And Evaluation Criteria.

**Questions For Authors:**

N/A

**Relation To Broader Scientific Literature:**

I think the paper adequately positions itself in the broader scientific literature.

**Theoretical Claims:**

The paper does propose theoretical claims and provide proofs in the appendix, although the correctness of the proof would not largely affect the claims and conclusions of the paper.

---

> ### Author Rebuttal · Authors · 2025-03-31
>
> We greatly appreciate your feedback.
> # Ablations and analysis
> Our submission included two ablations - removing the IQP component entirely (just prompting o1-preview directly to revise the menu) and removing the descriptions. We add three other ablations: removing the estimated preferences, removing the diversity term ($\lambda=0$), and removing both. Satisfaction declines ($p=0.01$) when both are removed. We also add two baselines: expert chef and replacing beef with chicken. Our method improves upon both. Please see our response to XTjM for our results table, which we did not have space to replicate in this response.
> # Ground-truth validation for menu-based emissions reductions
> Our setting, an online experiment, is commonly used in the sustainable food literature [1,2,3,4] since it mimics popular online food delivery platforms. We acknowledge that some prior work tried to align incentives via delivering the meals to 1 in 30 participants [3] or providing food vouchers to 1 in 20 participants [4]. Given the complexities of replicating this setup for LLM generated recipes, we leave this to future work. Regardless, it is not guaranteed that conclusions based on online environments will generalize to real-world food environments. We will note this in the Limitations section.
>
> [1] Attwood, S. et al. "Menu engineering to encourage sustainable food choices when dining out: An online trial of priced-based decoys." Appetite 149 (2020).
>
> [2] Weijers, RJ, et al. "Nudging towards sustainable dining: Exploring menu nudges to promote vegetarian meal choices in restaurants." Appetite 198 (2024).
>
> [3] Lohmann, PM., et al. "Choice architecture promotes sustainable choices in online food-delivery apps." PNAS Nexus 3.10 (2024).
>
> [4] Banerjee, S, et al. "Sustainable dietary choices improved by reflection before a nudge in an online experiment." Nature Sustainability 6.12 (2023).
> # LLM generated recipes are not tested for real taste, only inferred preferences
> Yes, in the menu design task, the generated recipes are not actually prepared and tasted by participants. This would involve a number of logistical and ethical challenges and is beyond the scope of the current study. We will note this in the Limitations section. We do use actual tasting data in our sensory profile prediction and experimental design tasks, and actual ratings in our recipe preference prediction task.
> # Sensory profile prediction task lacks stronger benchmarks
> Our baselines in the submitted version were constructed on the basis of the food science literature and the available nutritional information, e.g. for overall satisfaction and purchase intent, the average of normalized fat and sodium content is used. We agree that more sophisticated baselines could be tested.
>
> To address this, we have obtained an expert food scientist baseline for the Overall Satisfaction dimension of the sensory profile prediction task, which we think is the most relevant baseline for food science practitioners. The expert food scientist spent 2 hours on the task (approximately 1.5 minutes per pair). Statistically significant results (after Bonferroni correction) are in bold. We find that across all pairs for the Overall Satisfaction dimension, the expert food scientist does not achieve a statistically significant improvement over a random baseline, underscoring the difficulty of the task. The expert food scientist’s accuracy is similar to that of o1-preview, and both underperform our baseline based on the nutritional information and food science literature. We find that for coarse-grained prediction (quartile 4 of the ground truth preference gap), the expert food scientist again does not achieve a statistically significant improvement over a random baseline. However, both o1-preview and the nutritional baseline do outperform a random baseline (81% and 86% accuracy respectively), supporting the potential of automated methods, particularly for coarse-grained comparisons, which could save time in testing and development. We will include this result in the final version.
>
> | Data Subset (Quartile)                                                             | Expert Food Scientist  | o1-preview  | Nutritional Baseline  |   |
> |-|-|-|-|-|
> | All                                                                      | 0.65                   | 0.64        | 0.73                  |   |
> | Quartile 1                                                             | 0.62                   | 0.43        | 0.71                  |   |
> | Quartile 2                                                            | 0.60                   | 0.60        | 0.80                  |   |
> | Quartile 3                                                             | 0.77                   | 0.68        | 0.55                  |   |
> | Quartile 4 (largest gap in ground truth preferences; “coarse grained”) | 0.59                   | **0.81**        | **0.86**                  |   |
>
> Thank you for your support!

---

### Decision · Program_Chairs · 2025-05-01

**Decision:**

Accept (poster)

**Comment:**

This paper presents an application-driven study on the use of LLMs for sustainable food systems, introducing four tasks, real-world dataset, and a framework combining LLM outputs with combinatorial optimization. Authors demonstrate empirical benefits across multiple tasks, including outperforming expert chefs in a menu design setting. Reviewers generally appreciated the originality, domain relevance, and careful experimental design. Some concerns were raised about limited ML novelty and real-world testing, but most were resolved/mitigated by the rebuttal. The work fits well within the scope of the Application-Driven ML track and contributes a new benchmark domain of increasing societal relevance. AC recommends accepting the paper for the application track.